# Spatial fibroblast niches define Crohn's fistulae

Colleen McGregor[1,2,12], Xiao Qin[1,12], Marta Jagielowicz[1], Tarun Gupta[1,2], Zinan Yin[1], Verena Lentsch[1], David Fawkner-Corbett[1,3], Vy Wien Lai[1,4], Paula Gomez Castro[1], Esther Bridges[1], Chloe Hyun-Jung Lee[1,4], Huei-Wen Chuang[1,4], Lei Deng[1], Anna Aulicino[1], Renuka Teague[5], Sorayya Moradi[5], Jun Sung Park[1,4], Jeongmin Woo[1,4], Kexin Xu[1,4], Ruchi Tandon[6], Nicole Cianci[7], Jan Bornschein[1,2], Ling-Pei Ho[1], Paulina Siejka-Zielinska[1], Zoe Christoforidou[1], Sarah Hill[8], Johannes Lehmann[8], Rhea Kujawa[9], Paola Vargas Gutierrez[9], Carol Cheng[9], Maria Greco[9], Katherine Baker[10], Mark Bignell[10], Bruce George[10], Eve Fryer[6], Michael Vieth[11], Agne Antanaviciute[1,4]✉ & Alison Simmons[1,2]✉

Crohn's disease often presents with fistulae, abnormal tunnels that connect the intestine to the skin or other organs. Despite their profound effect on morbidity, the molecular basis of fistula formation remains unclear, largely owing to the challenge of capturing intact fistula tracts and their inherent heterogeneity[1–3]. Here we construct a subcellular-resolution spatial atlas of 68 intestinal fistulae spanning diverse anatomical locations. We describe fistula-associated epithelial, immune and stromal cell states, revealing abnormal zonation of growth factors and morphogens linked to establishment of tunnelling anatomy. We identify fistula-associated stromal (FAS) fibroblasts, which are assembled in concentric layers: a proliferative, lumen-adjacent zone beneath neutrophil and macrophage-rich granulation tissue, an active lesion core of FAS cells and a quiescent, pro-fibrotic outer zone. We examine the architecture of the extracellular matrix in the fistula tract and demonstrate that FAS populations associate with distinct collagen structures, exhibiting properties ranging from proliferation, migration and extracellular matrix remodelling to dense collagen deposition and fibrosis. We define niches supporting epithelialization of fistula tunnels and a FAS-like population that is detected at the base of ulcers in non-penetrating Crohn's disease. Our study demonstrates that common molecular pathways and cellular niches underpin fistulae across intestinal locations, revealing the cellular protagonists of fistula establishment and persistence. This resource will inform the development of model systems and interventions to mitigate aberrant fibroblast activity while preserving their regenerative properties in Crohn's disease.

Fistulating Crohn's disease (CD) is a phenotype of inflammatory bowel disease (IBD) with high morbidity, occurring in up to 30% of patients with IBD[1,4–6]. Heterogeneity in this manifestation of CD occurs at the level of anatomical location, morphology, symptom severity, co-existent luminal disease and response to medical or surgical interventions[3–5].

Current treatments primarily target the inflammatory component of CD, which improves symptoms in only a proportion of patients[7,8]. Surgical closure methods depend on the anatomy of the tract and may not be possible where fistulae exhibit complex branching morphology or specific anatomical locations. In severe refractory perianal fistulating CD, 31 to 49% of patients eventually undergo diverting ostomy, and symptoms persist in one-third of these patients, ultimately necessitating proctectomy[2]. Therefore, there is a pressing need to better define the molecular pathways that drive this sub-phenotype of CD with a view to developing medications that prevent the emergence or enhance the repair of fistula tracts.

A molecular understanding of CD fistula pathogenesis remains incomplete. Older literature alludes to a role for epithelial-to-mesenchymal transition of intestinal epithelial cells, upregulation of matrix metalloproteinases (MMPs) and overexpression of invasive molecules as contributing to fistula formation[3,9]. A recent study using a single-cell

[1]Medical Research Council Translational Immune Discovery Unit (MRC TIDU), Weatherall Institute of Molecular Medicine (WIMM), University of Oxford, Oxford, UK. [2]Translational Gastroenterology and Liver Unit, John Radcliffe Hospital, Oxford, UK. [3]Academic Paediatric Surgery Unit, Nuffield Department of Surgical Sciences, University of Oxford, Oxford, UK. [4]MRC WIMM Centre for Computational Biology, WIMM, Oxford, UK. [5]Oxford Centre for Histopathological Research, Oxford University Hospitals NHS Trust, Oxford, UK. [6]Department of Cellular Pathology, Oxford University Hospitals NHS Trust, Oxford, UK. [7]Center for Human Genetics, Nuffield Department of Medicine, University of Oxford, Oxford, UK. [8]Kennedy Institute of Rheumatology, University of Oxford, Oxford, UK. [9]MRC WIMM Advanced Single Cell OMICS Facility, WIMM, Oxford, UK. [10]Department of Colorectal Surgery, Oxford University Hospitals NHS Trust, Oxford, UK. [11]Institute of Pathology, Friedrich-Alexander University Erlangen-Nuremberg, Klinikum Bayreuth, Bayreuth, Germany. [12]These authors contributed equally: Colleen McGregor, Xiao Qin. ✉e-mail: agne.antanaviciute@imm.ox.ac.uk; alison.simmons@imm.ox.ac.uk

RNA sequencing (scRNA-seq) survey of rectal tissue from complete proctectomies undertaken for perianal fistulae demonstrated enrichment of myeloid cells and myeloid stromal cross-talk and identified *CHI3L1* as a top upregulated gene in stromal cells from fistulae expressing destructive and fibrotic gene signatures[10].

Challenges in molecular analysis of intestinal fistula tract cells within their intact morphological context have prevented a holistic understanding of fistula biology. Here we used spatial transcriptomics (ST), pan-lineage scRNA-seq, multiplexed immunofluorescence and quantitative collagen imaging[11] to generate a spatially resolved expression atlas using various types of intestinal fistula and control tissue from 92 individuals. Comparing heterogenous fistulae from diverse anatomical locations, we define how normal intestinal molecular zonation patterns are profoundly disrupted around fistula tracts.

## Spatial mapping of fistulating CD

In this study, we assembled a cohort of 68 fistulae obtained from patients with and without CD (Supplementary Table 1 and Supplementary Data). Histopathological examination by two independent pathologists (Extended Data Fig. 1a–e and Supplementary Data) revealed that the majority of CD fistulae penetrated the lamina propria and muscularis mucosae into the muscularis propria and serosal layers. Of these fistula tracts, 39.2% were non-epithelialized, lined with granulation tissue displaying neovascularization and dense inflammatory infiltrate (Extended Data Fig. 1b–d). By contrast, epithelialized tracts were covered by either squamous (29.4%) or columnar (25.5%) epithelium (Extended Data Fig. 1e), or showed a transition in epithelial types, particularly in enterocutaneous fistulae. Chronic fibrosis was present in nearly half of the cases, marked by hypertrophied muscularis propria, submucosal fibrosis and neural hyperplasia (Extended Data Fig. 1d,e). Inflammation was a common feature (68.3%), with both acute and chronic types, including lymphoid follicles, non-caseating granulomas, lymphoplasmacytic infiltrate and mural abscesses adjacent to the fistula (Extended Data Fig. 1c). Additionally, fissuring ulcers, mucosal erosion with regenerative epithelium, adipose deposition and oedema within the submucosa were commonly observed.

To better understand the molecular mechanisms and cellular pathology that underpin fistulating CD, we conducted comprehensive scRNA-seq and ST profiling of our sample cohort using both unbiased and targeted subcellular-resolution approaches (Fig. 1a,b, Extended Data Fig. 1f–h and Methods). Our scRNA-seq cohort yielded 129,204 high-quality cells from immune, stromal and epithelial compartments (Extended Data Fig. 1f and Supplementary Information). In our ST cohort, we sequenced 93,075 tissue-covered spots using the unbiased Visium ST platform (Extended Data Fig. 1g and Supplementary Information) and imaged 7,268,690 high-quality segmented single cells using the subcellular-resolution Xenium ST platform (Extended Data Fig. 1h and Supplementary Information).

Integrative clustering analysis identified 17 broad cell populations in our scRNA-seq dataset, 17 analogous cell populations in our Xenium dataset, and 11 distinct tissue regions in our unbiased Visium dataset (Extended Data Fig. 1f–n). We annotated these populations using canonical marker genes and previously published single-cell reference datasets[12–15].

To ensure adequate sampling across the highly heterogeneous disease presentations, we spatially profiled samples from internal and external fistulae (enterocutaneous, colocutaneous and perianal), capturing both epithelialized and non-epithelialized fistula tract presentations of the disease (Fig. 1b). Principal component analysis (PCA) of the Visium and Xenium cohorts confirmed strong sample segregation by these fistula subtypes, with epithelialization and location as the primary drivers of variability (Extended Data Fig. 2a–e), underscoring the value of our broad sampling approach.

## Fibroblast programmes in Crohn's fistulae

scRNA-seq analysis revealed nine subsets of fibroblasts within our dataset, which grouped into previously described core fibroblast subsets[15]: mucosal structural cells (Stromal 1, ADAMDEC1+ fibroblasts), telocytes (Stromal 2, F3+PDGFRA+ fibroblasts), submucosal or deep tissue fibroblasts (Stromal 3, C3+ fibroblasts) and follicular reticular cells (Stromal 4, CCL19+ fibroblasts) (Extended Data Fig. 2f). In fistulating CD samples, we identified an additional cluster of fibroblasts, which we termed FAS cells, that expressed markers associated with fibrosis and tissue remodelling, as well as several cytokines and chemokines (Extended Data Fig. 2f–i), together suggesting that FAS cells may have diverse functions in facilitating tissue repair, promoting extracellular matrix (ECM) remodelling and modulating immune functions[16,17].

We next investigated the regulation of FAS cells and identified several upregulated and/or uniquely expressed transcription factor genes and transcription factor-driven gene regulatory networks (Extended Data Fig. 2j,k), including *TWIST1*, *TWIST2*, *PRRX1*, *PRRX2* and *OSR2*. Together, these transcription factors are involved in developmental patterning, tube structure closure and morphogenesis, and mutations in these genes can result in developmental syndromes, including craniofacial abnormalities and cleft palate[18–22].

To validate these FAS gene expression patterns, we assessed *TWIST1* expression using immunohistochemistry and further performed multiplexed quantitative PCR (qPCR) for *TWIST1* in 84 patient samples, which showed that *TWIST1* expression was significantly upregulated in fistulating disease across our cohort (Extended Data Fig. 2l,m).

## Meta-analysis of intestinal fibroblasts

To shed further light on the nature of FAS cell states identified by scRNA-seq, we conducted a meta-analysis of IBD scRNA-seq datasets that reported[10,13–15,23–36] inflammation-associated fibroblast states. We integrated 487 samples from ulcerative colitis and CD, including perianal fistulae, from 11 studies. This analysis uncovered 17 clusters of fibroblasts, which we could broadly group into mucosal, submucosal and follicular fibroblasts (Extended Data Fig. 3a–g). Whereas some clusters reflected location, others were restricted to active IBD (Extended Data Fig. 3d–f,h–l). Comparing fistulating samples with active ulcerative colitis or CD, FAS cells—corresponding to the S3-CHI3L2+ meta-analysis cluster—were strongly enriched in fistulating CD (Extended Data Fig. 3k,l), but shared marker gene expression with other IBD fibroblast states (Extended Data Fig. 3g and Supplementary Information).

Consensus non-negative matrix factorization (cNMF) identified eight IBD-linked gene programmes that were enriched for interferon response, follicle-related functions, ECM deposition and remodelling and wound healing (Extended Data Fig. 3m,n). Most IBD fibroblasts, including FAS cells, could be defined by distinct combinations of these factors. Inflammatory fibroblasts that were most similar to FAS cells (S1/S2-CHI3L1+ clusters) expressed factor 10 (*AREG* and *PHDLA1*) and mucosal or telocyte programmes that were absent in FAS cells, whereas FAS-specific factors (factors 15, 22, 24 and 25) were reduced but not absent in inflammatory fibroblasts (Extended Data Fig. 3n–t). Thus, FAS cells are related to inflammatory fibroblasts but distinguished by loss of mucosal or telocyte depth-specific programmes and acquisition of additional wound-healing and fibrotic signatures.

For broader context, we examined a dataset of wound healing in diabetic ulcers[37], finding fibroblasts enriched in FAS signatures that segregated with healing samples (Supplementary Information). In a previously published colitis ST dataset[36], FAS programme activity was rare (0.8%, 214 out of 25,672 spots) but localized to areas of epithelial damage (Supplementary Information). Together, these findings suggest that FAS cells represent an intestinal wound-healing population with core pro-repair gene networks that is conserved across multiple organ systems.

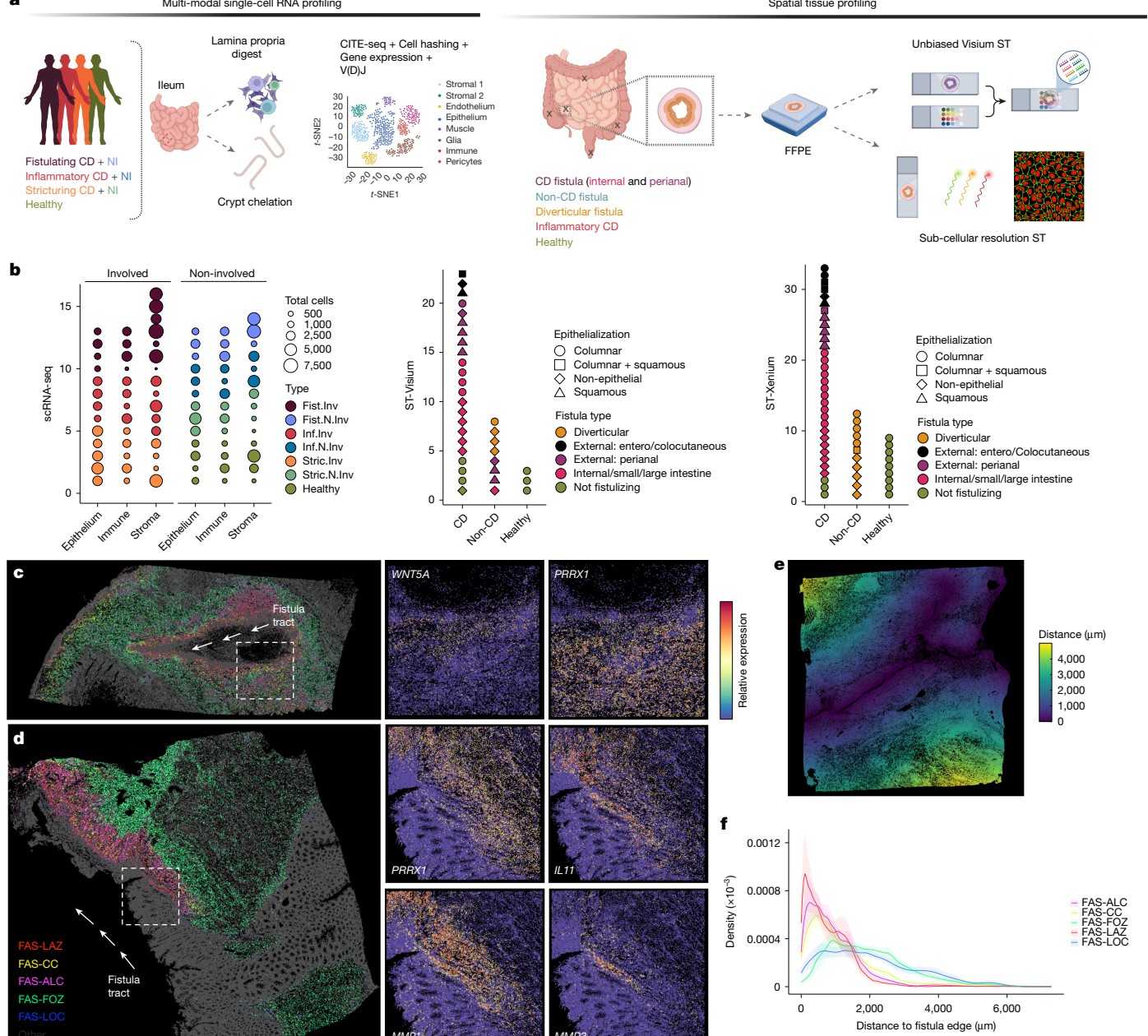

**Fig. 1 | Fistula-associated fibroblast states lining fistula tracts. a**, Schematic overview of the experimental design for spatial and single-cell analyses. FFPE, formalin-fixed paraffin-embedded sections; NI, non-involved tissue; *t*-SNE, *t*-distributed stochastic neighbour embedding. Created in BioRender. Group, S. (2025) https://BioRender.com/2lka2xp. **b**, Overview of the scRNA-seq and ST cohorts. Each point represents an individual sample. Fist, fistulating; Inf, inflammatory; Inv, involved; N.Inv, non-involved; Stric, stricturing. **c**, Left, a representative non-epithelialized CD fistulating tissue section profiled with Xenium ST (*n* = 53 total samples). A paired haematoxylin and eosin (H&E) image is shown in Supplementary Data. Right, closer views of the indicated lesion edge, showing *WNT5A* and *PRRX1* expression. Fistula tract is indicated by white arrows. **d**, Left, a representative partially epithelialized CD fistulating tissue

section profiled with Xenium ST (*n* = 53 total samples). A paired H&E image is shown in Supplementary Data. Right, closer views of the indicated lesion edge, showing *PRRX1*, *IL11*, *MMP1* and *MMP3* expression. Fistula tract is indicated by white arrows. **e**, Distance-based analysis with respect to fistula edge, with a representative (*n* = 13 total sections with a clearly histologically identifiable fistula tract) non-epithelialized fistula tract tissue section visualizing the distance of each cell centroid from the edge of the fistula tract. **f**, Density plot visualizing FAS cell subtype distribution over distance from fistula edge, measured across all fistulae with a clearly histologically identifiable fistula tract (*n* = 13). Shaded areas around lines show s.e.m. per cell type, representing variation across independent patient sections.

## Layered spatial zonation of CD fistulae

To explore the role of FAS cells further, we undertook subcellular spatial analysis of pathologist-guided regions of interest (ROIs) from fistula tract areas from patients with CD and without CD (for histopathology annotations and ROI selections, see Supplementary Data). In our in situ

cohort, our targeted Xenium panel identified around 60 cell-type clusters (Supplementary Information), which encompassed 11 fibroblast subclusters as well as neutrophils and neurons, which are typically underrepresented in scRNA-seq.

To determine how the clusters aligned with different fistula presentations, we performed differential abundance analyses (Extended Data

Fig. 4). This revealed consistent enrichment of several cell populations across all fistula types, with fewer fistula subtype-specific variations. Notably, even control fistulae originating from diverticular disease exhibited similar changes when location and epithelialization status were controlled for, suggesting commonality in intestinal fistula molecular pathology in established tracts, regardless of aetiology. We also detected region-specific variation in line with known intestinal diversity[38,39] (Supplementary Information). Among fibroblasts, five clusters were consistently over-represented across all fistulae: three most closely matched FAS cells, whereas two resembled follicular reticular (Stromal 4) and a subset of submucosal fibroblasts. FAS subclusters were distinguished by expression of distinct transcription factors, cytokines, chemokines, morphogens and ECM factors (Extended Data Fig. 5a–f).

Spatial mapping revealed FAS cells arranged in discrete layers around the tract (Fig. 1c,d and Extended Data Fig. 5g). FAS-LAZ (lesion adjacent zone) cells lined the lumen with proliferating FAS-CC (cell cycle) cells, expressing morphogens (*WNT5A* and *DLL1*) and *IL11*. Beneath them, FAS-ALC (active lesion core) cells formed a thicker lesion core, and fibrotic FAS-FOZ (fibrotic outer zone) cells localized deeper in the stroma. As distance from the lumen increased, immune signalling and remodelling pathways diminished. FAS-LOC (lymphoid organizer) cells, resembling follicular fibroblasts, were scattered throughout lesions and correlated with T cell and B cell infiltration, unlike healthy tissue, where such cells are restricted to follicles. In fistula tracts, however, their dispersion throughout the stroma suggests a loss of this regulated architecture, potentially contributing to immune dysregulation and chronic inflammation.

To quantify spatial variation, we calculated the distance of each cell from the tract edge and analysed gene expression gradients (Fig. 1e,f, Extended Data Fig. 5h–j and Methods). These were consistent across fistulae and confirmed zonation of FAS subsets. Spatial correlation highlighted relationships between FAS programmes and immune and ECM genes: *PRRX1* co-localized with TIMP genes (which mediate MMP inhibition), *WNT5A* and *CXCL13* co-localized with proliferation and lymphocyte markers, and *NRG1* and *F3* co-localized with epithelial cells at tract edges. Neutrophils and macrophages co-localized with *CXCL1* and *CXCL2*, reflecting active recruitment and innate immune response. Together, these data show how FAS cells adjust their phenotype in response to small variations in their local tissue microenvironment[40,41] (Supplementary Information).

Next, we projected our scRNA-seq data onto Xenium tissue sections using a shared nearest-spatial-neighbours approach (Methods). This confirmed the localizations detected previously (Extended Data Fig. 6a,b), with FAS cells mapping directly onto the fistula edge. Projecting scRNA-seq data shed light on zonation of genes that were not included in our in situ target panel. We analysed ECM-related gene expression patterns, as many collagen family genes are highly abundant and challenging to examine in situ owing to optical crowding. Cells mapping to the transitional edge between FAS-ALC and FAS-FOZ showed induction of type 1 and type 3 collagens, increases in *LOX* (cross-links collagen), loss of *TIMP3* (inhibits MMPs) and increases in *PLOD1* (hydrolyses fibres before cross-linking), *POSTN* (promotes fibroblast activation and recruitment), *CNN2* (promotes proliferation and collagen synthesis) and *SERPINE1* (plasminogen activator inhibitor)[42–44] (Extended Data Fig. 6c,d). Cells deeper in FAS-FOZ also expressed high levels of type 1 and type 3 collagens, whereas FAS-ALC cells did not express any type 1 and type 3 collagens and instead specifically expressed *COL7A1*, which is often associated with wound healing[45]. This FAS cell state-linked molecular zonation supports segregation of FAS functions between superficial active remodelling and deeper layer, pro-fibrotic regions.

At the protein level, multiplexed immunofluorescence (Extended Data Fig. 6e–o) confirmed these findings. FAS-LAZ cells expressed high levels of F3, and extracellular F3 was restricted to the thin zone separating granulation tissue from fibroblast-rich areas. COL7 was deposited along tract edges, forming a disordered, undulating fibril morphology that is consistent with relaxed or immature ECM, whereas POSTN deposition was detected in patches outside core lesions. COL7, which is classically required for anchoring fibrils at epithelial–stromal junctions, shows strong expression at non-epithelialized fistula edges in our data, where a basement membrane is absent. This suggests that COL7 may mark attempted but disordered re-epithelialization; however, this interpretation is speculative and requires functional validation beyond the present study.

## Immune–stromal niches in CD fistulae

Our spatial datasets enabled visualization of immune cell behaviours in fistula tracts. Neutrophil- and macrophage-rich zones lined non-epithelialized surfaces, corresponding to superficial granulation tissue, and were dominated by *SPP1*+ macrophages producing chemokines (*CXCL5*, *CCL3*, *CCL4* and *CXCL2*) for feed-forward neutrophil and macrophage recruitment (Extended Data Fig. 7a–g). Within adjacent FAS-LAZ and FAS-ALC rich layers, *MMP9*+ macrophages displayed both remodelling and immunoregulatory functions (*LYZ*, *IDO1*, *C1QA/B*, *MRC1* and *STAT1*), and a further cluster expressed T cell-recruiting chemokines (*CXCL9*, *CXCL10* and *CXCL11*)[40]. The latter co-localized with *CD8*+, *CD4*+ and regulatory T cells, dendritic cells and B cells, occasionally forming follicular-like aggregates with FAS-LOC fibroblasts, suggesting a shift towards a broader, adaptive immune response beyond the immediate lesion surface (Extended Data Fig. 7h,i). More distal FAS-FOZ-rich zones showed reduced immune infiltration but proliferating endothelial cells and pericytes, consistent with angiogenesis.

Spatial intercellular signalling analysis highlighted cytokine–chemokine networks, angiogenesis, ECM remodelling, adhesion–migration and growth pathways (FGF, PDGF and WNT) in fistula lesions (Supplementary Information). These reflected strong macrophage–fibroblast cross-talk, including *LRP1*–*MMP9* or *SERPINE1* in ECM turnover, integrin–*TGFB1* or *SPP1* in fibrosis, *PDGFRB* promoting proliferation and *SPP1*–*CD44* in fibroblast activation[46] (Extended Data Fig. 7j).

Of note, morphogen pathways were also highly active. FAS cells induced *WNT2*, *WNT4* and *WNT5A* alongside Frizzled (FZD) receptors, with *WNT4* and planar cell polarity (PCP) components (*CELSR1* and *DVL1*) being enriched at invasive fistula 'leading edges'[41,47] (Extended Data Fig. 7k). These regions also contained proliferating MKI67+ fibroblasts, linking aberrant PCP and morphogen signalling to tract expansion. Together with fistula-specific cellular co-localization dynamics (Extended Data Fig. 8a–h), these findings implicate dysregulated immune–stromal interactions, abnormal fibroblast–macrophage communication and disrupted developmental signalling in driving fistula invasion and persistence.

## Developmental transcription factors regulate FAS cells

We next explored how transcription factors expressed in FAS cells might direct fibroblast functions. We over-expressed *TWIST1* and *OSR2* in primary intestinal fibroblasts using lentivectors (Fig. 2a). Successful transduction was confirmed by qPCR and bulk RNA sequencing (RNA-seq) (Fig. 2a and Extended Data Fig. 9a). RNA-seq revealed *TWIST1*-induced ECM organization pathways, MMP activity, fibrillar collagens and morphogenesis-related signalling, including WNT (Fig. 2b,c and Extended Data Fig. 9b). *TWIST1* also induced several other FAS-expressed or fistula-linked transcription factor genes (*PRRX2*, *VDR*, *SNAI1*, *SNAI2* and *GLI1*). *GLI1* and *SNAI1* are indicative of sonic hedgehog signalling, which has been linked to fibroblast activation, production of ECM and development of renal interstitial fibrosis[48]. *TWIST1* overexpression also resulted in strong downregulation of integrins and other adhesion molecules (Fig. 2b,c and Extended Data Fig. 9b).

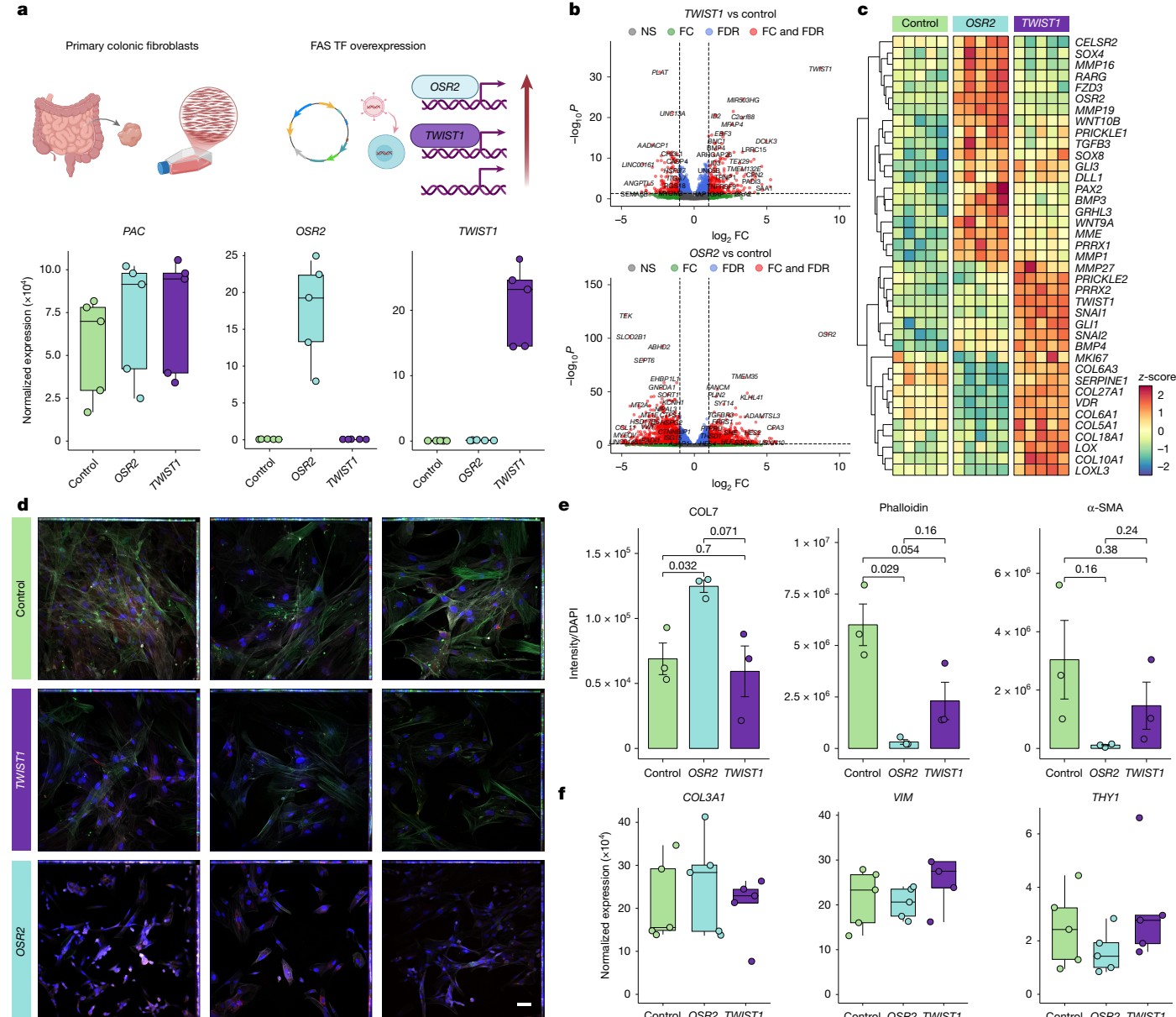

**Fig. 2 | Developmental transcription factor control of fistula-associated fibroblast functions. a**, Schematic representation of experimental design of the *OSR2* and *TWIST1* lentiviral overexpression system (top) and expression of puromycin *N*-acetyltransferase (*PAC*, puromycin resistance gene), *TWIST1* and *OSR2* measured by RNA-seq (bottom). *n* = 5 independent biological replicates per group. TF, transcription factor. Created in BioRender. Group, S. (2025) https://BioRender.com/ushj5nk. **b**, Top, volcano plot visualizing genes that are differentially expressed in fibroblasts that are overexpressing *TWIST1* compared with vector controls. Top differentially expressed genes are labelled (*n* = 5 biological replicates per group). Bottom, comparison of differentially expressed genes in fibroblasts overexpressing *OSR2* and vector controls. Differential expression was assessed using DESeq2 with Wald tests and Benjamini–Hochberg correction for multiple testing. FC, fold change; FDR, false discovery rate; NS, not significant. Red, log$_2$FC > 2 or < −2, −log$_{10}$*P* > 1; blue, −2 < log$_2$FC < 2, −log$_{10}$*P* > 1; green, log$_2$FC > 2 or < −2, −log$_{10}$*P* < 1; black, −2 < log$_2$FC < 2, −log$_{10}$*P* < 1. **c**, Heat map visualizing selected differentially expressed genes in fibroblasts overexpressing *TWIST1* or *OSR2* compared with controls. *n* = 5 biological replicates per group. **d**, Representative immunofluorescence imaging of primary intestinal fibroblasts from *n* = 3 independent donors. Control cells, *TWIST1*-overexpressing and *OSR2*-overexpressing cells are shown. Blue, DAPI; red, COL7, green: α-SMA; white, phalloidin. Scale bar, 50 μm. **e**, Quantification of COL7, phalloidin and α-SMA intensity from images represented in **d**, normalized to overall DAPI signal. *n* = 3 independent replicates per group. *P* values were calculated using two-sided Student's *t*-test. **f**, Gene expression of the fibroblast marker genes *COL3A1*, *VIM* and *THY1*, quantified by RNA-seq in intestinal fibroblasts overexpressing *TWIST1* or *OSR2* and in control cells. All box plots show the median (centre line) and interquartile range (box), and whiskers extend to the most extreme values within 1.5× the interquartile range; individual data points are plotted.

By contrast, *OSR2* induced programmes linked to developmental patterning and organogenesis (*SIX2, ALX4, HOXA3, HOXA5, DLX2, NKX3-2, GLI3* and *FOXH1*), planar cell polarity-linked genes (*CELSR2, FZD3, DLL1, PRICKLE1* and *GRHL3*), as well as WNT and TGFβ signalling (*FZD3, FZD5, RARG, BMP4* and *TGFB3*) (Fig. 2b,c and Extended Data Fig. 9b). *OSR2* also upregulated *TWIST1* (Extended Data Fig. 9c); accordingly, 41% of

genes upregulated and 42% of genes downregulated by *TWIST1* were similarly significantly upregulated and downregulated, respectively, by *OSR2* (Extended Data Fig. 9d). Overexpression of *OSR2*, but not *TWIST1*, strongly downregulated cytoskeletal and cell polarity regulators, including genes encoding filopodia and lamellipodia proteins (*FMNL2, FMNL3* and *DIAPH1*) and Rho GTPase pathway genes.

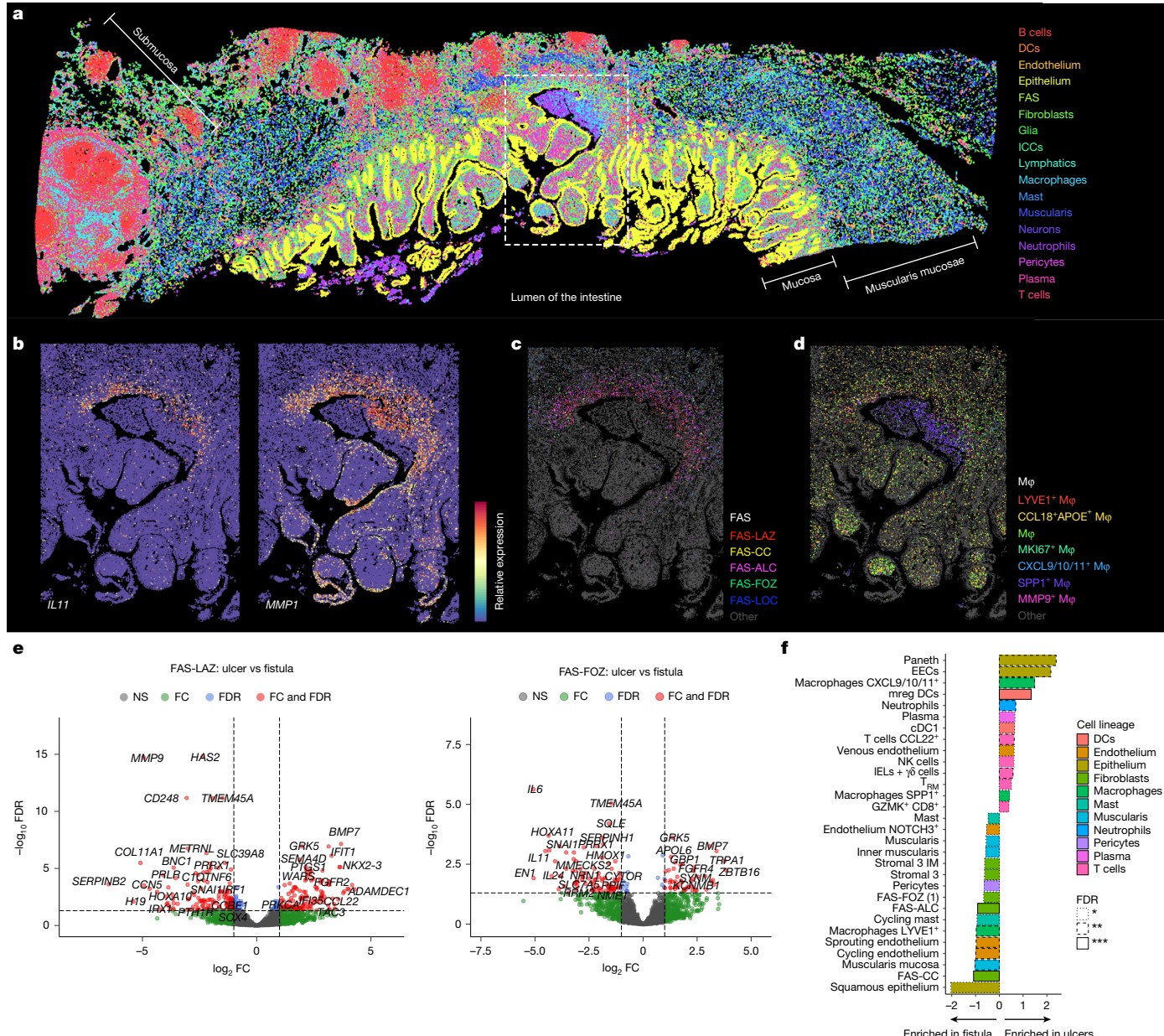

**Fig. 3 | Fistula-associated fibroblasts line ulcer bases. a**, A Xenium ST tissue section (*n* = 53 total samples) of an early fistula or deep fissuring ulcer penetrating past the muscularis mucosae layer into the ileal submucosa. Each point represents a cell centroid, coloured by broad cell lineage as indicated. A white dashed line indicates a ROI around a lesion. DCs, dendritic cells; ICCs, interstitial cells of Cajal. **b**, Gene expression of selected FAS gene markers visualized in the ROI indicated in **a**. **c**, Spatial distribution of FAS cell types around the lesion edge in the ROI indicated in **a**. **d**, Spatial distribution of macrophage cell subtypes around the lesion edge of the ROI indicated in **a**. Mφ, macrophage. **e**, Volcano plot comparing differentially expressed genes between FAS-LAZ cells (left) and FAS-FOZ cells (right) identified in CD ulcers versus fistulae, profiled using 5100-plex Xenium ST cohort. Differential expression was assessed using

DESeq2 with Wald tests and Benjamini–Hochberg correction for multiple testing on pseudobulk gene expression per sample (*n* = 5 CD fistulae, *n* = 7 CD ulcers, *n* = 7 healthy controls). Colours as in Fig. 2b. **f**, Differential cell-type abundance between CD ulcers and fistulae in tissue niches near ulcers or fistula edge or core lesion. *n* = 5 CD fistulae, *n* = 7 CD ulcers, *n* = 7 healthy controls. Bands represent the 95% Bayesian credible interval of the slope (logit fold change in cluster proportion per unit change in the covariate), indicating the range of effect sizes compatible with the data, given the model. cDC1, conventional type 1 dendritic cells; EEC, enteroendocrine cells; IELs, intra-epithelial lymphocytes; IM, inner muscularis; mreg DC, mature dendritic cells enriched in immunostimulatory molecules; NK cells, natural killer cells; $T_{RM}$, tissue-resident memory T cells. **P* < 0.05, ***P* < 0.01, ****P* < 0.001.

Morphological analysis showed that OSR2+ fibroblasts lost their spindle-like shape, and exhibited reduced phalloidin staining and α-SMA expression (Fig. 2d,e), in line with scRNA-seq data showing that FAS downregulated *ACTA2* (Extended Data Fig. 9e). Despite changes in cellular morphology, *OSR2* cells did not show a significant reduction in expression of fibroblast markers (*COL1A1*, *COL3A1*, *THY1* and *VIM*) (Fig. 2f), suggesting that these cells retained their fibroblast identity.

## Fistula-linked fibroblasts in CD ulcers

It has been hypothesized that CD fistulae may originate from ulcer bases[49], but the exact site of origin has not been fully clarified. In our cohort, we captured non-fistulating, inflammatory CD ulcers as well as an early stage of fistula formation characterized by a deep fissuring ulcer penetrating past the muscularis mucosae (Fig. 3a). At this early stage, we observed the presence of FAS fibroblasts and

fistula-associated myeloid cells in ulcer bases (Fig. 3b–d), but also observed key differences in both gene expression and cellular distribution (Supplementary Information).

To better understand differences between fibroblasts found at ulcer bases and in fistulae, we specifically macro-dissected a further cohort of non-fistulating CD ulcers ($n = 7$) together with fistula ($n = 5$) and healthy control samples ($n = 7$) and imaged them using a 5,100-plex ST panel. Clustering analysis and annotation identified 15 broad cell populations and 71 fine-grained cellular phenotypes from 1.8 million total segmented cells (Extended Data Fig. 10a–e). Critically, we were able to identify the same FAS cell populations, corresponding to the same spatial zonations as in previous analyses (Extended Data Fig. 10f–l). cNMF factor analysis identified overlapping and distinct gene expression programmes in each subset (Extended Data Fig. 10m,n and Supplementary Information), which we could map onto the scRNA-seq meta-analysis results.

We found FAS-like cells at the base of every ulcer sample that we imaged, suggesting that they are not unique to ulcers that are predisposed to forming fistulae, but are a general feature of most ulcers in CD. Thus, we next explored the properties that these cells acquire in fistulae that may confer pathogenicity. Comparing gene expression differences between FAS subpopulations located at ulcer bases or fistulae (Fig. 3e and Extended Data Fig. 10o), we identified 643 differentially expressed genes. Ulcer FAS-like cells retained mucosal gene expression (*ADAMDEC1*) profiles when compared with FAS cells, together with functions linked to epithelial proliferation and interferon responses. Conversely, in fistulae, these cells upregulated ECM and patterning-related gene expression programmes, including the transcription factor genes *SNAI1*, *SNAI2*, *PRRX1*, *SOX4* and *HOXA10*.

Comparing the composition of tissue niches between fistulae and ulcers, we found that FAS cells were consistently enriched in fistula lesions compared with ulcer lesions (Fig. 3f and Extended Data Fig. 10p–r). Thus, FAS cells differ in their expression profiles and abundance from FAS-like cells at ulcer bases. Their micro-environments were also different—FAS-like cell niches at ulcer bases were enriched in T cells, plasma cells, dendritic cells and mature dendritic cells enriched in immunostimulatory molecules. Conversely, in fistula, FAS niches were enriched in SPP1+ and LYVE1+ macrophages as well as sprouting and proliferating endothelial cells (Fig. 3f and Extended Data Fig. 11a,b).

Thus, fistula fibroblasts upregulate ECM-remodelling enzymes (for example, *MMP9*, *FAP* and *ADAMTS14*) and developmental transcription factors (for example, *PRRX1*, *RUNX2* and *HOXB7*), enabling tissue invasion and structural reprogramming. These cells activate fibrotic and morphogenic pathways that are distinct from the immune-interacting, interferon-responsive fibroblasts in ulcer bases, reflecting a shift from repair to invasion. This transition is supported by reactivation of TGFβ signalling, collagen processing and the generation of a permissive, pro-migratory ECM niche.

## FAS cells promote re-epithelialization

Epithelialization has a dual role in CD fistulae. It is crucial in wound repair, but epithelialization of deep, penetrating fistula tracts prevents resolution, necessitating surgical curettage, which removes not only debris and infected tissue but also the epithelial lining of the fistula tracts to promote healing[50,51].

In our ST cohort, we profiled non-epithelialized ($n = 12$), partially epithelialized ($n = 7$) and fully epithelialized fistulae ($n = 19$), with the partially epithelialized tracts enabling assessment of the active process of epithelialization at the leading edge (Fig. 4a and Extended Data Fig. 11c,d). Fistula tracts were lined by either columnar or squamous epithelium, with squamous epithelium being restricted to external or perianal fistulae; in some cases, both types were present where cutaneous and intestinal epithelium joined.

Canonical intestinal subtypes were largely preserved, although irregularities such as Paneth cell metaplasia were observed. Across

epithelialized regions we also identified five epithelial clusters enriched in fistulae compared to controls (Extended Data Fig. 4). Clusters with increased inflammatory and regenerative marker expression (NOS2+ and REG1A+) were most pronounced at transition zones (Extended Data Fig. 11e). Distance-based analyses with respect to the leading edge of epithelialization in these tracts highlighted that these epithelial cell states tapered out as homeostasis was rapidly established in trailing crypts (Extended Data Fig. 11f,g).

A distinct FAS-rich niche was present beneath these transition zones. FAS cells adjacent to epithelium upregulated *WNT5A*, *WNT5B*, *NRG1* and *F3*, resembling telocyte-like fibroblasts supporting epithelial crypt niches (Fig. 4a,b). *NRG1* showed the strongest induction precisely at the last detectable crypt in these lesions, with evidence of signalling to epithelial ERBB receptors (Extended Data Fig. 11h–j). At the protein level, *F3* expression was detected at the fistula tract surface, both in the presence and in the absence of epithelial coverage (Extended Data Fig. 6h,i and Supplementary Information). Epithelial cell gene expression was similarly found to co-vary with the phenotypes of adjacent fibroblast and FAS cells (Extended Data Fig. 11k). This suggests a role for FAS cells in supporting focal epithelial regeneration of the fistula tract[52].

*WNT5A* was also strongly induced in these regions, yet some fistula epithelia lacked *FZD5*, suggesting impaired responsiveness to regenerative cues. qPCR confirmed broader WNT pathway disruption, with *FZD7* being upregulated and *FZD6* being downregulated (Extended Data Fig. 11l). Additional defects included loss of normal BMP gradients, reduced differentiation marker expression, expansion of proliferative *MKI67*+ zones, and decreased expression of the transcription factor gene *ASCL2* in stem cells (Extended Data Fig. 11e,m,n), together indicating a dysregulated crypt niche with impaired maturation and increased proliferation.

Notably, FAS cells became very sparse where fistula tracts were fully epithelialized, extending only 5–10 crypts past the epithelialized–non-epithelialized tract transition zone, suggesting that once re-epithelialization is established, these niches dissemble, favouring emergence of a stabilized fistula tract stromal architecture. FAS cells underlying non-epithelialized tract regions also strongly induced *NRG1* and *WNT5A*, indicative of a drive to rescue epithelialization in these regions. This suggests a model in which FAS cells attempt to promote epithelial regeneration even in the absence of epithelial stem cells within their proximity.

## ECM remodelling in Crohn's fistulae

Finally, we investigated whether FAS subtypes contribute to fistula persistence via ECM alterations. Picrosirius red staining of ST-adjacent sections enabled integration of transcriptomics data with ECM architecture profiling. Collagen structure was visualized under standard and polarized light to better discriminate between dense fibrotic structures[11].

Image analysis (Extended Data Fig. 12a and Methods) identified 15 cross-sample regions with varying collagen content (Fig. 5a–c and Extended Data Fig. 12b,c). Lower-density zones mapped to adipose or mucosal regions, whereas highly dense regions were enriched in fistulae (Fig. 5c). A distinctive hypo-dense cluster 1 localized to non-epithelialized fistula lumen edges, whereas dense fibrotic regions lay deeper, forming tract boundaries.

Geometry metrics (see Methods) revealed that fistula-specific dense clusters contained thick, parallel bundles with high anisotropy and fractal dimension, consistent with fibrosis (Fig. 5b and Extended Data Fig. 12b,c). By contrast, lumen-adjacent ECM beneath non-epithelialized surfaces showed sparse, discontinuous collagen with low anisotropy and high lacunarity. ECM in healthy submucosa contained short, crimped bundles imparting elasticity and mechanical flexibility, but in fistulae this was replaced by rigid, high-density collagen. Analysis of transitional regions in partially epithelialized fistula tracts showed that submucosa beneath epithelialized tracts was highly fibrotic,

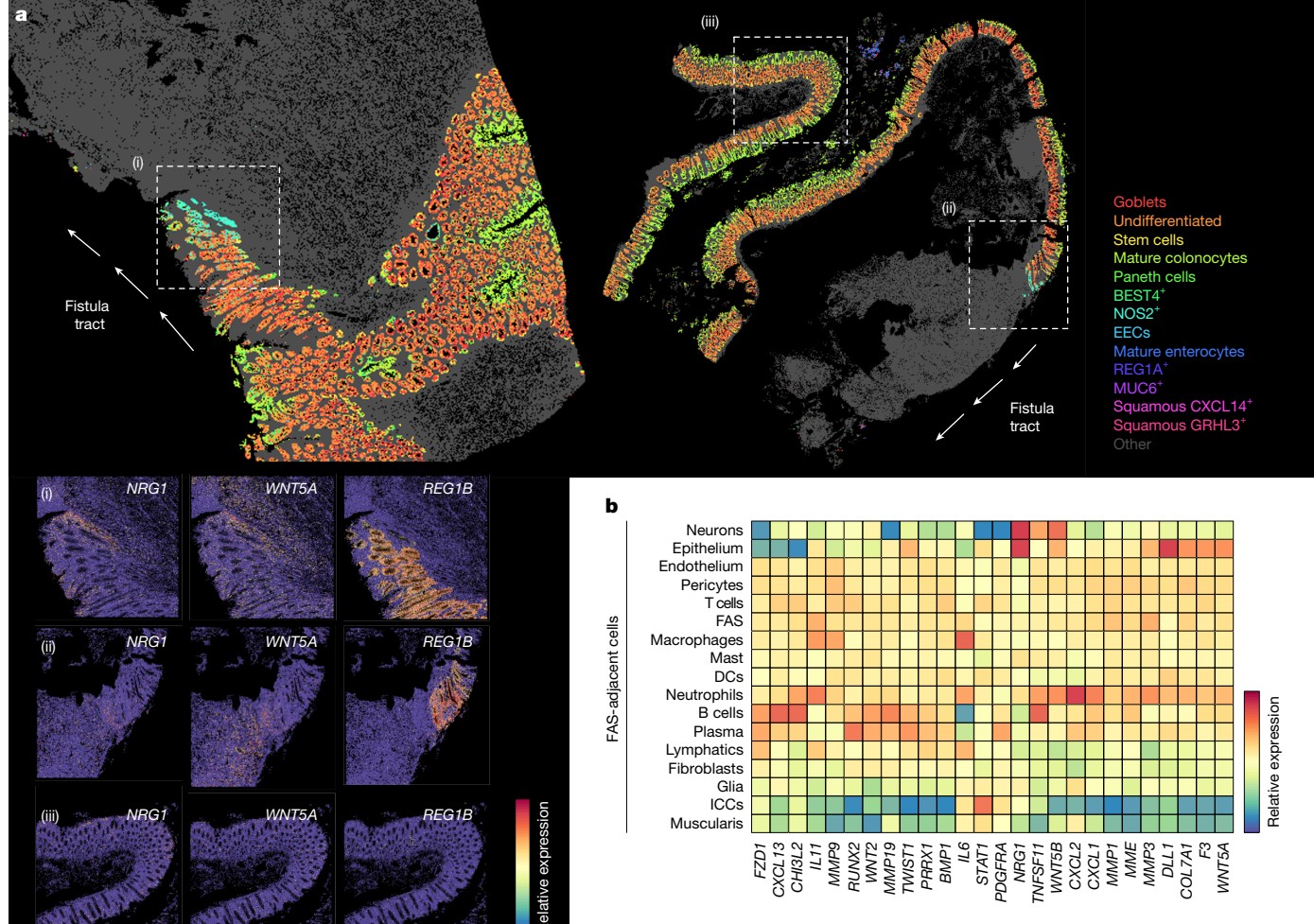

**Fig. 4 | Fibroblast states that support fistula tract epithelialization.**
**a**, Two representative tissue sections (*n* = 53 Xenium ST samples) of partially epithelialized CD fistula tracts profiled with Xenium ST. Points represent cell centroids and epithelial cell subtypes are highlighted. All other cell types are shown in grey. Selected ROIs showing the transition zone between epithelialized and non-epithelialized parts of the fistula tract are indicated by dashed lines in regions i and ii. Dashed lines around region iii show a representative normal-appearing mucosa distal to the fistula lesion. Bottom left, gene expression of selected epithelial and stromal genes across regions i, ii and iii. Fistula tracts are indicated by white arrows. **b**, Heat map visualizing gene expression variation in FAS cells conditional on which other cell types are in the local spatial neighbourhood of FAS cells. Gene expression is aggregated and scaled (*n* = 53 Xenium ST samples). For instance, FAS cells near B cells express higher levels of *CXCL13*, whereas FAS cells near neutrophils express higher levels of *CXCL1* and *CXCL2*.

abruptly shifting to areas of lower density and active remodelling in non-epithelialized regions at the fistula edge. This suggests a progression whereby re-epithelialization resolves hypo-dense collagen at the lumen, whereas deeper fibrotic ECM persists.

We next correlated these patterns with our ST data (Fig. 5d,e and Extended Data Fig. 12d–k). FAS-LAZ and FAS-ALC cells localized to hypodense regions, precisely corresponding to the cluster 1 region, whereas FAS-FOZ cells were enriched near dense fibrosis. Notably, the expression of key ECM-remodelling factors such as MMP3 corresponded precisely with the boundaries between high- and low-fibrosis regions, indicating highly targeted ECM editing by these cells at the fistula edge. Thus, distinct FAS subtypes differentially modulate fistula ECM structure: FAS-LAZ and FAS-ALC cells associate with remodelling and low density, whereas FOZ cells promote fibrosis that reinforces persistence of tract stabilization.

## Discussion

By combining scRNA-seq and high-resolution spatial analysis, we identified the contributions of distinct fibroblast, immune and epithelial cells to the initiation, maintenance and progression of CD fistulae (Extended Data Fig. 12l). FAS cells emerged as central to fistula niches, forming layered structures around tracts. Their zonated expression of ECM-remodelling, fibrotic and immune genes suggests specialized roles in lesion evolution. Transcription factors such as *TWIST1*, *OSR2*, *PRRX1* and *RUNX2* support the involvement of FAS cells through developmental pathways that drive aberrant tissue morphogenesis. In vitro, *TWIST1* and *OSR2* promoted divergent fibroblast programmes, consistent with our spatial observations: *TWIST1* induced pro-fibrotic behaviour, whereas *OSR2* activated PCP- and matrix-degrading pathways linked to epithelialization and remodelling. These findings point to dysregulated, atypical developmental cues shaping fistula biology, which will be important areas for future work.

FAS cells also form inflammatory niches through interactions with *SPP1*+ macrophages and other immune cells. *SPP1*+ macrophages, which are enriched at the lumen surface, facilitate ECM turnover and recruit additional immune cells, supporting a persistent inflammatory environment. ECM analysis revealed that FAS cells are likely to have a critical role in driving fibrosis, contributing to tissue rigidity and stabilization of the fistula tract.

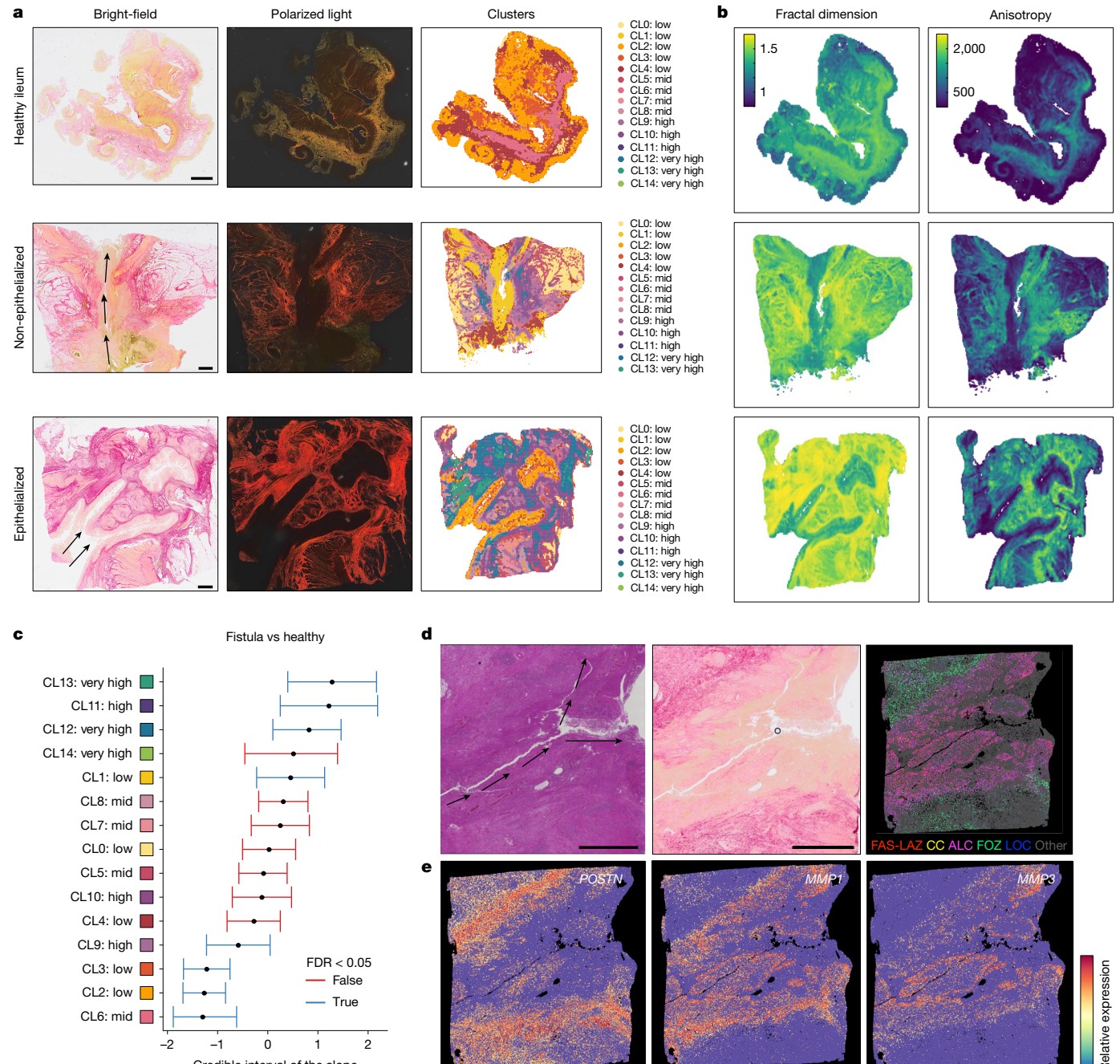

**Fig. 5 | Disparate functions of fistula-associated fibroblasts in tract evolution and stabilization. a**, Representative tissue sections ($n = 27$ samples) stained with picrosirius red from healthy ileum, non-epithelialized CD fistula and epithelialized CD fistula, visualized under normal (left) or polarized (middle) light and the corresponding spatial distribution of detected image clusters (right). Arrows indicate fistula tracts. Scale bars, 2 mm. **b**, As in **a**, with visualization of fractal dimension and anisotropy measures. **c**, Differential abundance plot comparing picrosirius red image cluster abundance in CD fistula and healthy control sample tissue sections ($n = 27$ samples imaged).

Bands represent the 95% Bayesian credible interval of the slope (logit fold change in cluster proportion per unit change in the covariate), indicating the range of effect sizes compatible with the data, given the model. **d**, Representative ($n = 27$ samples) non-epithelialized fistula tissue section with fistula tract as seen by H&E staining (left) and paired picrosirius red imaging (middle). Right, paired Xenium ST data for FAS cell types, with FAS cell types highlighted and all other cells shown in grey. Arrows indicate fistula tract. Circle is an imaging artefact (small bubble). Scale bars, 2 mm. **e**, Gene expression of FAS subtype markers visualized in the representative fistula section shown in **d**.

How FAS cells acquire their pathogenic features remains unclear. They may gain invasive and fibrotic properties through local signalling and inflammation-induced epigenetic reprogramming, priming them for aggressive behaviour during recurrent injury in CD. Our observation that fistula fibroblasts closely resemble those at ulcer bases suggests a shared origin or induction cues; however, the many differences that we observed point to subsequent reprogramming that drives their distinct pathogenic features, which warrants future study.

Our collagen staining also revealed that the muscularis mucosae structure was frequently altered and often entirely absent in epithelialized fistulae and inflammatory CD. The loss of defined ECM in the muscularis mucosae may be an early step that enables FAS cells to penetrate into the submucosa, initiating the fistulating process. FAS cells themselves are likely to be involved in this ECM remodelling. In several tissue sections, we profiled ROIs with the fistula tract penetrating through the muscularis propria, where FAS cells congregated at the muscle

edge and produced localized MMPs (Extended Data Fig. 12m,n). This suggests that these cells are capable of making a path through diverse gut structures, facilitating the invasive progression of the fistula.

Overall, our study identifies distinct fibroblast subtypes and their interactions with immune and epithelial cells as central to fistula pathology. FAS-ALC and FAS-LAZ cells mediate wound healing and remodelling, indicating that therapies will need to preserve their regenerative functions while limiting invasive and fibrotic activity. FAS-FOZ cells reinforce tracts by depositing fibrotic matrix, suggesting that selectively inhibiting their emergence could aid fistula resolution.

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

# Methods

## Human tissue samples

Written informed consent was obtained from adult patients undergoing elective or emergency IBD surgery in accordance with NHS Research Ethics Committee (REC) approvals (TIP-18/WM/0237; GI Biobank-16/YH/0247; IBD Biobank-09/H1204/30). Healthy control samples were sourced from patients undergoing elective colorectal cancer surgery (uninvolved tissue) or stoma reversal. Patient metadata are summarized in Supplementary Table 1.

Archived diagnostic formalin-fixed paraffin-embedded (FFPE) blocks for ST and validation were obtained from Oxford University Hospitals NHS Foundation Trust via the Oxford Radcliffe Biobank (ORB; REC:19/SC/0173) and the Oxford Centre for Histopathology Research. Additional FFPE blocks were obtained from the Friedrich-Alexander University Erlangen-Nuremberg (ethics: 23–131 bp).

## Sample collection, handling and processing

Full-thickness resection tissue samples, surplus to clinical needs, were obtained by the operating surgeon, who identified involved and uninvolved tissue macroscopically. Pathology was confirmed through clinical histopathology reports. Samples were stored in Dulbecco's modified Eagle's medium (DMEM, Gibco) supplemented with 10% fetal calf serum (FCS; Sigma-Aldrich), 100 U ml$^{-1}$ penicillin, and 100 U ml$^{-1}$ streptomycin (Sigma-Aldrich) and transported to the laboratory on ice.

Tissue samples were rinsed with phosphate-buffered saline (PBS, Gibco) and examined for anatomical landmarks. Blunt dissection was performed on ice to isolate full-thickness (FT), submucosa (SM) and mucosa (MUC) layers. Following dissection, samples were immediately preserved in CryoStor CS10 (Sigma-Aldrich) and stored in liquid nitrogen for subsequent digestion following the manufacturer's protocol.

For FFPE sections, tissues were oriented in cassettes, fixed in 10% buffered formaldehyde (VWR International) for 48 h, and transferred to 70% ethanol before being processed through a graded alcohol series for paraffin embedding. FFPE blocks were stored at −20 °C for long-term use until required for histological staining, immunohistochemistry or ST experiments.

## Cell isolation, staining and sorting

Full thickness tissue samples were dissociated into single-cell suspensions for scRNA-seq using optimized protocols for epithelial and lamina propria (LP) fractions.

**Epithelial cell enrichment.** Epithelial cell dissociation was performed by adapting previously described protocols[12,53]. In brief, cryopreserved FT or MUC tissue samples were thawed for a minute in a 37 °C water bath then washed in 15 ml of supplemented DMEM before centrifugation. Tissue samples were cut into small fragments followed by further centrifugation. Samples were incubated in pre-warmed chelation medium (HBSS (Lonza) supplemented with penicillin and streptomycin, HEPES, FCS, 5 mM EDTA (Invitrogen) and 2 mM dithiothreitol (Thermo Fisher)) for 20 min, vortexed periodically, and then centrifuged to isolate crypt-containing supernatants. The chelation protocol was completed a total of two times to increase yield of isolated crypts. Recovered crypts were digested with TrypLE Express (Gibco) and DNase (Sigma) for 30 min to achieve a single-cell suspension, filtered through 70-µm and 40-µm filters to remove debris, and counted for viability. Between 0.5 and 1 million highly viable cells per sample were taken forward for cell staining. Cells were incubated with TruStain FcX (BioLegend) for 10 min at 4 °C, to block non-specific binding. Cells were then stained with a cellular indexing of transcriptomes and epitopes by sequencing (CITE-seq) antibody mix (CD4, CD8, CD45RO, CD45RA, PD1, CD103, CD56 and CD3, each at 1:100) (BioLegend), to identify and exclude any contaminating immune subsets or intra-epithelial lymphocytes and also stained with hashtag-oligo (HTO) antibodies

at 0.75 µl per 1 × 10$^6$ cells per 100 µl staining volume (BioLegend), for 30 min at 4 °C. After staining, cells were immediately spun down, and resuspended at the desired pooling concentrations prior to loading on the 10X Chromium platform.

**Immune (CD45$^+$) and stromal (CD45$^-$EPCAM$^-$) cell enrichment.** LP fractions were isolated using an optimized Miltenyi Biotech LP Dissociation Kit protocol tailored for FT ileal tissue. Cryopreserved samples were rapidly thawed and dissected into small fragments and resuspended in buffer L (DMEM (high glucose), FCS and HEPES). Tissue fragments underwent digestion at 37 °C for 2 h, with periodic mechanical dissociation using a blunt needle and syringe. Following digestion, single-cell suspensions were filtered through a 100-µm filter, washed with DMEM followed by centrifugation and counted to ensure high viability prior to further processing.

For staining and sorting, single-cell suspensions were incubated with antibody panels targeting markers to identify immune (CD45$^+$ (1:100)) and stromal (CD45$^-$ EPCAM$^-$ (1:50)) populations following TruStain FcX (BioLegend) blockage. TruStain FcX was added at 5 µl per 1 × 10$^6$ cells in 100 µl staining volume. Additional CITE-seq and hashtag antibodies were used, as above, to enable multiplexing for scRNA-seq. DAPI stain (1:1,000) just prior to FACS sorting for live-dead differentiation. Flow cytometry (BD LSR II) and fluorescence-activated cell sorting (BD FACSAria III and BD FACSAria Fusion (BD FACS Diva Software), BD Biosciences) were employed to enrich cell populations. The accuracy of sorting gates was confirmed using compensation controls generated using single colour controls and fluorescence minus one samples. Live cells sorted into stromal (CD45$^-$EPCAM$^-$) and immune (CD45$^+$) subsets were collected in Eppendorfs containing FCS and immediately spun down, counted, resuspended and pooled prior to loading onto the 10X Chromium platform.

**LeviCell-based stromal cell enrichment.** The LeviCell platform (LevitasBio) offered an alternative method for stromal enrichment. In brief, FT tissue samples underwent both epithelial crypt chelation and LP digestion. LP-derived single-cell suspensions were then stained with respective hashtag-oligo antibodies, counted and pooled in desired one-to-one ratio. Levitation buffer and CD45 depletion beads were added to the stained LP-derived single-cell suspensions per the manufacturer's instructions (1004001). The suspension was loaded into the LeviCell-1.0 system, which employs paramagnetic levitation to separate cells based on viability. Live stromal cells (crypt-depleted (CD45$^-$)) were enriched from the top output well, and dead and CD45$^+$ cells were removed via the bottom output well. The isolated stromal fraction was directly processed for scRNA-seq.

## Droplet-based scRNA-seq

Droplet-based scRNA-seq was undertaken using the 10X Chromium Single Cell platform in accordance with the manufacturer's instructions (10X Genomics, 5′ v1.1 chemistry, CG000208, Rev F).

Approximately 30,000–40,000 cells were loaded per pool. The workflow involved encapsulating individual cells within gel beads in emulsion (GEMs), barcoding, reverse transcription of complementary DNA (cDNA) within GEMs, cDNA clean-up, amplification, and library construction for gene expression (GEX) and antibody-derived tags (ADT) data.

To enhance sequencing quality and depth, the workflow was modified by incorporating Jumpcode CRISPRclean Single Cell RNA Boost Kit (KIT1018 v1.0, Jumpcode Genomics). Post-ligation clean-up of GEX libraries included CRISPR RNA depletion using the Cas9–single guide RNA ribonucleoprotein complex. This step targeted unaligned reads, highly expressed ribosomal and mitochondrial genes, and non-variable genes for depletion. Following this, library construction proceeded in line with the manufacturer's protocol.

Library quality and integrity were assessed using Agilent Bioanalyzer TapeStation to ensure compliance with sequencing requirements. Final

libraries were pooled at a 4 nM concentration and sequenced on an Illumina NextSeq 500/550 (High Output v2.5, 150 cycles) or outsourced to a NovaSeq PE150. Sequencing depth and run parameters followed the specifications provided by the manufacturer (CG000208, Rev F).

## Spatial transcriptomics

Five to ten 5-μm scrolls were sectioned from each FFPE tissue block and stored at −80 °C for RNA quality assessment. Total RNA was extracted using the Qiagen RNeasy FFPE Kit (73504) following the manufacturer's protocol. RNA integrity was assessed using the RNA Pico Assay (Agilent), which quantified DV200 values. Samples with DV200 values above 50% were considered suitable for ST.

All FFPE blocks were stained with H&E to evaluate tissue morphology and pathology representation. Only samples meeting both RNA quality and tissue morphology criteria were selected for spatial gene expression analysis.

**Visium FFPE protocol.** The sample preparation workflow for 10X Genomics Visium Spatial Gene Expression (CG000408 and CG000409 rev. D) was followed, with specific optimizations. FFPE blocks were stored at −20 °C, slide drying time was extended to 48 h at room temperature, and deparaffinization and alcoholic rehydration steps were modified to minimize tissue detachment.

For the ST experiment, 5-μm sections were placed onto Visium FFPE spatial GEX slides, which feature 4 capture areas (42.25 mm² each) containing 5,000 gene expression spots per area. Tissue sections were deparaffinized, stained with H&E (imaged using a Zeiss AxioScan at 10× and 20× magnifications), and de-crosslinked. Tissue digestion, gene-specific probe hybridization and cDNA synthesis were performed directly on the slide, incorporating spatial barcoding, per the Visium protocol (CG000407, rev. D). cDNA quantification via qPCR (KAPA SYBR FAST) informed the number of DNA amplification cycles. Libraries were then constructed following the manufacturer's instructions. Library quality was verified using the Bioanalyzer High Sensitivity DNA Kit (Agilent) and concentrations were measured using the Qubit dsDNA HS Assay Kit (ThermoFisher). Libraries were pooled at 4 nM, with sample proportions adjusted based on the ratio of tissue section area to capture area.

Sequencing was performed on an Illumina NextSeq 500 platform (High Output v2.5, 150 cycles) with the following run parameters: read 1 (28 cycles), i7 index (10 cycles), i5 index (10 cycles), and read 2S (50 cycles).

**Xenium FFPE in situ protocol.** Slides for Xenium in situ were prepared following the manufacturer's instructions with modifications. In brief, FFPE tissue blocks were sectioned at 5-μm thickness and placed on Xenium slides with a capture area of 235 mm² (CG000578, rev. C). After drying, sections were deparaffinized and de-crosslinked (CG000580, rev. C) to facilitate probe hybridization. Custom-designed Xenium probes were hybridized to the tissue overnight, followed by probe ligation and annealing of rolling circle amplification primers. Sequential washing steps were performed to minimize autofluorescence and nuclei were stained with DAPI for imaging (CG000582, rev. E). Imaging was conducted using the automated Xenium Analyzer (CG000584, rev. E) which identified transcript counts through unique optical signatures for each gene. Transcripts with quality scores (Q-score) exceeding 20 were retained for downstream analysis. Cell boundaries were initially defined using DAPI images (see 'Computational analysis' for cell segmentation details). Xenium Prime analysis was carried out as above, with the addition of cell boundary staining per the user guide (CG000749, rev. B).

After the Xenium run, slides underwent H&E staining (CG000613, rev. A) to enable direct alignment of gene expression data with histological information from the same tissue section. H&E slides were imaged at 10× and 20× magnifications using the Zeiss AxioScan Z1.

## Histopathological characterization and validation

**Histopathological analysis and annotation.** For ST analysis, corresponding digitized H&E images were reviewed. Two gastrointestinal pathologists (M.V. and E.F.) independently examined the slides with limited prior knowledge of patient clinical status. Manual annotations captured cellular details, pathology and tissue architecture. Both pathologists reviewed the same slides, and minimal inter-observer variability was observed, ensuring robust assessments. A comprehensive review of the formal histopathology report, including macro- and microscopic details, supplemented the findings.

**Haematoxylin and eosin staining.** Paraffin-embedded tissue sections (5 μm) were placed on Superfrost slides (Avantor) and dried. Slides were deparaffinized and rehydrated through a graded ethanol series. Sections were stained with modified Harris haematoxylin (Sigma) and eosin Y solution (Sigma), with staining durations adjusted for optimal colour development. After air-drying, slides were mounted using Leica mounting medium and coverslips. Stained sections were digitized at 10× and 20× magnifications using the Zeiss AxioScan Z1.

**Immunohistochemistry.** Paraffin-embedded tissue sections (5 μm) were mounted on Superfrost slides and pre-heated at 60 °C for 1 h. Slides were deparaffinized in 100% Histoclear and rehydrated through graded ethanol. Heat-induced antigen retrieval was performed in pH 6 or pH 9 buffer (EnVision FLEX, Dako) using a Decloaking Chamber. Endogenous peroxidase activity was blocked with $H_2O_2$, and non-specific binding reduced using 2.5% horse serum.

Slides were incubated with primary antibodies (F3, pH 6 1:250; TWIST1, pH 6 1:800; CD45, pH 6 1:50) for 1 h at room temperature or overnight at 4 °C. After washing with Flex Buffer (Dako), slides were treated with horseradish peroxidase-conjugated secondary antibodies (Dako) and washed again. Peroxidase activity was visualized using DAB substrate (Dako), and slides were counterstained with haematoxylin for nuclear detection. Following washing, slides were dried and mounted with Leica Mounting Medium. High-resolution images (20× magnification) were captured using the Zeiss AxioScan Z1.

**Collagen fibre staining and imaging.** FFPE sections (4 μm) were deparaffinized through a standard xylene and ethanol series to water. They were incubated with Picrosirius Red (HB6179) for 60 min at room temperature. Once excess was blotted off, slides were washed twice in 200 ml 0.5% acetic acid solution for 1 min each, followed by 100% ethanol washes for 4 min each. After dipping in Histoclear, slides were covered with micromount medium and coverslip. Imaging was performed at 40× magnification on a Zeiss Axioscanner (brightfield imaging). Polarized light imaging was performed on an Evident VS200 slide scanner (polarized light illumination), using a 40× 0.95 NA air objective lens, with an iDS VS-264C colour camera. An automated stage was used to tile and stitch images of tissue sections using the VS200 ASW software.

**Quantitative PCR validation.** RNA from FFPE tissue blocks of fistulating CD patients and controls (CD, non-CD and healthy individuals) was reverse-transcribed into cDNA using the Reverse Transcription Master Mix (1006300, Standard Bio). TaqMan assays (Thermo Fisher) were pre-amplified, diluted, and used for qPCR.

The Fluidigm Flex Six IFC workflow (1006308) was employed for high-throughput amplification and target detection. Samples and assays were loaded into the IFC, followed by thermal cycling on the HX controller using the GE FlexSix Fast v2 protocol. Post-run data analysis was performed with Fluidigm Real-Time PCR software (68000088). GAPDH and HPRT1 served as endogenous controls, and gene expression was calculated using $C_t$ values. All assays were run in duplicate.

**Primary colonic fibroblast derivation and culture.** Primary colonic fibroblasts were isolated and expanded from endoscopic biopsies obtained from healthy donors using the Lamina Propria Dissociation Kit (Miltenyi Biotec 130-097-410). In brief, biopsies were cut into 1–2 mm fragments and washed in DMEM, high glucose, GlutaMAX Supplement, pyruvate (Gibco #39166-021) supplemented with 10 mM HEPES (Gibco #15690-056) and Penicillin/Streptomycin (Sigma P0781). To remove epithelial cells, tissue fragments were incubated 3 times in HBSS (Gibco 14025-050) containing Pen/Strep, 10 mM HEPES, 5 mM EDTA (Thermo Fisher 15575-038), and 2 mM dithiothreitol (Thermo Fisher P2325) at 37 °C for 5 min each, with vortexing between incubations. This process was repeated 3 times in total (cumulative incubation ~45 min) and the remaining tissue fragments were used for stromal cell isolation.

For each digestion, tissue fragments were incubated in enzyme mix prepared according to the Lamina Propria Dissociation Kit instructions at 37 °C for 15 min, repeated 3 times. Between incubations, the tissue was mechanically dissociated using a blunt-end needle to facilitate digestion. The resulting suspension was filtered through a 70-µm cell strainer and centrifuged at $300g$ for 5 min at 4 °C. Pelleted cells were resuspended in DMEM supplemented with 10% FBS, Pen/Strep, and 1× ITS (Gibco 41400-045), and seeded into 6-well plates. Cultures were maintained at 37 °C with 5% $CO_2$, and medium was changed every 2–3 days. Cells were passaged using TrypLE Express upon reaching confluency.

**Lentivirus production.** Lentiviral expression vectors for TWIST1 and OSR2 were obtained from F. Zhang via Addgene (1428908 and 144039, respectively)[54]. The empty vector was generated by excising the insert from the lentiviral vector using NheI-HF (NEB R3131S) and SpeI-HF (NEB R3133S) restriction enzymes, followed by gel purification and self-ligation of the linearized vector.

HEK293 cells were obtained directly from ATCC (CRL-1573), authenticated by the supplier, and tested mycoplasma-free. HEK293 cells were maintained in DMEM supplemented with 10% FBS and penicillin/streptomycin One day prior to transfection, $2 \times 10^6$ cells were seeded into T25 flasks. Transfection was performed the following day at 90–99% confluency. For each flask, 3.4 µg of the expression plasmid, 2.6 µg of psPAX2 (Addgene 12260), and 1.7 µg of pMD2.G (Addgene 12259) were transfected using 8.75 µl of Lipofectamine 3000 (Thermo Fisher L3000015), 7.5 µl of P3000 reagent, and 1.25 ml of Opti-MEM (Gibco 31985070). Culture medium was replaced 5 h post-transfection. Viral supernatant was collected 48 h later, filtered through a 0.45-µm PVDF membrane, aliquoted, and stored at −80 °C.

**Lentiviral transduction.** For transduction, primary fibroblasts were cultured to confluency in T25 flasks and incubated with an appropriate volume of lentiviral supernatant. After 48 h, the medium was replaced with fresh medium containing 1 µg ml$^{-1}$ Puromycin (Thermo Fisher 15490717), which was refreshed every other day. Cells were passaged after seven days of selection. Lentiviral titres were estimated by transducing cells with three different volumes of virus and assessing cell viability following three days of complete Puromycin selection.

**Immunofluorescence staining and quantification.** Primary human colonic fibroblasts were cultured in 8-well µ-Slides (ibidi 80826). Following removal of the culture medium, cells were washed with PBS and fixed with 4% paraformaldehyde (Thermo Fisher J19943K2) for 30 min at 4 °C. Fixed cells were washed twice with PBS and permeabilized with 0.2% Triton X-100 (Sigma T8787) in PBS for 30 min at room temperature. After washing, cells were blocked in PBS containing 1% BSA (Cell Signaling Technology 9998) and 0.3% Triton X-100 for 30 min, then incubated at room temperature for 2 h with primary antibodies diluted in the same blocking solution: anti-Collagen VII (Abcam ab198899, 1:100) and anti-α-SMA (Dako M0851, 1:400). Cells were then washed with PBS 3 times and incubated for 1 h at room temperature in the dark

with fluorophore-conjugated secondary antibodies, DAPI (Thermo Fisher D1306), and Alexa Fluor 647 Phalloidin (Thermo Fisher A22287). Cells were washed with PBS and mounted with Surgipath Micromount Mounting Medium (Leica 3801731). Samples were imaged with a Zeiss LSM980 confocal microscope and images were analysed using FIJI.

For quantification, approximately five images were acquired per condition to ensure sufficient cell numbers (>150 cells per condition). Maximum-intensity $z$-projections were generated for each fluorescence channel (Col7A, α-SMA and phalloidin), followed by thresholding and measurement of total signal intensity across the entire field of view. DAPI staining was used to identify nuclei and count cells using a standard FIJI workflow (contrast enhancement, Gaussian blur, thresholding, watershed segmentation, and particle analysis). Signal intensity for each marker was normalized to cell number, and for each patient, the normalized signal intensity in OSR2- or TWIST1-overexpressing conditions was calculated relative to the corresponding control.

**Quantitative PCR.** RNA was extracted from cultured primary fibroblasts using the RNeasy Plus Micro Kit (QIAGEN, 73034) and reverse-transcribed into cDNA using the High-Capacity cDNA Reverse Transcription Kit (Thermo Fisher 4374966). Quantitative PCR was performed using TaqMan assays (Thermo Fisher) according to the manufacturer's instructions.

**Bulk RNA-seq.** RNA was extracted from cultured primary fibroblasts using the RNeasy Plus Micro Kit (QIAGEN 73034) and submitted to Novogene for bulk RNA sequencing. Library preparation and quality control were performed by Novogene, targeting 50 million 150-bp paired-end reads per sample.

**Multiplexed immunofluorescence analysis.** Section rehydration, antigen retrieval and blocking FFPE tissue sections were baked overnight at 60 °C and then rehydrated using the automatic slide processor Gemini Autostainer (Epredia/Shandon Diagnostics) by serial incubation for 5 min twice in each 100% xylene, 100% ethanol, 95% ethanol, 70% ethanol, 50% ethanol, and finally PBS. Subsequently, tissue slides were incubated in 0.3% Triton X-100 for 10 min and washed twice with PBS for 5 min. Antigens were retrieved using the NxGen decloaking chamber (Biocare Medical), wherein slides were heated to 110 °C for 4 min in pH 6 citrate buffer (S1699; Agilent or H-3300-250; Vector Laboratories) at 6.1 PSI then cooled to 100 °C over 16 min, rinsed in deionized water and transferred into a pH 9.0 Tris-EDTA buffer (12.1 g Tris base, 3.7 g EDTA, 1 l distilled water) cooling from 100 °C to 85 °C over 20 min. Slides were then allowed to cool to room temperature and washed twice in PBS. Tissue sections were blocked overnight at 4 °C with 3% bovine serum albumin (A7906; Merck Life Science) and 10% donkey serum (C06SB; Bio-Rad) in PBS. Afterwards slides were washed twice with PBS, submerged in 1 mg l$^{-1}$ of DAPI (D357; Thermo Fisher Scientific) in PBS for 15 min at room temperature and subsequently again washed twice in PBS. Mounting medium combining 50% glycerol (G5516; Merck Life Science) in PBS was then added to the sections and the slides covered with no. 1 glass coverslips (Leica Microsystems).

Iterative imaging, staining and dye deactivation slides were imaged using the Cell DIVE system (GE Research) through iterative rounds of staining with fluorescently labelled antibodies and deactivation of the fluorophores. Initially, entire slides were scanned to select the ROIs, then the selected regions were scanned to acquire the tissue-inherent autofluorescence signal. Subsequently, coverslips were removed by incubation in ice cold PBS and the sections incubated for 2 h at room temperature with three primary antibodies diluted in 3% bovine serum albumin and 10% donkey serum in PBS underneath a polycarbonate coverslip (Grace Bio-Labs). In the initial round primary antibodies were detected using fluorophore-labelled cross-adsorbed secondary antibodies raised in donkey (Thermo Fisher Scientific) — in all subsequent rounds primary antibodies directly conjugated to fluorophores were

used. Fluorophores with excitation maxima around 488 nm, 555 nm and 647 nm were used, dependent on the antibody either Alexa Fluor 488, Alexa Fluor 555 and Alexa Fluor 647 (all Thermo Fisher Scientific) or CF 488 A, CF 555 and CF 647 (all Biotium). The stained slides were then washed twice in PBS, stained with DAPI and re-mounted as above. The selected ROIs per slide were then again scanned on the CellDIVE, and the autofluorescence signal previously acquired was removed from the signal of the staining round. The fluorophores were deactivated by de-coverslipping slides, washing them twice in PBS and incubating them at room temperature 3 times for 15 min in 0.1 M NaHCO$_3$ at pH 11.2 (S6297; Merck Life Science) with 3% H$_2$O$_2$ (216763; Merck Life Science) with intermittent incubation in PBS for 5 min. After dye inactivation, slides were again incubated with DAPI, mounted and scanned as above to acquire the background anew. Subsequently, slides were stained with primary antibodies conjugated to fluorophores and imaged as in the first round. This background-staining-deactivation cycle was repeated until all markers had been acquired.

Primary antibodies were either purchased conjugated to fluorophores as indicated or conjugated with either the Mix-n-Stain Antibody CF488A/555/647 dye Labelling Kits (92233, 92234 and 92238; Biotium), the Alexa Fluor 488/555/647 Antibody Labelling Kits (A88062, A88065 and A88068; Thermo Fisher Scientific) or with a combination of the oYo-Link Thiol Antibody Reagent (AT3001; AlphaThera) and CF 488A maleimide, CF 555 maleimide and CF 647 maleimide (all Biotium). For the Mix-n-Stain and the Alexa Fluor kit conjugation followed manufacturer instructions, for the AlphaThera oYo-Link thiol kit, per 1 µg of antibody 33 pmol of oYo-Link thiol reagent and 155 pmol of maleimide-fluorophores were used. The oYo-Link thiol reagent was mixed with the maleimide-fluorophores in PBS and incubated at 37 °C on a heat block (Thermomixer comfort; Eppendorf) shaking for 2 h at 600 RPM in the dark. Subsequently, the resulting fluorophore-linked oYo reagent was mixed with the antibody and crosslinked to it for 2 h using 365 nm UV light at 4 °C (LED photo-cross-linking device AT8001-D; AlphaThera). Finally, one part of the conjugated antibody mix was diluted with two parts of PBS-based antibody stabilizer (131 050; Candor Bioscience).

## Computational analysis

**Raw sequencing data processing.** All raw sequencing data were converted from BCL to FASTQ format using Illumina bcl2fastq (v.2.20.0.422) software. Up to one mismatch was allowed for each sample index barcode. Raw sequencing reads were then quality checked using FastQC software (v.0.11.9)[55].

For each sequenced scRNA-seq pool, 10X Genomics Cellranger software (v.7.1.0) was used to process, align and summarize unique molecular identifier (UMI) and cell barcode counts into raw counts matrices. Human hg38 (refdata-gex-GRCh38-2020-A) reference genome was used for all alignments and gene annotation. Paired CITE-seq and hashing antibody panel data were processed together with gene expression libraries using the Cellranger feature barcoding workflow. Antibody UMI counts were summarized using custom made joint feature barcoding sequencing reference, pooling together TotalSeq antibody sequences of hashing and protein expression antibodies. Feature barcoding sequences are deposited at the Gene Expression Ominibus (GEO).

Visium FFPE ST library FASTQ files were similarly processed using 10X Genomics Spaceranger (v.2.1.0) software. Paired H&E images and Slide IDs used as input are deposited on GEO. Human hg38 reference genome was used, together with Visium Human Transcriptome Probe Set v.v1.0_GRCh38-2020-A.

**scRNA-seq data analysis.** Raw UMI counts matrices from gene expression and feature barcoding libraries were imported into R for further processing. Cell calling on raw gene expression matrices of all 10X whitelist barcodes was carried out using 'emptyDrops' function from DropletUtils R package (v.1.24)[56]. Raw counts matrices were further corrected for Illumina index swapping with swappedDrops[57].

Cell QC metrics were calculated using scCustomize R package (v.2.1.2) and cell barcodes with low total UMI counts, low complexity and high mitochondrial RNA gene counts were filtered out from further analysis. The thresholds were determined per cell lineage basis from thresholding the empirical distributions after an initial clustering solution, as for instance epithelial cells typically contain higher percentage of mitochondrial UMIs than other cell types.

Following cell QC for each pool, cells were demultiplexed using hashing antibody counts matrices as follows. Non-hashing antibodies and tags absent from any given reaction were filtered out from the counts matrix. Counts matrices were normalized using centred log ratio transformation. Counts were clustered using clara $k$-mediods ($k$ = number of samples + 1). The 99th percentile of a negative binomial distribution fit was defined as a positive threshold for each antibody tag. Doublets were identified as cells positive for multiple tags and filtered out from further analysis. Given sufficiently large number of samples in each pool, the majority of doublets will be heterotypic due to random mixing and therefore this approach enables robust, experimental label driven doublet removal. Cells which could not be confidently assigned to a sample label were also filtered out from further analysis. Demultiplexed cells were visualized as $t$-SNE embeddings and sex-specific gene expression (for example, *XIST*) was further examined to ensure the cells were segregated correctly as expected with sample-of-origin assignments and patient meta data.

All samples were merged using R package Seurat (v.5.1.0)[58]. Gene expression counts were normalized. Highly variable genes were detected separately within each sample pool in order to deemphasize between-batches variability and the union was used as input for dimensionality reduction by PCA. Principal components were then further batch-corrected using Harmony (v.1.2.1)[59] algorithm for sample integration, and harmonized components were used as input for Louvain clustering and dimensionality reduction using uniform manifold approximation and projection (UMAP). Optimal clustering resolution was assessed using R package clustree (v.0.5.1)[60] to assess cluster stability. Cell clusters were annotated using a combination of known marker genes and cross-classification with previously published scRNA-seq reference atlas datasets[12–15,25,26,36,38,61] using Seurat label-transfer workflow.

**scRNA-seq differential expression analysis.** Differential gene expression analyses, including marker gene identification, were conducted using negative binomial generalized linear models for both ST and scRNA-seq data. To account for potential confounders, such as variability in gene detection rates and batch, donor or slide effects, these factors were incorporated as covariates into the model. Significance was assessed using Benjamini–Hochberg correction, considering genes with a false discovery rate (FDR) below 5% as significantly differentially expressed. Additionally, the MAST package[62] was used to detect genes that exhibit more binary on/off expression patterns by modelling both continuous and discrete components.

For pseudobulk analyses of scRNA-seq data, raw UMI counts were aggregated to pseudobulk samples as previously described. The DESeq2 package[63] was then used to normalize these counts and perform differential gene expression testing, again blocking for confounding variables. The Wald test was applied to derive *P* values, which were adjusted using the Benjamini–Hochberg correction.

**Pseudobulk PCA analysis.** Pseudobulk PCA analysis was carried out by aggregating UMI raw counts matrices of each individual sample following cell dehashing by summing counts for each gene. Epithelial, immune and stromal cells were considered separately. Sample pseudobulk counts were normalized using library size factor normalization, as implemented in DESeq2 R package (v.1.44.0)[63] for bulk RNA-seq analysis. Counts were further transformed using variance stabilizing

transformation (vst) and the top 1,000 most variable genes were selected to compute PCA. The first two principal components were visualized to assess the overall sample-level variability within each cohort. The above approach was also applied to assess sample-sample variability within the ST cohorts.

**Differential abundance analysis.** Cell-type or spatial region differential abundance analyses were carried out using three different approaches. R package miloR (v.2.0.0)[64] was used to carry out graph-based differential abundance analyses based on data UMAP embeddings, an analysis which is independent of cell cluster assignments and can highlight differences between populations which may exist on a phenotypic continuum rather than falling into discrete clusters. In each case, integrated, batch-corrected harmony components were used for nearest-neighbour graph construction ($k = 10$). The R package sccomp[65] (v.1.9.5) was used to further test cluster-level abundance differences. Cell proportions per sample were also compared using a two-sided Wilcoxon rank test.

**Transcription factor regulon analysis.** The pySCENIC pipeline (v.0.12.2b0)[66] was utilized to identify active transcription factor modules in scRNA-seq datasets. Initially, the normalized single-cell gene expression matrix was filtered to exclude genes expressed in fewer than 20 cells. The RcisTarget database, which provides transcription factor motif scores for gene promoters and transcription start sites based on the hg38 human reference genome, was downloaded from the following resource. The expression matrix was then further filtered to include only genes present in the RcisTarget database.

Next, a gene–gene correlation matrix was constructed for co-expression module detection using the GENIE3 algorithm[67]. SCENIC[66] was employed to analyse transcription factor networks, identifying co-expression modules enriched for target genes of candidate transcription factors from the RcisTarget database. Subsequently, the AUCell package was used to calculate a score for each transcription factor module in individual cells.

To pinpoint condition- or cluster-specific transcription factor modules, generalized linear models were applied to test for the dependence of transcription factor AUC values on conditions or clusters, incorporating batch effects and gene detection rates as covariates in the model to account for the significant influence of detection rate on AUC values. The resulting *P* values were adjusted for multiple comparisons using the Benjamini–Hochberg correction method.

**Cell cycle scoring.** Cell cycle phases were predicted using Seurat's CellCycleScoring function, using human 2019 cell cycle gene reference.

**Xenium ST cell segmentation.** Cell segmentation for Xenium ST was conducted by first re-segmenting cells using a transcript-density-based approach implemented in Baysor algorithm[68]. Nuclei-based segmentation from XeniumRanger outputs was integrated as a prior with a weighting factor of 0.7. –n-clusters parameter was set to 12, determined from the initial clustering solution of nuclei-segmented data. Following segmentation, a new cell-by-gene matrix was constructed by including only transcripts with a transcript assignment probability above 0.9. Transcripts with a confidence score below 0.9 were subsequently excluded from all downstream analyses. These analyses were similarly repeated for Xenium Prime assay, except cell boundary image-based cell segmentation was used as a prior instead.

Cell segmentation quality was assessed both visually and via transcriptome metrics. We calculated silhouette scores, as well as mutually exclusive co-expression ratio (MERC)[69] metrics to evaluate the amount of transcript wrongly assigned from adjacent cells. For MERC score, we assessed both universal cell-type markers, as well as scores focused on problem areas in intestinal tissue specifically. These included T cells and B cells, which form dense aggregates in follicles and therefore are difficult to segment and epithelial cells and T cells, where intra-epithelial lymphocytes wedged in between epithelial cells are often segmented as part of the nearby epithelial cells. MERC gene sets used are provided as part of Mendeley Data.

**Xenium ST clustering analysis.** Cell by gene counts matrix following cell re-segmentation was converted into Seurat R object for further analysis. Very small cells, which likely represent segmentation artefacts, and cells with very few detected transcripts (<15 per cell) were filtered out from further analysis. Cells with higher than expected (>99th percentile) negative probe signal were also filtered out from further analysis. As the remaining cells formed a prohibitively large dataset (>7M cells in total), undertook a sketch-based analysis workflow, as implemented in Seurat R package. In brief, for each tissue section, we sampled 5,000 cells using the LeverageScore method to select most informative cells. Data from the sketched subset cells was then dimensionality-reduced using PCA, followed by slide integration using Harmony, as described before. Nearest-neighbour graph was constructed for downstream clustering analysis and clusters were visualized using UMAP embeddings. The remaining cells from the full dataset were then projected onto the sketched data and onto the sketched UMAP model representation using ProjectIntegration and ProjectData functions. All subsequent analyses and statistics were computed on integrated dataset from all slides.

Low resolution clusters were annotated as broad cell-type lineages based on marker gene expression and label transfer of scRNA-seq reference data onto the sketched dataset. Each broad lineage identified (such as epithelium, fibroblasts or T cells) was then subset and further sub-clustered to identify cell subsets, which were similarly annotated as before. In the case where clusters were identified by our Xenium panel markers that did not have a one-to-one relationship to a scRNA-seq reference cluster and/or our panel did not contain subtype-defining markers, we annotated clusters based on their cell lineage combined with most specific gene marker expression (for example, *SPP1*+ macrophages, *NOS2*+ epithelium).

**Xenium ST cell niche detection.** Cell niche detection was performed using an unsupervised approach implemented as part of the spamsc R package (https://github.com/ChloeHJ/spamsc). Initially, for each field of view within the ST dataset, the spatial coordinates of cell centroids were extracted to determine the nearest neighbours for each cell, with the number of neighbours set to 30. Utilizing these coordinates, two separate neighbour matrices were constructed for each cell: (1) expression counts of each cell's nearest neighbours were aggregated, generating a cell-by-gene matrix that encapsulates the local transcriptomic environment; and (2) cell types, as defined by high-resolution cell subclusters from previous analyses, in each cell neighbourhood were counted and aggregated into a cell-by-cell-type matrix that encapsulates the local cell-type neighbours for each cell. This process was iteratively applied across all slides and the matrices were merged. To identify tissue niches, unsupervised clustering of the neighbourhood matrices was used by first carrying out PCA (for gene expression matrices only), integrating the principal components using Harmony across different slides and detecting niche clusters using Louvain clustering. Niches were labelled according to their location, tissue architecture features or cell-type enrichment.

**Xenium ST cell co-localization analysis.** For cell–cell co-localization analysis in Xenium ST dataset, the following approach was taken. First, using spatial coordinates of each cell centroid, nearest 30 spatial neighbours for each cell were determined and cell-type counts in each spatial cellular neighbourhood were aggregated. We then constructed a cell-type correlation matrix between each cell and its neighbours, representing the frequency of co-occurrence between different cell types. A separate co-occurrence matrix was constructed for each different experimental group. Cell-type co-occurrence networks were

visualized as graphs, where edges of low co-occurrence (<0.15), as well as negative co-occurrence (that is, cells always localize away from each other—for example, muscle cells and epithelium) were filtered out. Graphs were laid out using force-directed layout and visualized using ggpraph R package (v.2.2.1).

**Xenium ST custom panel design.** A custom 480-gene panel was designed for Xenium ST to delineate over 100 fine-grained cell types within the gut, as defined by previously published single-cell reference datasets and scRNA-seq data generated as part of this study. Additional genes from literature were included to cover cell types that are typically difficult to capture using droplet-based scRNA-seq approaches—for example, neurons and neutrophils, as well as squamous epithelium markers, a cell type that is typically not present in the intestine. Genes were expert-selected to ensure comprehensive representation across stromal, immune, and epithelial lineages while excluding highly abundant transcripts to prevent optical crowding. For each identified cell type, 5 to 10 redundant genes were incorporated to ensure cell typing robustness. The panel's performance was iteratively evaluated by assessing its ability to reconstruct scRNA-seq clustering patterns in the reference datasets. Probe balance was further optimized using 10X online Xenium Panel Designer tool. Xenium Prime 100 gene custom add-on panel was designed using the same procedure, focusing on key missing intestinal marker genes and disease-specific markers. All custom panel designs have been provided as part of Mendeley Data.

**scRNA-seq meta-analysis.** For scRNA-seq meta-analysis, data were downloaded from GEO, ENA or Zenodo repositories (see Data availability). Individual datasets were initially clustered to replicate analyses from their respective studies and fibroblast clusters from each study were subset for further downstream analyses. Fibroblast cells from public datasets and this study were then merged and integrated using harmony, correcting for both between dataset differences and known within-dataset batch variables.

To identify transcriptional programmes across transcriptomic data, cNMF[70] was applied to gene expression matrices. Prior to factorization, harmony[59] was used to correct for donor-specific batch effects. cNMF was run to extract 30 factors, each representing a gene expression programme recurrent across samples. Resulting gene loadings and usage scores were used for downstream interpretation and spatial mapping of inferred programmes. To visualize the relative contribution of cNMF-derived gene expression programmes (GEPs) across cell states, we calculated per-cluster usage proportions. First, harmony-corrected cNMF usage scores (that is, per cell factor loadings) were integrated into the Seurat metadata. For each annotated cell cluster, average usage scores were computed per factor. GEPs with low average usage (<0.05) were excluded to reduce noise. Remaining average scores were normalized within each cluster to sum to one, yielding relative proportions per GEP. These proportions were visualized as pie charts, with all non-disease cluster associated GEPS collapsed into an 'other' category for ease of visualization. All visualizations were generated using ggplot2 with coord_polar() for circular layout.

cNMF factors derived from meta-analysis were then applied to Xenium in situ 5100-plex dataset, calculating activity scores for each cell in the spatial dataset. Furthermore, cells from a diabetic wound-healing dataset[37] were similarly scored. Separately, cNMF factors were similarly calculated for fibroblast cells from Xenium in situ 5100-plex dataset, and reciprocally applied to the scRNA-seq meta-analysis to cross-map gene expression programmes identified with these different approaches.

**Xenium spatial autocorrelation analysis.** To identify gene pairs with spatially correlated expression patterns, we used two approaches. Firstly, local spatial neighbourhood gene expression matrices were computed as described in 'Niche analysis' and gene–gene Pearson's

correlation was calculated to detect co-localization gene expression patterns regardless of cell type. Additionally, for each gene pair, we also calculated a bivariate Moran's *I* statistic, which quantifies how the spatial distribution of each gene related to that of another. Statistical significance was assessed by permutation testing, in which the spatial positions were repeatedly randomized to generate a null distribution of Moran's *I* values. The resulting *P* values were adjusted for multiple testing, and significant positive or negative bivariate Moran's *I* values indicated spatially co-expressed or contrasting gene pairs, respectively.

**scRNA-seq spatial projection.** Spatial projection of fibroblast scRNA-seq data was carried out using the spamsc R package workflow (https://github.com/ChloeHJ/spamsc). In brief, to project single-cell fibroblast data onto a Xenium spatial dataset, we first identified genes shared between the scRNA-seq fibroblast dataset and the Xenium data, removing any features not expressed in either dataset. After merging these filtered datasets into a single Seurat object, we normalized and scaled them, then applied Harmony to batch correct for differences between the single-cell and spatial modalities. We extracted a reduced dimensional embedding and used a *k* nearest-neighbour approach to map fibroblast single cells onto the Xenium slide. For each fibroblast cell in the single-cell dataset, we searched for neighbouring Xenium spatial cells in the integrated reduced space; if spatial fibroblast cells were among the top neighbours, the spatial coordinates of the closest such neighbour were assigned to that cell. The accuracy of this mapping was assessed by comparing gene expression patterns between projected fibroblasts and the local spatial environment, using correlation analyses and evaluating enrichment of known fibroblast markers. These methods were applied using SPAMsc functions MergeDatasets and RunProjection.

**Xenium cell adjacency-dependent gene expression analysis.** To assess how a given cell type's gene expression patterns differ depending on the identity of its neighbouring cells, we implemented a procedure that aggregates gene expression counts based on spatial adjacency. First, we extracted cell centroid $x,y$ coordinates for each tissue section. For each section, a nearest-neighbour graph was constructed using Euclidean distances between cell centroids, and for each cell, the $k$ nearest neighbours were identified. We then focused on a particular cell type or group of cell types—for example, only epithelial cells, or FAS cells. Cells not belonging to the target identity group were excluded by zeroing out their counts. The expression values of the selected gene set (from a specified spatial assay) for the $k$ nearest neighbours of each cell were then summed, yielding a 'niche expression matrix' that represents the aggregated gene expression within local neighbourhoods populated by the specified cell types. By applying this process across multiple images and combining the results, we obtained a matrix that captures how gene expression in a cell of interest changes when it is surrounded by different cell populations. The aggregated niche expression matrix was then filtered to remove cells which do not have any cells of the selected cell type in their local neighbourhood. Gene expression counts of the remaining cells were then normalized and subjected to downstream differential expression analyses to quantify gene expression differences conditioned on spatial adjacency to particular cell types.

**Distance gradient-based analyses.** For fistula edge gradient-based analyses, fistula were selected where the tract was clearly delineated, with samples excluded where the edge of the tract was not fully captured. Fistula tract edges were then manually annotated using interactive XeniumExplorer software and coordinates of the polygonal annotations were exported as csv files. These were imported and converted into sf polygon objects. Then for each tissue section, centroid coordinates of tissue-detected cells were extracted, converted into spatial features, and distances to the nearest polygon edge were computed

using st_distance() from the sf package. These distances were added to the Seurat metadata and visualized using ImageFeaturePlot().

For downstream analysis, cells were grouped by cell type and donor (slide ID), and kernel density estimates of cell abundance as a function of distance were computed independently per donor using the density() function in R. To summarize across donors, mean densities were calculated at each distance point, and standard errors were estimated as the standard deviation of densities across donors divided by the square root of the number of samples per group. These were visualized as mean ± standard error using geom_line() and geom_ribbon(). To identify genes with expression patterns that vary as a function of distance from fistula edges, we modelled gene expression counts using negative binomial generalized additive models with distance as a smooth spline term. For each gene, significance of the distance effect was assessed by comparing the full spline model to a null model using likelihood ratio test. These tests were carried out within lineage—that is, distance varying genes were identified in fibroblasts and macrophages separately by subsetting these cells individually. Genes with significant spatial associations were visualized using heat maps, plotting loess-smoothed gene expression over distance.

Epithelial-edge distance-based analyses were carried out as above, except starting from manually annotated epithelial edges and excluding all non-epithelial cells from the analysis.

**Picrosirius red image analysis.** Individual images were first processed in Qupath software (v.0.5.0). In brief, tissue-covered areas of each slide slides were first selected as ROIs, with areas containing imaging or tissue artefacts filtered out. A groovy script was then used to systematically partition and export image tiles of defined size (512 × 512 pixels) with partial overlap (64 pixels), with x,y coordinates for each image tile exported along the image for downstream processing.

All subsequent image analyses were then performed in Python (v.3.9). All analyses were conducted using skimage for image processing[71,72], numpy and pandas for data handling, matplotlib and seaborn for visualization, and scipy and scikit-learn for statistical and machine learning operations. For each image, the red colour channel was extracted, pixel intensities were normalized to 0–1 range and uniform global thresholds of 0.15 and 0.18 for all images were set for red channel binarization, after initial exploratory analyses with Otsu's thresholding to guide parameter settings. The red channel image was then binarized at the two thresholds for parallel analyses, with the higher threshold excluding less well stained, sparser collagen bundles. Following binarization, we undertook morphological pre-processing of each image tile with erosion and opening using skimage.morphology operations to remove small specs of noise and smooth the collagen structures. Following pre-processing, each binarized tile was skeletonized using skimage. morphology.skeletonize, producing one-pixel-wide paths representing the collagen bundle backbone structures. Small, spurious branches were pruned. The resulting skeleton was used to compute several morphological and structural metrics as follows.

1. Length and thickness: connected components in the skeleton were first identified using skimage.measure.label and their perimeter approximations were used as a proxy for collagen length distribution. This yielded total length, mean length and selected quantiles (5th, 50th and 95th) as features for each tile. Thickness was estimated by calculating Euclidean distance transform with scipy.ndimage.distance_transform_edt on the binary mask and extracting the distance values along the collagen skeleton. Total, mean and quantile values for each image patch were calculated as features.
2. Connectivity and morphological descriptors: the number of connected components within the skeleton was used to calculate collagen network connectivity in each image tile. We also calculated perimeter-area ratio, eccentricity (shape elongation) and average elongation (major/minor axis ratio of fitted ellipses) using skimage. measure.regionprops.

3. Fractal and textural measures: fractal dimension was estimated using a box-counting method across multiple scales. Lacunarity, a measure of textural heterogeneity, or 'gappyness', was computed at four different box sizes for each image patch. Additionally, Shannon entropy, which can provide a measure of image complexity, was calculated using skimage.measure.shannon_entropy.
4. Anisotropy and orientation: anisotropy was calculated from the image patch skeletons by calculating the variance in the orientations of the skeleton segments. Anisotropy measures the overall directional coherence of the network; that is, how parallel each segment is to each other.
5. Mean free path and collagen fraction: a mean free path along both collagen regions and regions between collagen areas was estimated by computing pairwise distances between feature points. To compute the total collagen fraction, the proportion of thresholded pixels at both thresholds used was calculated for each image path.

All metrics were mapped to $x,y$ spatial coordinates of the image patches for further spatial analysis and visualization.

For image clustering analysis, a variational autoencoder was trained on exported image tiles, integrated with a pretrained VGG16 encoder as a feature extraction backbone. In brief, a custom image tile generator class was implemented to load and preprocess image tiles to resize, normalize intensities and apply gamma correction. The generator also computed Shannon entropy for each tile to determine sample weights to reduce the weights of any texturally uninformative tiles (for example, edge tiles) during training. For VAE architecture, the encoder consistent of an initially frozen VGG16 network pretrained on ImageNet to extract robust feature representations, followed by a fully connected layer and two dense layers producing the latent parameters with a latent dimensionality of 64. A sampling layer generated latent vectors from these parameters. The decoder employed a series of transposed convolutional and up sampling layers to reconstruct the original image resolution from the latent space. The entire model was compiled with a mean-squared-error loss and trained until convergence on total of 301,941 image tiles. 64 dimension latent vector was then extracted for each image tile and used as input for Louvain clustering (resolution – 0.3). Clusters were intersected with morphological and structural tile metrics calculated as described above, and annotated on the basis of average collagen density per cluster, with highly fibrotic tissue regions annotated as 'very high' clusters and tissue regions with lower levels of collagen deposition (such as mucosa) as 'low' or 'mid' regions. Comparative analysis of relative cluster abundance between fistulae and control samples were carried out using sccomp R package, as described above.

**Fibroblast lineage prediction analysis.** pySCENIC was used to reconstruct transcription factor regulons for fibroblast cell subsets specifically and cell-by-AUC matrices for each transcription factor were imported into R for further processing. Transcription factor activity scores were scaled and used as input for dimensionality reduction by PCA. Fibroblasts cells from healthy control samples were grouped into broad fibroblast subtypes -Stromal 1–4 subsets and served as a reference dataset, while all other cells, including fistula-associated fibroblasts, were then classified with respect to the reference using Seurat's label-transfer workflow (FindTransferAnchors and TransferData). For each query cell, a prediction probability was calculated on the basis of transcription factor regulon activity score embeddings, scoring the cell as most similar to each of the four main fibroblast subtypes, Stromal 1–4.

**Pathway enrichment analysis.** Gene Ontology (GO) and Reactome pathway enrichment analyses for both scRNA-seq and ST data were conducted using the clusterProfiler R package[73] (v.4.12.6). Gene identifiers were mapped with the org.Hs.eg.db (v.3.19.1) annotation DBI package (v.1.66.0). Individual cluster marker sets and differentially expressed

genes were analysed for overrepresentation, using all expressed or detected genes as the background reference. Hypergeometric $P$ values were adjusted for multiple testing through the Benjamini–Hochberg method. The enrichment results were visualized with clusterProfiler and ggplot2 packages. For pathway activity analysis at the single-cell or spot level, pathway scores were derived as gene module scores using the AddModuleScore function in the Seurat R package.

**Visium ST clustering analysis.** Raw UMI count matrices output by 10X Spaceranger pipeline, corresponding images, spot coordinates, and scale factors were first imported into R for further processing. The spot matrix was filtered to retain only spots that overlapped with tissue. To assess background signal, we fitted a negative binomial distribution to the total UMI counts of spots located outside tissue sections. Using this distribution as a reference, we further removed any tissue-overlapping spots whose UMI counts were indistinguishable from the non-tissue background, as these likely represented areas of under-permeabilization. Most spots removed via this procedure were either section-specific anomalies or associated with tissue artefacts. Additional tissue irregularities (e.g., folds, tears) were annotated based on H&E images, and the corresponding spots were also excluded from the downstream analysis manually. Spots under coverslip air bubbles, however, were retained, as these defects affected only the H&E image and not the underlying tissue integrity.

Next, raw UMI counts were normalized using SCTransform, a regularized negative binomial regression-based method[74], to account for variability in total RNA content across spots. PCA was used for dimensionality reduction, using the union of spatially variable genes across all slides detected using Moran's I spatial autocorrelation method. For integrative analysis, spots from individual slides were combined using the Harmony[59] algorithm to correct for slide-specific effects. Subsequent clustering analysis was performed using the Louvain algorithm at a resolution of 0.5, and results were visualized using UMAP. Clusters were visualized on H&E images with a spot size scaling factor of 1.6. The resulting integrated clusters were compared to those derived from individual slides to confirm that no biologically relevant heterogeneity was lost after batch correction. In addition, we ensured that corresponding tissue regions across slides converged into the same integrated clusters, confirming successful alignment of equivalent anatomical areas.

**Visium ST cell-type deconvolution.** Visium ST cell-type deconvolution was carried out using R package CARD[75]. Transcriptomic signatures and histopathological features detected in our Visium ST analysis indicated the presence of both squamous epithelium and neutrophils in our disease tissue sections, which were not captured by our scRNA-seq analysis. To account for this missing data and ensure a more accurate, robust cell-type deconvolution result, a combined scRNA-seq single-cell reference dataset was constructed as follows. scRNA-seq from the intestine generated in this study was merged with neutrophil cells identified in the GEO GSE163668 dataset[76], as well as squamous epithelium cells identified in the GEO GSE201153 dataset[77], the data were normalized, merged and re-embedded together using only common genes between gene matrices of all three reference datasets and Visium ST data.

**Visium ST cell–cell signalling analysis.** To identify region-specific, spatially co-localized cellular signalling events, we undertook analysis as previously described in[36]. We first obtained receptor–ligand databases from[78] and[79]. Each ST spot was scored for receptor–ligand co-expression by incorporating weighted expression from neighbouring spots. We calculated pairwise Euclidean distances for all ST spots and assigned distance-based weights, setting weights to zero for spots beyond two neighbours and normalizing weights for closer spots. For each spot, we computed a distance-weighted receptor–ligand product score, scaled by the number of contributing spots to account for tissue edges.

To determine significance, we performed 100 random permutations of spot locations to generate an empirical background distribution and calculated $P$ values for each receptor–ligand pair based on this distribution. After applying Benjamini–Hochberg correction for multiple testing, spots with FDR < 5% were considered positive for receptor–ligand cross-talk. We then prioritized region-specific interactions using generalized linear modelling, assessing the dependence of receptor–ligand scores on spatial clusters while controlling for gene detection rates. Condition-specific interactions were similarly modelled, and all significant receptor–ligand pairs were further subject to multiple testing correction to ensure robust identification of region- and condition-specific cellular communication events.

**Bulk RNA-seq analysis.** Raw sequence reads were quality checked using FastQC software[55]. Cutadapt[80] was used to trim poor-quality bases and Illumina universal adapter sequences from raw reads before alignment.

The human hg38 reference genome analysis set was obtained from the University of California Santa Cruz (UCSC) ftp site[81]. Full lentiviral construct sequences (Addgene plasmids #142908 and #142826) were included as additional reference contigs and indexed together with hg38 as a reference genome using STAR aligner[82]. Reads were then aligned to this custom reference.

Picard tools[83] were used to mark duplicate sequences as an additional quality control step. Raw gene expression counts were summarized with featureCounts[84] using a custom hg38 and plasmid GTF file containing joint annotations. The MultiQC tool was used to aggregate quality metrics. Sample quality metrics and raw read counts were imported into R for further processing. The DESeq2 R package[63] was used to estimate library size factors, normalize counts and perform differential expression analyses. Benjamini–Hochberg multiple testing correction was used to compute FDR, and genes were considered significantly differentially expressed at <5% FDR. PCA was performed in R using the top 1,000 most variable genes, with normalized DESeq2 variance-stabilized transformation expression as input. Combat[85] was used to correct donor-specific effects for heat map visualizations only.

**Multiplexed immunofluorescence quantification.** Multiplexed immunofluorescence images were analysed by selecting ROIs in each sample, avoiding areas with tissue artefacts or mismatched initial and final DAPI staining. In fistula samples, ROIs were annotated as granulation tissue, lesion, lesion edge or fibrotic zone, in order to be consistent with ST region definitions. Cell segmentation was performed in QuPath software (v.0.5.0) using DAPI for nuclear detection and expansion of cell boundaries by 5 μm to approximate full cell outlines. Mean marker intensity was then quantified per cell. To enable cross-sample comparison, intensities were clipped at the 95th percentile within each sample to exclude high-intensity outliers. Cells were binarized as positive or negative per marker based on sample-specific thresholds, and the proportion of marker-positive cells was calculated per region and compared across samples.

### Reporting summary

Further information on research design is available in the Nature Portfolio Reporting Summary linked to this article.

## Data availability

Raw sequencing data have been deposited to GEO (series GSE284232), accessions GSE283945 (Visium ST), GSE305631 (bulk RNA-seq) and GSE284230 (scRNA-seq). All processed scRNA-seq, Visium ST and Xenium ST Seurat data RDS object files, image data and scRNA-seq and ST data tables have been deposited as Mendeley Data (https://doi.org/10.17632/tn972brm9s.1; https://doi.org/10.17632/mxy6p-6wfmy.1 and https://doi.org/10.17632/64fkdfcpzb.1. Data for scRNA-seq

meta-analysis were obtained from GEO: GSE189185, GSE114374, GSE134809, GSE260842, GSE260833, GSE266546; ArrayExpress: E-MTAB-8901; Broad Single Cell Portal SCP259, SCP1884; and *Zenodo* https://doi.org/10.5281/zenodo.13768607 (ref. 86). Additional scRNA-seq reference data for cell-type deconvolution was downloaded from GEO: GSE201153 and GSE163668. All spatial gene expression data can be accessed through the manuscript companion data portal app at https://simmonslab.shinyapps.io/cd-fistula-data-portal/. Source data are provided with this paper.

## Code availability

Code used in the data analysis has been deposited on Github: https://github.com/agneantanaviciute/cdfistulaspatialtranscriptomics and https://github.com/ChloeHJ/spamsc.

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

**Acknowledgements** Patient recruitment was undertaken by ORNID research nurses and Oxford TGLU Biobank, led by S. Fourie and J. Chivenga, respectively—we thank them and the patients. We acknowledge the support of the Oxford University Hospitals NHS Foundation Trust Colorectal Surgical team and the TGLU Investigators. We acknowledge support of J. Koth and C. Lai from the WIMM Imaging Facility; C. Waugh and K. Clark from the WIMM Flow Cytometry Lab; and WIMM Advanced Single Cell OMICS Facility and T. Rostron from the WIMM Sequencing Laboratory. We thank E. Drydale, S. Chava, K. Holden and J. Bancroft for advice and for generation of microscopy data. The facility was initially set up with Wellcome Trust Core Award Grant Number 203141/Z/16/Z and is located in the Centre for Human Genetics, Nuffield Department of Medicine, University of Oxford. We acknowledge the support of S. Jones, D. Maldonado-Perez and E. Kite from OCHRe; and J. Hollis for assistance with data collection. Funding for this study was generously provided by the UK Medical Research Council; The Leona M. and Harry B. Helmsley Charitable Trust; the Oxford NIHR Biomedical Research Centre and the NIHR Clinical Research Network (CRN) Thames Valley; an NIHR Senior Investigator Award (NIHR201410); a Wellcome Investigator Award (219523/Z/19/Z) (A.S.); a Wellcome Career Development Award (315684/Z/24/Z) to A. Antanaviciute; NIHR Academic Clinical Lectureships to D.F.-C. and T.G.; an MRC Clinical Academic Partnership Award to J.B. and a CSC-COI DPhil scholarship to Z.Y. C.M. and T.G. were also supported by Oxfordshire Health Services Research Committee (OHSRC, ref. 1407), part of Oxford Hospitals Charity. We thank OUHFT and Bayreuth Medical Center patients for donating samples.

**Author contributions** A. Antanaviciute, C.M., A.S., X.Q. and T.G. wrote the manuscript. C.M., A. Antanaviciute and A.S. conceived and designed the study. C.M., X.Q., M.J., T.G., Z.Y., E.B., V. Lai, V. Lentsch, P.G.C., D.F.-C., A. Aulicino, L.D., H.-W.C., P.S.-Z. and Z.C. designed, performed and analysed wet laboratory experiments. C.M., Z.Y., T.G. and A. Aulicino performed scRNA-seq experiments. C.M. and Z.Y. performed Visium ST experiments. D.F.-C., V. Lentsch, T.G., V. Lai, L.D. and C.M. performed Xenium ST experiments. X.Q., M.J. and H.-W.C. performed OSR2 and TWIST1 overexpression experiments, bulk RNA-seq, immunofluorescence, qPCR and cell imaging. T.G. and M.J. performed picrosirius red imaging. P.G.C., M.J., T.G. and C.M. performed immunohistochemistry experiments. C.M. performed biomark multiplex qPCR experiments. E.F., M.V. and R. Tandon undertook histopathological analysis and review. C.M., B.G., M.B., K.B., N.C., S.M., M.V., L.-P.H., J.B. and A.S. coordinated clinical samples and provided input. C.M and S.M. extracted patient metadata from electronic patient records. R. Teague, M.G., R.K., P.V.G. and C.C. provided technical assistance with in situ ST workflows. A. Antanaviciute, J.S.P. and K.X. performed ST analyses. A. Antanaviciute performed collagen imaging analysis. A. Antanaviciute and C.H.L. developed spatial analysis methodologies. A. Antanaviciute and C.H.L. performed scRNA-seq analysis. A. Antanaviciute and J.W. performed meta-analyses. C.M. carried out meta-analysis literature review. A. Antanaviciute analysed bulk RNA-seq data. S.H. and J.L. performed multiplexed immunofluorescence experiments. S.H., J.L. and A. Antanaviciute analysed immunofluorescence data. A.S. and A. Antanaviciute co-directed.

**Competing interests** The authors declare no competing interests.

**Additional information**
**Correspondence and requests for materials** should be addressed to Agne Antanaviciute or Alison Simmons.

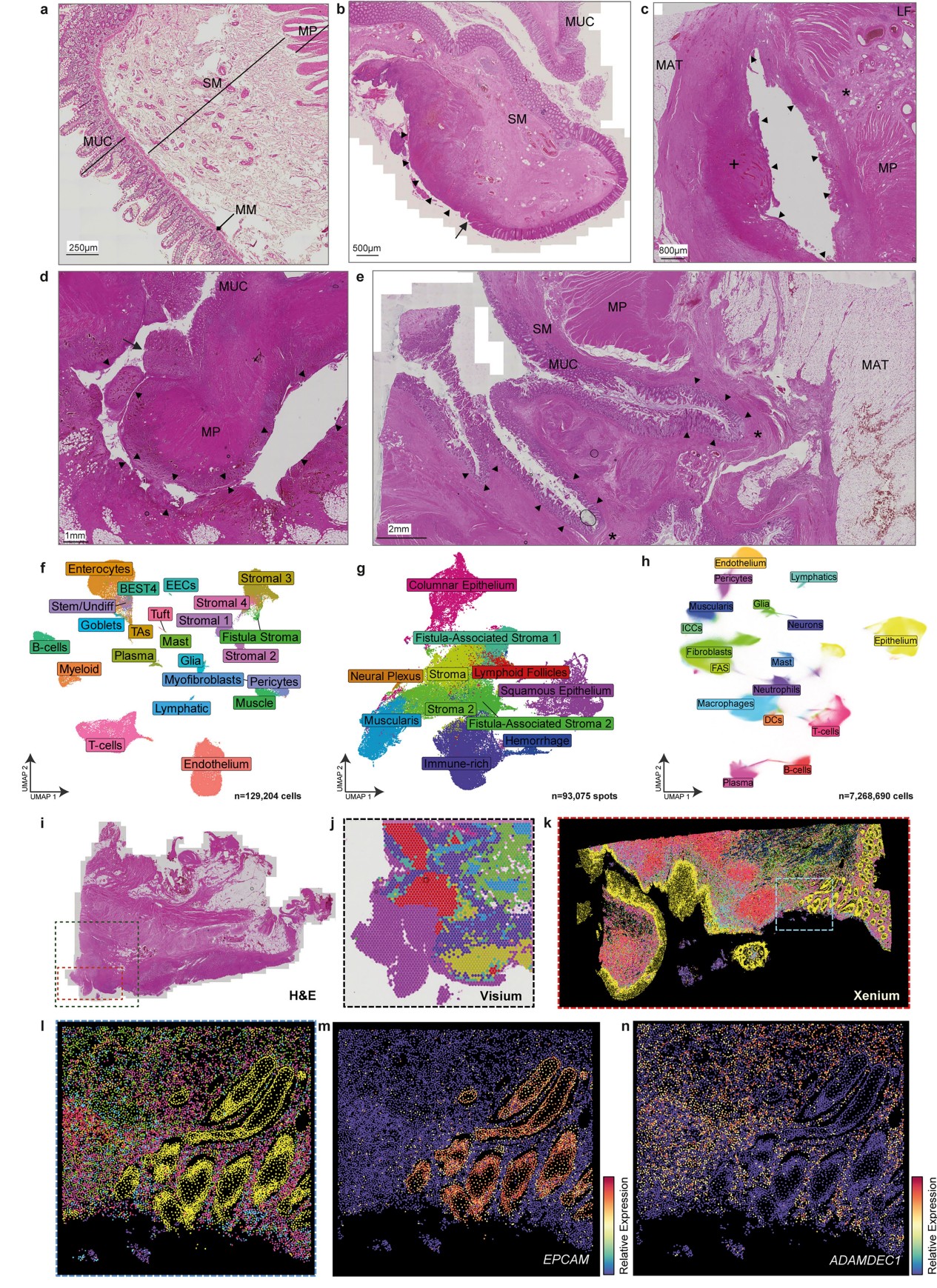

**Extended Data Fig. 1** | See next page for caption.

**Extended Data Fig. 1 | Spatial atlas of Crohn's fistulae.** Related to Fig. 1. A. Haematoxylin and eosin (H&E) tissue section from healthy ileum displaying characteristic full-thickness bowel layers, including mucosa (MUC), muscularis mucosae (MM), submucosa (SM) and muscularis propria (MP). n = 92 total samples imaged. B. Haematoxylin and eosin (H&E) tissue section from colonic CD fistula showing loss of the lining columnar epithelium (black arrow) toward the fistula tract (arrowheads). The fistula tract is lined with granulation tissue exhibiting neovascularisation. n = 92 total samples imaged. C. Haematoxylin and eosin (H&E) tissue section from ileocolonic CD fistula (arrowheads) lacking lining epithelium, with mesenteric adipose tissue (MAT), an inflammatory infiltrate and lymphoid follicle (LF) formation. Adjacent mural abscess formation (+) and disruption of the muscularis propria (asterisk) are evident. n = 92 total samples imaged. D. Haematoxylin and eosin (H&E) tissue section from colonic CD fistula demonstrating epithelial loss along the fistula tract (arrowheads) with underlying submucosal fibrosis. Granulation tissue, neovascularisation and dense inflammatory infiltration line the tract. n = 92 total samples imaged. E. Haematoxylin and eosin (H&E) tissue section from epithelialised ileocolonic CD fistula (arrowheads) originating from the ileocecal valve. The fistula tracts branch deep into the muscularis propria, disrupting muscle fibres (asterisk), but do not penetrate all intestinal layers or extend into MAT. While no acute inflammation is present, fibrotic scarring is evident through hypertrophy of the muscularis propria and submucosal fibrosis. n = 92 total samples imaged. F. UMAP visualisation of integrated data across scRNA-Seq cohort summarised in Fig. 1b. Each point represents a cell, points are coloured by broad cell types. n = 22 donors, n = 82 samples, n = 129,204 cells. G. UMAP visualisation of integrated data across Visium ST cohort, summarised in Fig. 1b. Each point represents a tissue covered spot, points are coloured by broad tissue regions. n = 34 samples, n = 93,075 spots. H. UMAP visualisation of integrated data across Xenium ST cohort, summarised in Fig. 1b. Each point represents a cell, points are coloured by broad cell types. n = 53 samples, n = 7,268,690 cells. I. A representative tissue section from an epithelialised, perianal fistula stained with H&E (n = 92 samples), with dashed lines indicating a region selected for profiling with Visium ST in black, and Xenium ST in red. J. Visium ST spot cluster overlay of tissue section shown in I, with reference UMAP embedding and region annotation visualised in G. (n = 34 samples). K. Xenium ST cell type cluster overlay of tissue section shown in I, with reference UMAP embedding and cell type annotation visualised in H. Dashed blue lines indicate a smaller ROI visualised in I-N. (n = 53 samples). L. Zoomed in view visualising ROI indicated by dashed blue lines in panel K. Each point represents a centroid of a cell, with cells coloured by cell type as indicated in panel H. M. Zoomed in view visualising ROI indicated by dashed blue lines in panel K, visualising expression of an epithelial cell marker gene *EPCAM*. N. Zoomed in view visualising ROI indicated by dashed blue lines in panel K, visualising expression of a fibroblast cell marker gene *ADAMDEC1*.

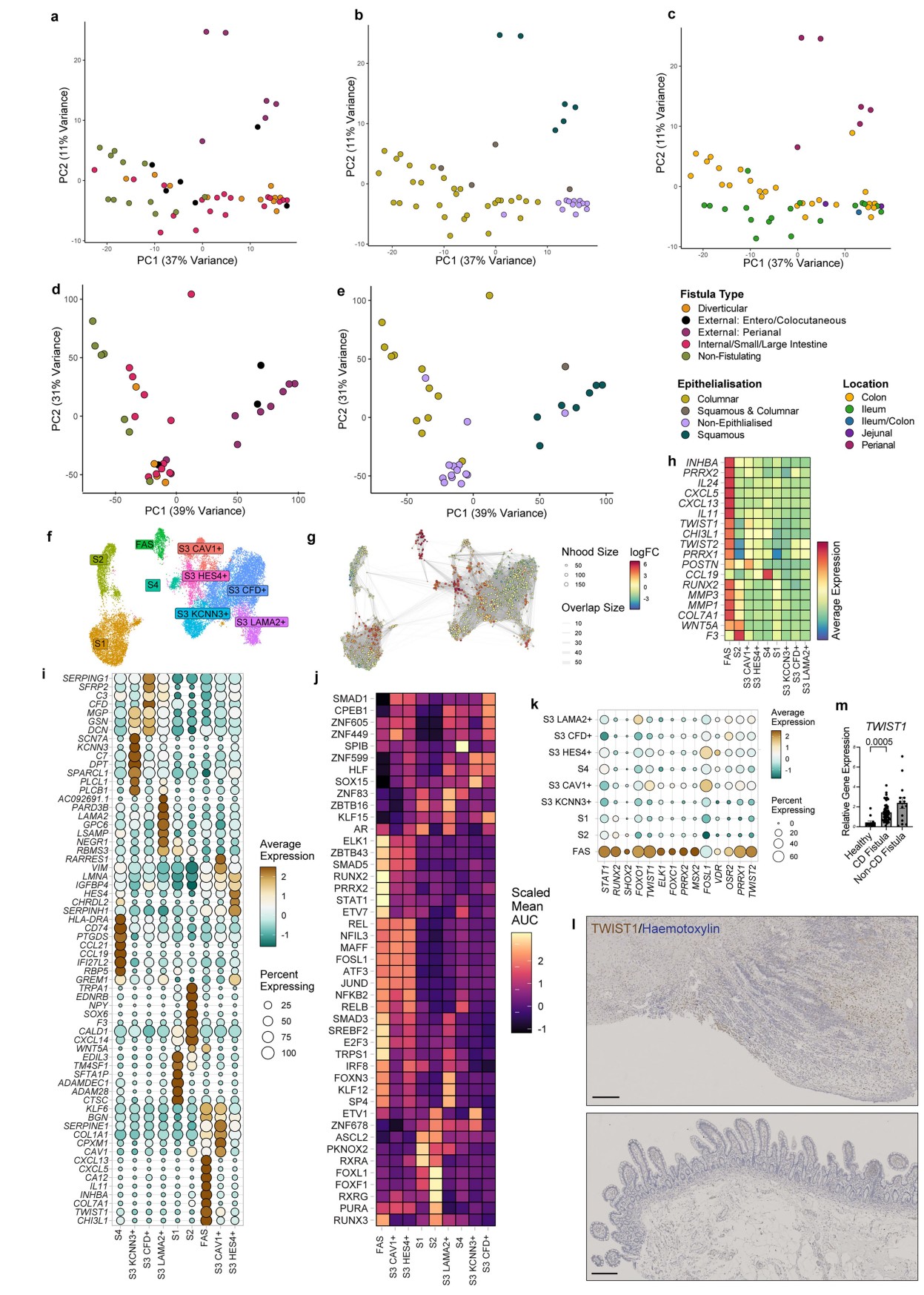

**Extended Data Fig. 2** | See next page for caption.

**Extended Data Fig. 2 | Fibroblast states in Crohn's fistulae.** Related to Fig. 1.
A. Section pseudobulk PCA plot of all Xenium ST tissue sections, coloured by broad fistula type. n = 53 samples. B. Section pseudobulk PCA plot of all Xenium ST tissue sections, coloured by epithelialisation status. n = 53 samples. C. Section pseudobulk PCA plot of all Xenium ST tissue sections, coloured by sample location. n = 53 samples. D. Section pseudobulk PCA plot of all Visium ST tissue sections, coloured by broad fistula type. n = 34 samples. E. Section pseudobulk PCA plot of all Visium ST tissue sections, coloured buy epithelialisation status. n = 34 samples. F. UMAP embedding visualising fibroblast sub-clusters detected in scRNA-Seq cohort. Fibroblast nomenclature is used as per Kinchen et al.[15]: Stromal 1 (S1) cells refer to mucosal fibroblasts, Stromal 2 (S2) cells refer to telocytes/epithelial crypt niche supporting fibroblasts, Stromal 3 (S3) cells are submucosal fibroblasts and Stromal 4 (S4) cells are follicular niche organiser cells. n = 22 donors, n = 23,176 fibroblast cells. G. A differential abundance plot of local cellular neighbourhoods within the UMAP embedding space, comparing fistulating CD samples with healthy controls. Positive log fold change (FC) indicates an increase in cells from fistulating samples. Neighbourhood differential abundance was tested using miloR with negative binomial GLMs and Benjamini–Hochberg correction for multiple testing. n = 22 donors. H. Heatmap of selected key significant gene markers of FAS cluster cells. Mean, scaled values per cluster are shown. I. Dotplot visualising fibroblast sub-cluster top marker gene expression. J. Heatmap visualising mean scaled TF regulon AUC values per fibroblast subcluster. K. Dotplot visualising FAS-specific TF gene expression across fibroblast sub-clusters. L. Representative (n = 3) immunohistochemistry (IHC) image of a CD enterocutaneous fistula section stained for TWIST1. Top ROI shows deeper stromal layers of the fistula tract, while bottom indicates an ROI corresponding to the mucosa. Scale bars: top – 250 μm, bottom–200 μm. TWIST1-CD45 co-stain (n = 11) is shown in Supplemetary Information. M. Relative gene expression of *TWIST1*, assessed by qPCR in the validation cohort. Data are presented as mean values, with error bars representing the standard error of the mean (SEM). Statistical significance was determined using a two sided unpaired t-test for non-parametric data, with significance set at $p < 0.05$. n = 84 samples.

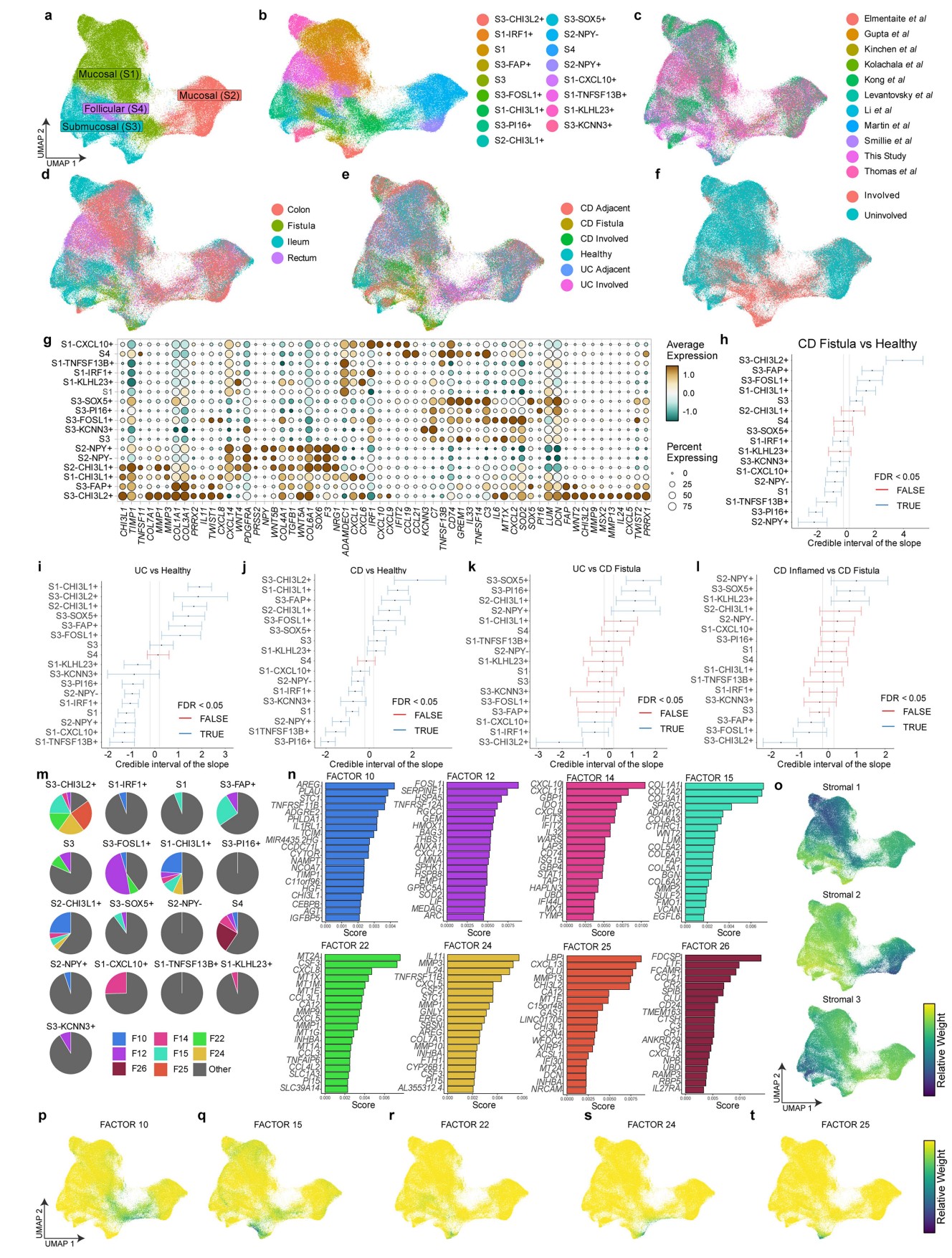

**Extended Data Fig. 3** | See next page for caption.

**Extended Data Fig. 3 | scRNA-Seq fibroblast meta analysis in inflammatory bowel disease.** Related to Fig. 1. A. UMAP overlay visualising all fibroblast cells from integrated scRNA-Seq meta analysis, coloured by broad fibroblast lineage (n = 487 samples, 185,572 cells). B. UMAP overlay visualising all fibroblast cells from integrated scRNA-Seq meta analysis, coloured by high resolution clusters (n = 487 samples, 185,572 cells). C. UMAP overlay visualising all fibroblast cells from integrated scRNA-Seq meta analysis, coloured by study origin (n = 487 samples, 185,572 cells). D. UMAP overlay visualising all fibroblast cells from integrated scRNA-Seq meta analysis, coloured by sample colocation (n = 487 samples, 185,572 cells). E. UMAP overlay visualising all fibroblast cells from integrated scRNA-Seq meta analysis, coloured by sample type (n = 487 samples, 185,572 cells). F. UMAP overlay visualising all fibroblast cells from integrated scRNA-Seq meta analysis, with samples grouped into actively involved/inflamed tissue and healthy or non-involved control groups (n = 487 samples, 185,572 cells). G. Dotplot visualising gene expression markers for the clusters identified from integrated fibroblast scRNA-Seq meta analysis, with cluster reference shown in A. H. Differential abundance analysis of fibroblast scRNA-Seq meta analysis subpopulations comparing all CD fistula samples with healthy controls (n = 487 samples). Bands represent the 95% Bayesian credible interval of the slope (logit fold change in cluster proportion per unit change in the covariate), indicating the range of effect sizes compatible with the data, given the model. I. Differential abundance analysis of fibroblast scRNA-Seq meta analysis subpopulations comparing all UC samples with healthy controls (n = 487 samples). Bands represent the 95% Bayesian credible interval of the slope (logit fold change in cluster proportion per unit change in the covariate), indicating the range of effect sizes compatible with the data, given the model. J. Differential abundance analysis of fibroblast scRNA-Seq meta analysis subpopulations comparing all non-fistulating CD samples with healthy controls (n = 487 samples). Bands represent the 95% Bayesian credible interval of the slope (logit fold change in cluster proportion per unit change in the covariate), indicating the range of effect sizes compatible with the data, given the model. K. Differential abundance analysis of fibroblast scRNA-Seq meta analysis subpopulations comparing all UC samples with CD fistulae (n = 487 samples). Bands represent the 95% Bayesian credible interval of the slope (logit fold change in cluster proportion per unit change in the covariate), indicating the range of effect sizes compatible with the data, given the model. L. Differential abundance analysis of fibroblast scRNA-Seq meta analysis subpopulations comparing all non-fistulating CD samples with fistulating CD samples (n = 487 samples). Bands represent the 95% Bayesian credible interval of the slope (logit fold change in cluster proportion per unit change in the covariate), indicating the range of effect sizes compatible with the data, given the model. M. Consensus negative matrix factorisation (cNMF) analysis of fibroblast scRNA-Seq meta analysis cells. Pie charts indicate average disease-associated factor usage per cluster, with other factors shown in grey. N. Barplot visualising cNMF factor top gene loadings for selected factors enriched in disease associated fibroblast clusters. Colours as in M. O. UMAP overlay of cNMF factors corresponding to broad fibroblast lineage Stromal 1, Stromal 2 and Stromal 3 cells in scRNA-Seq meta analysis (n = 487 samples, 185,572 cells). P-T. UMAP overlay visualising selected cNMF factor cell scores (n = 487 samples, 185,572 cells).

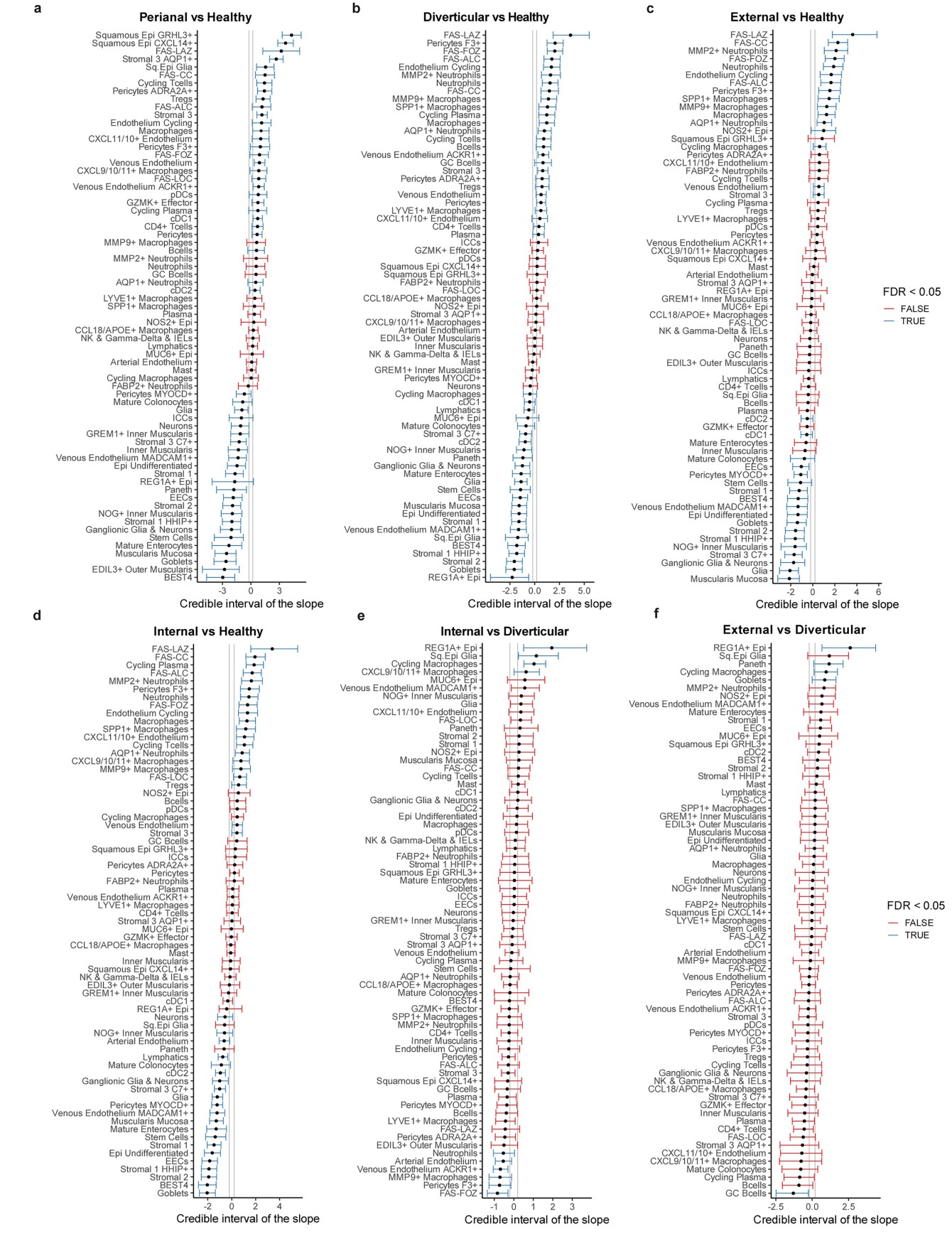

**Extended Data Fig. 4** | See next page for caption.

**Extended Data Fig. 4 | Cellular abundance changes in Crohn's fistulae.**
Related to Fig. 1. A. Differential cell subtype abundance analysis comparing all cell types detected in Xenium ST cohort between CD perianal fistula samples and healthy controls (n = 53 samples). Bands represent the 95% Bayesian credible interval of the slope (logit fold change in cluster proportion per unit change in the covariate), indicating the range of effect sizes compatible with the data, given the model. B. As in A, except diverticular disease fistula and healthy control sections are compared (n = 53 samples). C. As in A, except external CD fistula sections and healthy control sections are compared (n = 53 samples). D. As in A, except internal CD fistula sections and healthy control sections are compared (n = 53 samples). E. As in A, except internal colonic CD fistulae are compared with diverticular disease fistulae (n = 53 samples). F. As in A, except external colonic CD fistulae are compared with diverticular disease fistulae (n = 53 samples).

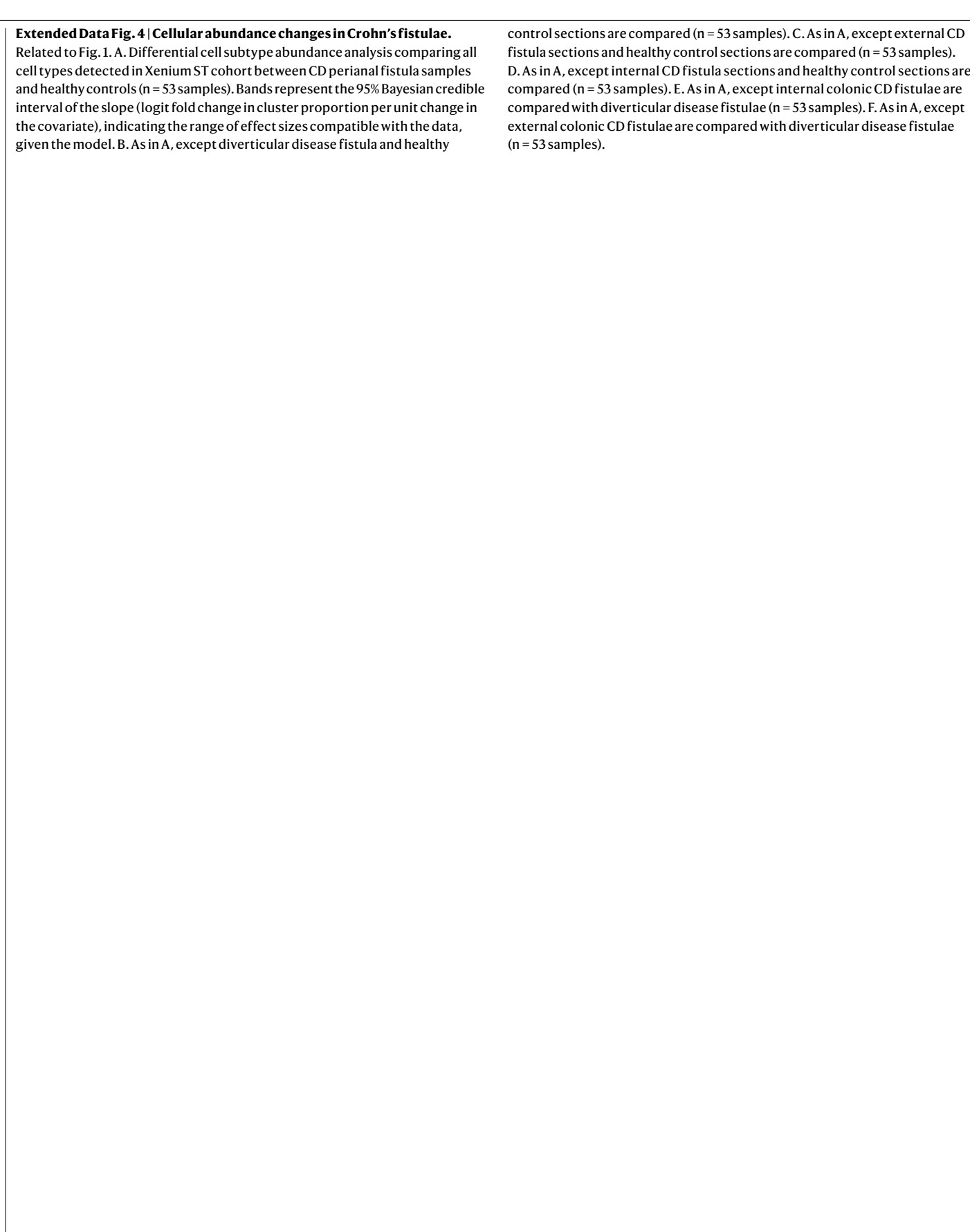

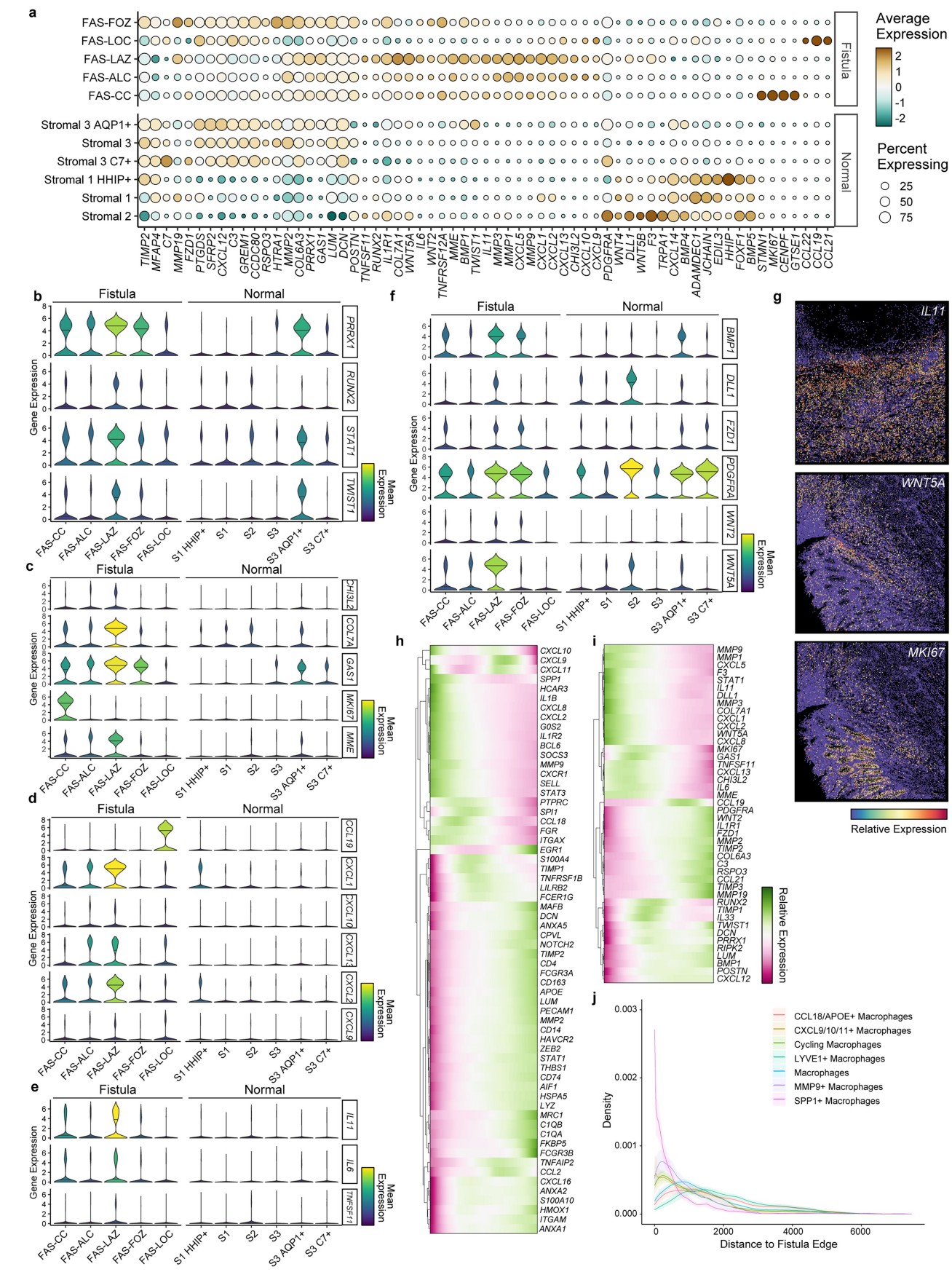

**Extended Data Fig. 5** | See next page for caption.

**Extended Data Fig. 5 | Gene expression programs in FAS cells.** Related to Fig. 1. A. Dotplot visualising fibroblast gene marker expression in Xenium ST cohort in FAS-associated fibroblast cell clusters ("Fistula" group) and normal appearing fibroblasts ("Normal"). B. Violin plots visualising selected TFs specific to FAS subtypes in Xenium ST cohort. C. Violin plots visualising selected ECM factors specific to FAS subtypes in Xenium ST cohort. D. Violin plots visualising selected chemokines specific to FAS subtypes in Xenium ST cohort. E. Violin plots visualising selected cytokines specific to FAS subtypes in Xenium ST cohort. F. Violin plots visualising selected morphogens specific to FAS subtypes in Xenium ST cohort. G. Expression of selected genes in fistula tracts in Xenium ST cohort, visualised over ROIs indicated in Fig. 1c (top panel) and Fig. 1d (middle and bottom panel). H. Heatmap visualising macrophage expressed gene variation over distance from fistula luminal edge into deeper tissue fibrotic zones, from left to right. Smooth expression fit from macrophages across all fistulae are shown. I. As in H, except FAS cell expressed genes are shown. J. Density plot visualising macrophage cell subtype distribution over distance from fistula edge. Standard error represents variation across sections and is shown in a lighter colour shade per cell type.

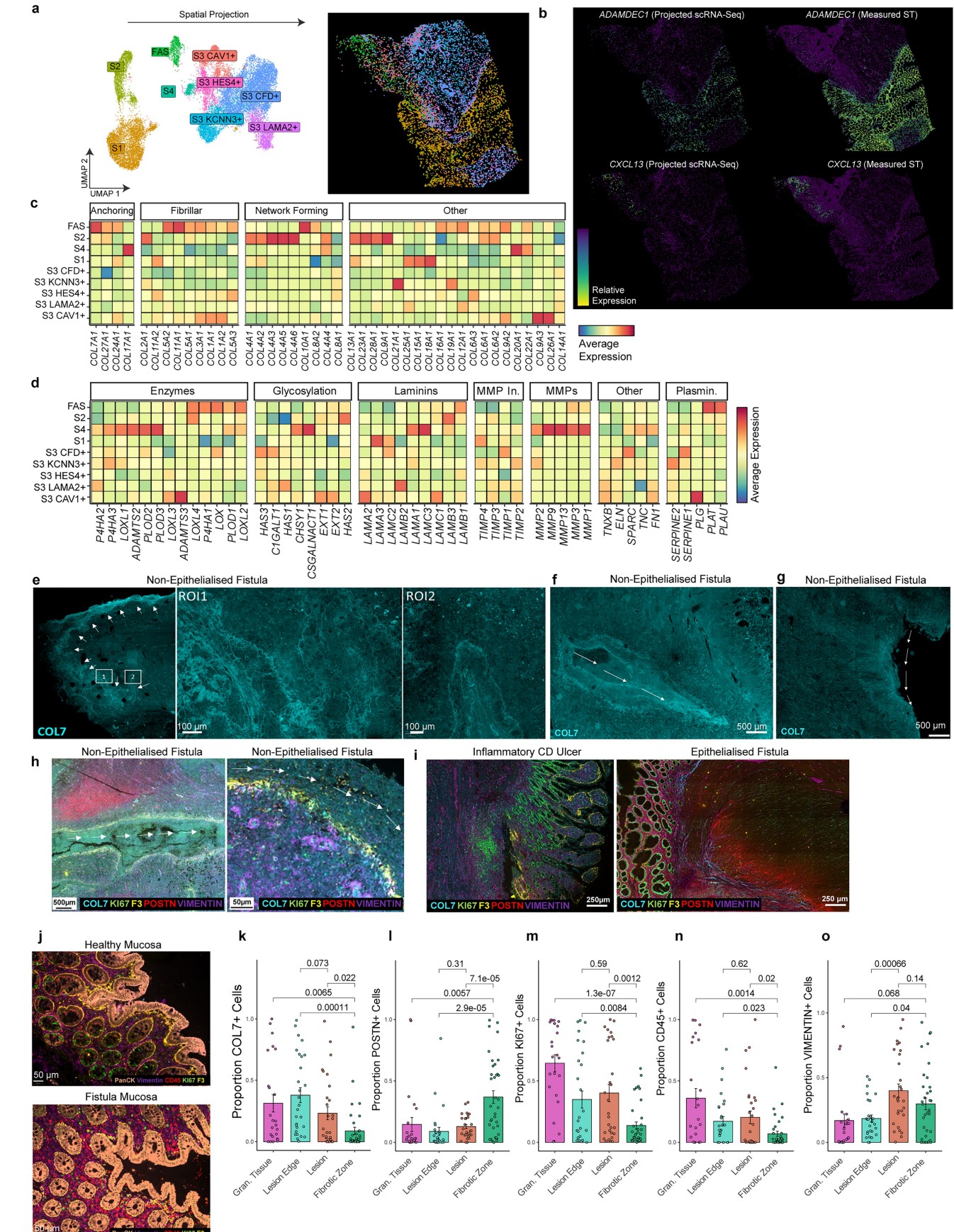

**Extended Data Fig. 6** | See next page for caption.

**Extended Data Fig. 6 | Spatial projection and protein expression analysis of FAS cells.** Related to Fig. 1. A. Spatial projection of scRNA-Seq fibroblast data (n = 22 donors), onto a representative partially epithelialised CD fistula tissue section from Xenium ST cohort (n = 53 samples). Projected cells are coloured by their scRNA-Seq cell clusters, as indicated in the UMAP (left) embedding. B. Selected fibroblast genes *ADAMDEC1* (Stromal 1) and *CXCL13* (FAS/Stromal 4) are shown in projected, scRNA-Seq data and as measured on the original Xenium ST slide. C. Heatmap visualising expression of collagen family genes across all scRNA-Seq fibroblast subclusters, highlighting the unique profiles of FAS fibroblast cells subclusters projecting to the fistula lesion edge. D. As in (C), except other extracellular matrix related factors are visualised. E. COL7 IF staining of a non-epithelialised fistula tract, with selected zoomed in ROIs visualising fistula tract edges. White arrows indicate the fistula tract. n = 16 total samples imaged. F. As in E, except non-epithelialised fistula tract is shown from an adjacent tissue section to that profiled with Xenium ST shown in Fig. 1c. White arrows indicate the fistula tract. n = 16 total samples imaged. G. As in E, a separate fistula tract sample is shown. White arrows indicate the fistula tract. n = 16 total samples imaged. H. Multiplex IF staining of two representative non-epithelialised fistula tracts, visualising COL7, KI67, POSTN, VIMENTIN and F3 protein expression. White arrows indicate the fistula tract. n = 16 total samples imaged. I. As in H, except inflammatory CD ulcer edge and non-epithelialised CD fistula tissue sections are visualised. n = 16 total samples imaged. J. Multiplex IF staining of healthy control mucosa (top) region and mucosa from a fistula CD sample (bottom). PanCK, Vimentin, CD45, KI67 and F3 channels are visualised. n = 16 total samples imaged. K. Quantification of COL7+ cells in fistula tracts imaged with multiplex IF. Artefact-free FOVs (n = 227 FOVs, n = 16 samples) from granulation tissue, lesion edge, lesion and fibrotic zone were quatified. Data are shown as mean values per group, with bars representing standard error of the mean. P values were calculated using unpaired two-sided Student's t-tests. L. Quantification of POSTN+ cells in fistula tracts imaged with multiplex IF. Artefact-free FOVs (n = 227 FOVs, n = 16 samples) from granulation tissue, lesion edge, lesion and fibrotic zone were quatified. P values were calculated using unpaired two-sided Student's t-tests. M. Quantification of KI67+ cells in fistula tracts imaged with multiplex IF. Artefact-free FOVs (n = 227 FOVs, n = 16 samples) from granulation tissue, lesion edge, lesion and fibrotic zone were quatified. P values were calculated using unpaired two-sided Student's t-tests. N. Quantification of CD45+ cells in fistula tracts imaged with multiplex IF. Artefact-free FOVs (n = 227 FOVs, n = 16 samples) from granulation tissue, lesion edge, lesion and fibrotic zone were quatified. P values were calculated using unpaired two-sided Student's t-tests. O. Quantification of VIMENTIN+ cells in fistula tracts imaged with multiplex IF. Artefact-free FOVs (n = 227 FOVs, n = 16 samples) from granulation tissue, lesion edge, lesion and fibrotic zone were quatified. P values were calculated using unpaired two-sided Student's t-tests.

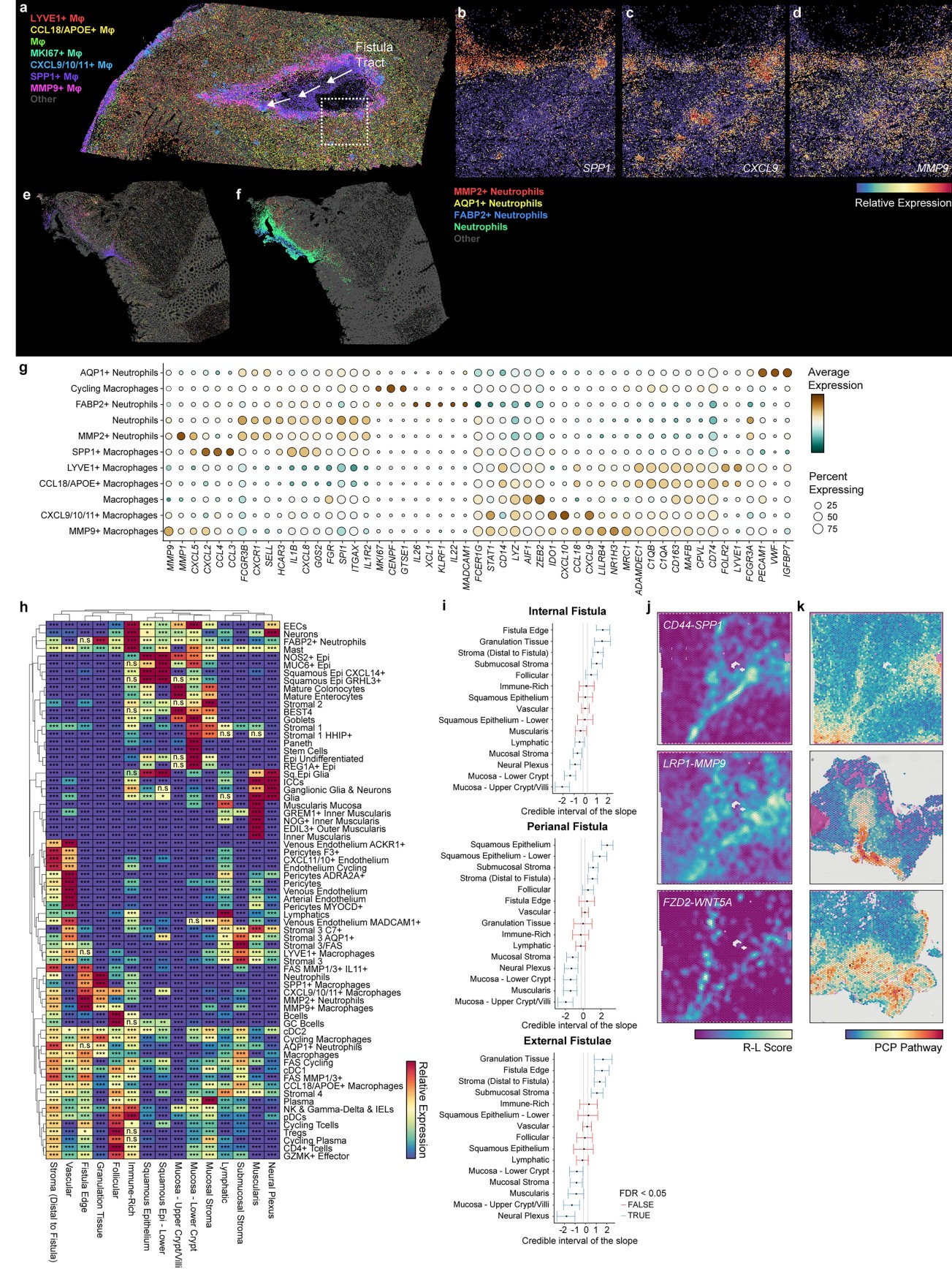

**Extended Data Fig. 7** | See next page for caption.

**Extended Data Fig. 7 | Immune-stromal niches in Crohn's fistulae.** Related to Fig. 1. A. A non-epithelialised fistula tissue section in Xenium ST cohort (n = 53 samples) visualising spatial distribution of macrophage cell subtypes. All other cells are shown in grey. A selected ROI zoom at the edge of the fistula tract is indicated in a white box. B. Spatial gene expression of macrophage marker *SPP1* visualised in ROI indicated in panel A. C. Spatial gene expression of macrophage marker *CXLC9* visualised in ROI indicated in panel A. D. Spatial gene expression of macrophage marker *MMP9* visualised in ROI indicated in panel A. E. Partially epithelialised fistula tissue section in Xenium cohort, visualising spatial distribution of macrophages partially overlapping the epithelialised part of the fistula lesion, with all other cells shown in grey. F. As in A, except neutrophils are shown. G. Dotplot visualising cluster marker gene expression of neutrophil and macrophage subcluster genes in Xenium ST cohort. H. Heatmap visualising cell type enrichment across all detected tissue niches in Xenium ST cohort. Cell type enrichment within a niche is indicated by red, positive values, while cell type depletion within a niche is indicated in blue. Enrichment/depletion significance FDR is indicated as * <0.05, ** <0.01, *** <0.001, n.s. > 0.05. I. Differential abundance plot comparing all identified tissue niches between internal fistulae and health, external fistulae and health and perianal fistulae and health (n = 53 samples). Bands represent the 95% Bayesian credible interval of the slope (logit fold change in niche proportion per unit change in the covariate), indicating the range of effect sizes compatible with the data, given the model. J. Spatial distribution of receptor-ligand product score for *CD44-SPP1*, *LRP1-MMP9* and *FZD2-WNT5A* interactions in a representative (n = 43 samples) Visium ST tissue section of a non-epithelialised fistula tract. Paired adjacent Xenium tissue section is visualised in panel A. K. PCP Reactome Pathway activity score spot overlay over representative (n = 34 samples) non-epitheliased fistula section (top) and epithelialised fistula sections (middle, bottom).

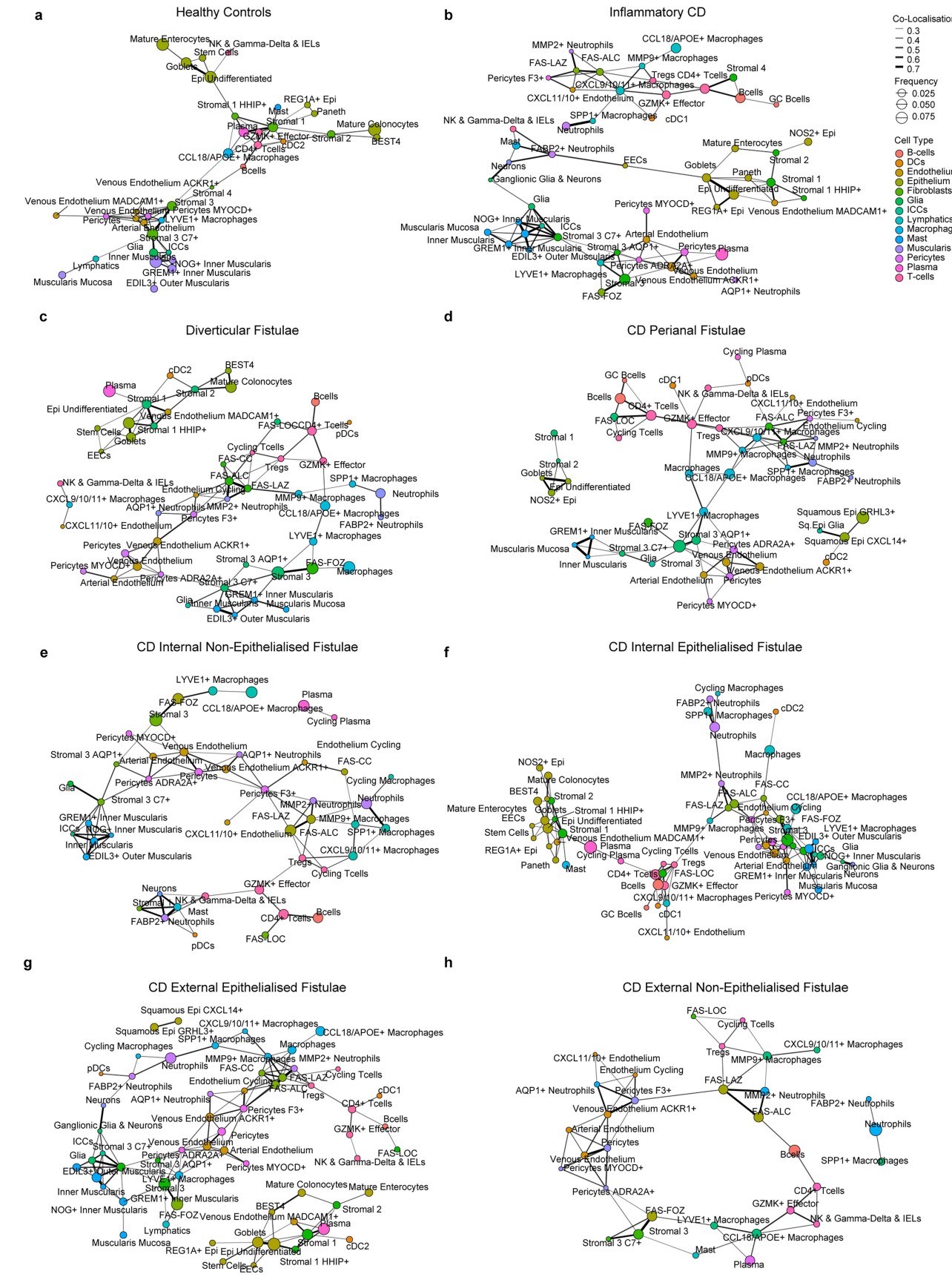

**Extended Data Fig. 8** | See next page for caption.

**Extended Data Fig. 8 | Cellular co-localisation in Crohn's fistulae.** Related to Fig. 2. A. Cell type co-localisation network plot visualising cell subsets identified in Xenium ST cohort (n = 53 total samples) in healthy control tissue sections. Edges represent degree of co-localisation between two cell types, node size indicates the overall abundance of the cell type, nodes are coloured by cell lineage. Graph edges with low degree of co-localisation between cell type pairs (<0.15) are omitted for clarity. B. As in A, except for inflammatory CD tissue sections. C. As in A, except for diverticular disease fistulae tissue sections. D. As in A, except for external, perianal fistulae tissue sections. E. As in A, except for internal, non-peri anal non-epithelialised fistula tissue sections. F. As in A, except for internal, epithelialised fistula tissue sections. G. As in A, except for external, epithelialised fistula tissue sections. H. As in A, except for external, non-epithelialised fistula tissue sections.

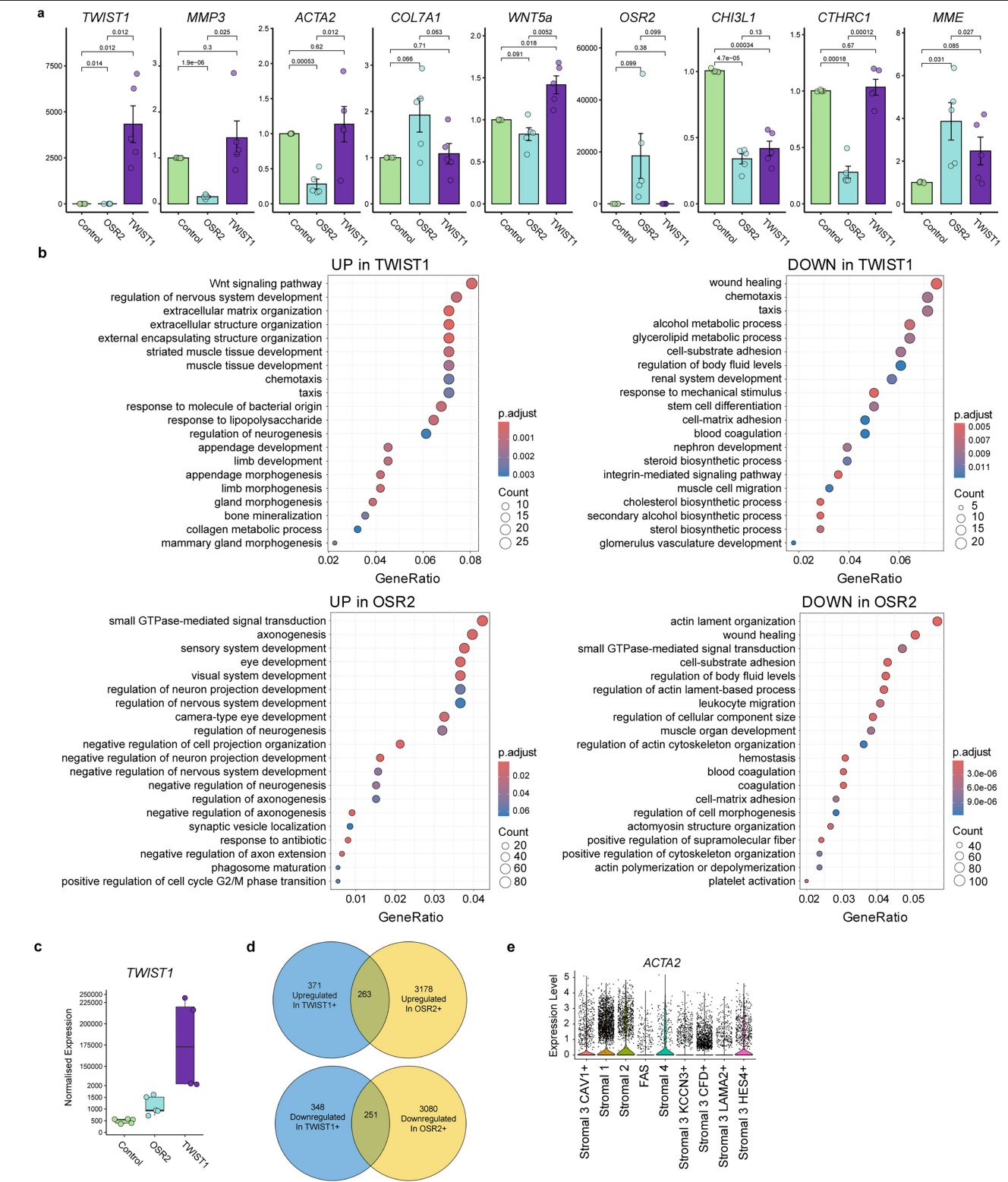

**Extended Data Fig. 9 | Biological processes regulated by OSR2 and TWIST1.**
Related to Fig. 2. A. qPCR quantification of selected FAS marker genes, as well
as TFs *TWIST1* and *OSR2* in *OSR2* and *TWIST1* overexpressing and control
intestinal fibroblasts. n = 5 donors per group. Data are presented as mean
values with error bars representing standard error of the mean. P values were
calculated using paired two-sided Student's t-tests. B. Dotplots visualising top
most enriched gene ontology (GO) biological process (BP) terms in genes
found to be significantly upregulated or downregulated in *TWIST1* or *OSR2*
overexpressing fibroblast when compared to intestinal controls. G. RNA-Seq

quantification of *TWIST1* expression, as in Fig. 2a, except split axes are shown to
visualise an increase in OSR2 overexpressing fibroblasts over controls. Boxplots
show the median (centre line), interquartile range (box), and whiskers extending
to the most extreme values within 1.5× the interquartile range; individual data
points are plotted. N = 5 donors per group. C. Venn diagrams visualising
overlapping differentially expressed genes between *OSR2* and *TWIST1* over
expressing fibroblasts. D. Violin plot visualising ACTA2 (α-SMA) gene expression
in scRNA-Seq data of fibroblasts from this study. Cells from n = 22 donors.

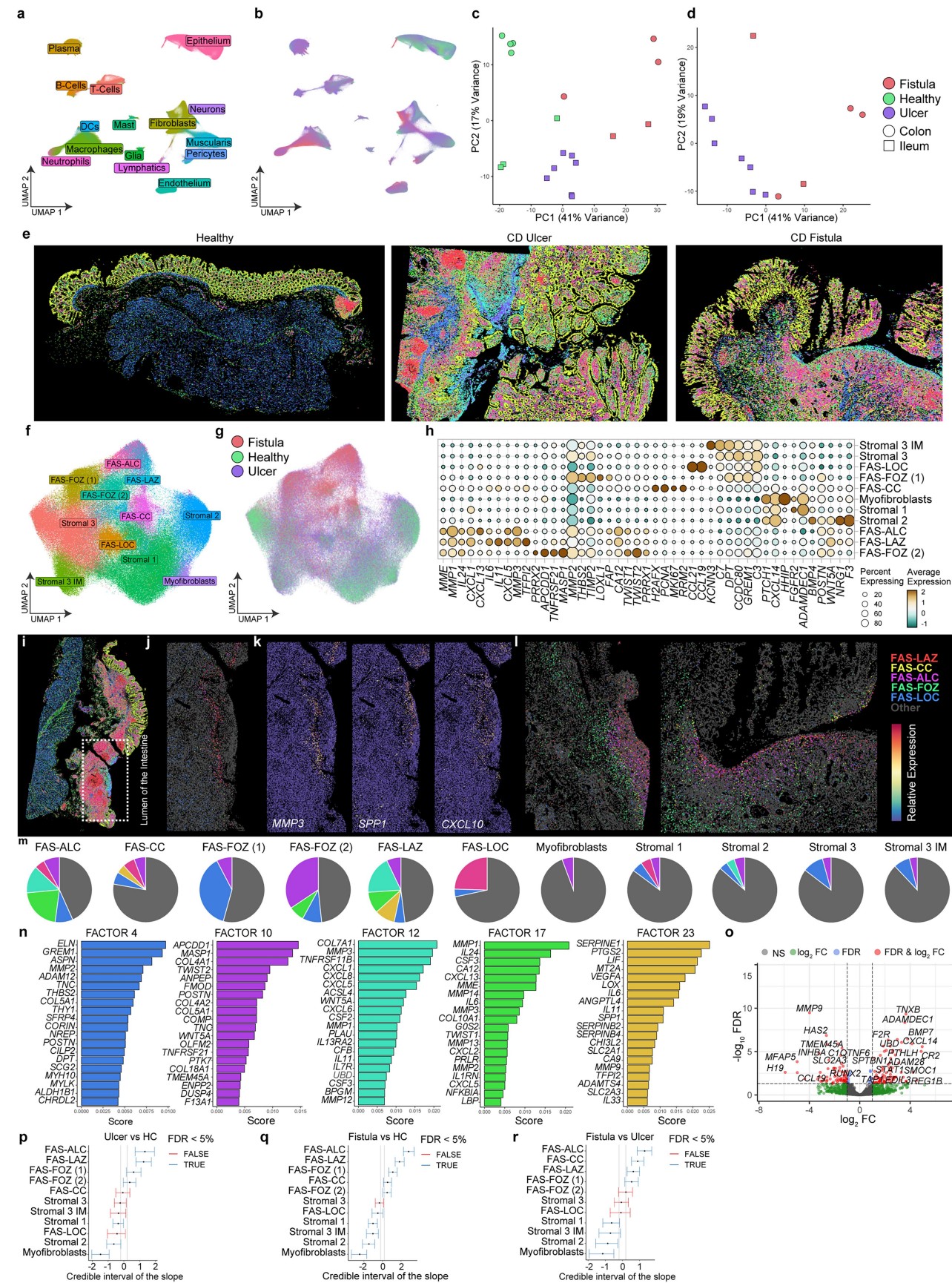

**Extended Data Fig. 10 |** See next page for caption.

**Extended Data Fig. 10 | Spatial analysis of ulcer bases in Crohn's disease.**
Related to Fig. 3. A. UMAP embedding visualising all cells from integrated
Xenium in situ 5100-plex analysis, cells are coloured by broad cell lineage. n = 19
samples, n = 1,860,368 cells. B. As in A, except cells are coloured by sample
type. Colour legend as in D. n = 19 samples, n = 1,860,368 cells. C. Sample
pseudobulk PCA analysis of all tissue sections analysed in the 5100-plex ST
cohort. Colour legend as in D. n = 19 samples. D. As in C, except pseudobulk PCA
was calculated only on cells from lesion (fistula or ulcer) areas as defined by
niche analysis (Methods). Healthy control samples were excluded from the
analysis, as no lesional areas were present. n = 12 samples. E. Representative
tissue sections (n = 19 samples) visualising broad cell type tissue distribution in
healthy control tissue (left), CD ulcer (middle) and fistula tract (right). Colour
legend as in A. F. UMAP embedding visualising all fibroblast subpopulations
detected from clustering analysis of integrated Xenium in situ 5100-plex ST
cohort, coloured by detected fibroblast subclusters. N = 19 samples. G. As in F,
except points are coloured by sample type. n = 19 samples. H. Dotplot
visualising top marker gene expression of fibroblast subtypes visualised in F.
I. A representative CD inflammatory tissue section profiled with Xenium
5100-plex ST panel (n = 19 samples). Colour legend as in A. A white box with
dashed lines indicates an ROI with ulcer. J. FAS cell type distribution around the
ROI indicated in panel i. All other cells are shown in grey. N = 19 samples in
cohort. K. Selected gene expression of FAS and macrophage cell marker genes
in the ulcer ROI indicated in panel I. n = 19 samples in cohort. L. FAS cell type
distribution around two representative fistula tracts profiled with Xenium
5100-plex analysis. Colour legend as in k, all other cells are shown in grey.
N = 19 samples in cohort. M. cNMF analysis of fibroblast cells from Xenium
5100-plex in situ ST cohort, with pie charts indicating average disease
associated factor usage per cluster. Factor colours are as in bar charts below in
panel n, with other factors shown in grey. N. Barplots visualising cNMF factor
top gene loadings for selected factors enriched in disease-associated
fibroblast clusters. O. Volcano plot showing differential expression analysis
between ulcer vs fistula FAS-ALC cells. Differential expression was assessed
using DESeq2 with Wald tests and Benjamini–Hochberg correction for multiple
testing. Testing was carried out on pseudobulk expression in n = 19 samples. P.
Differential abundance analysis of fibroblast subpopulations presented in
panel F, comparing cells from CD ulcer samples with healthy controls (n = 19
samples in cohort). Bands represent the 95% Bayesian credible interval of the
slope (logit fold change in cluster proportion per unit change in the covariate),
indicating the range of effect sizes compatible with the data, given the model.
Q. Differential abundance analysis of fibroblast subpopulations presented in
panel F, comparing cells from CD fistula samples with healthy controls (n = 19
samples in cohort). Bands represent the 95% Bayesian credible interval of the
slope (logit fold change in cluster proportion per unit change in the covariate),
indicating the range of effect sizes compatible with the data, given the model.
R. Differential abundance analysis of fibroblast subpopulations presented
in panel F, comparing cells from CD fistula samples with CD ulcer samples
(n = 19 samples in cohort). Bands represent the 95% Bayesian credible interval
of the slope (logit fold change in cluster proportion per unit change in the
covariate), indicating the range of effect sizes compatible with the data, given
the model.

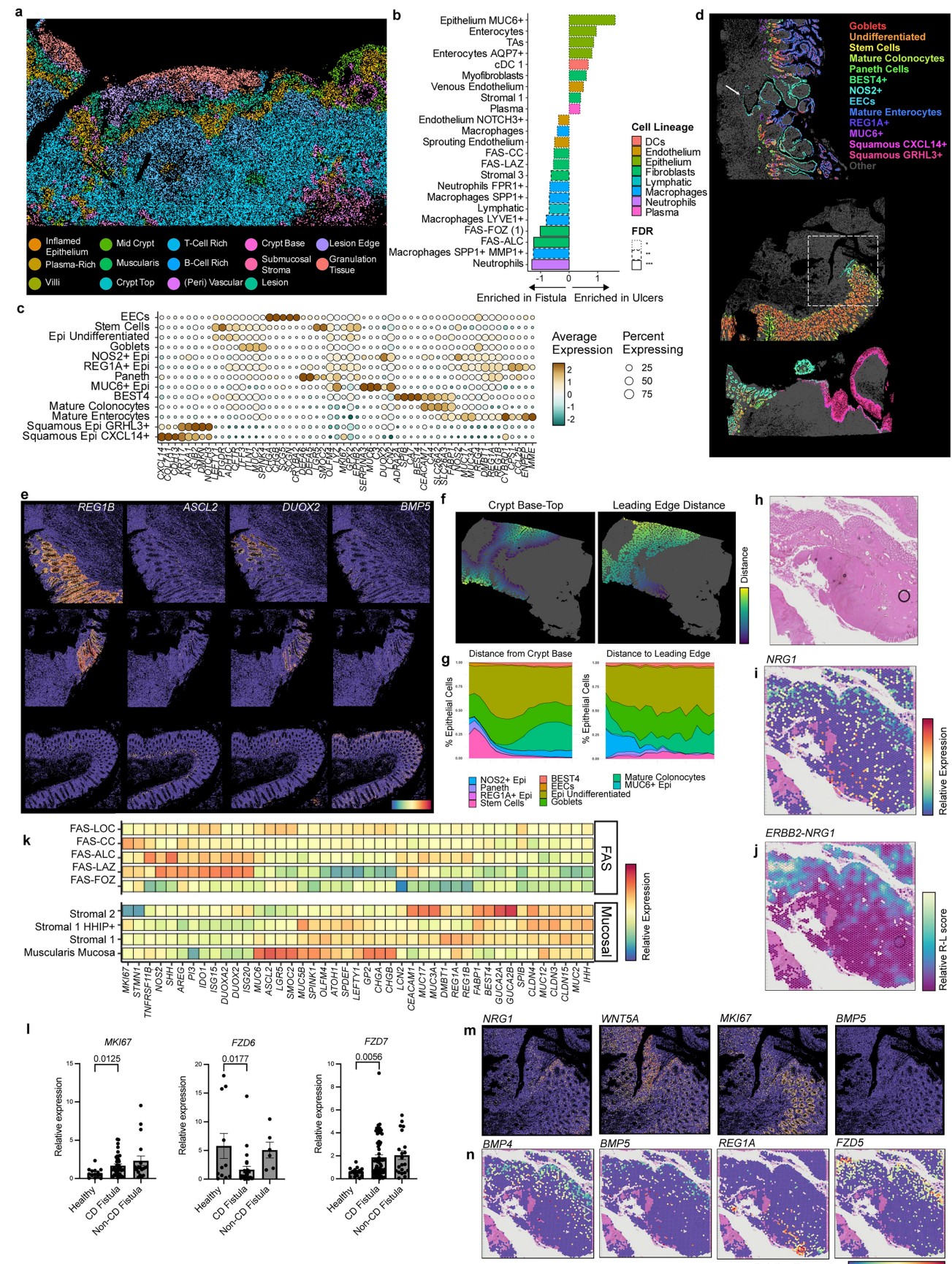

 to Figs. 3 and 4. A. Spatial layering of tissue niches identified by niche analysis of 5100-plex Xenium ST dataset. A representative region (n = 19 samples in cohort) around inflammatory CD ulcer is visualised, the ROI from the full section visualised in Extended Data Fig. 9i. B. Differential cell type abundance between CD ulcers and fistulae within more distal and fibrotic outer zone niches around fistula/ulcer lesions. Bars represent the 95% Bayesian credible interval of the slope (logit fold change in cluster proportion per unit change in the covariate), indicating the range of effect sizes compatible with the data, given the model. Dashed lines indicate FDR. Only significant cell type enrichments (FDR < 0.05) are shown. N = 19 samples in cohort. C. Dotplot visualising epithelial cell subtype markers identified in Xenium ST cohort. D. Representative tissue sections visualising spatial distribution of epithelial cell types detected in Xenium ST cohort (n = 53 samples in cohort). All non-epithelial cell types are shown in grey. An early fistula/deep fissuring ulcer is visualised in the top panel, a partially epithelialised, columnar epithelium fistula tract visualised in the middle panel, and a representative tract containing both squamous and columnar epithelium from a peri-anal fistula is visualised at the bottom. E. Gene expression of selected marker genes visualised in ROIs indicated in Fig. 4a. F. A representative partially epithelialised fistula section visualising distances of epithelial cells from crypt base to crypt top (left) and from the leading edge of epithelialisation of the fistula tract into lagging mucosa. Epithelial cells are coloured, all other cells indicated in grey. N = 53 samples in cohort, n = 6 fistulae with partially epithelialised edge quantified. G. Epithelial cell type proportion distribution over distance with respect to crypt axis gradient (left) and distance from the leading edge of the epithelialisation of the fistula tract (right) as indicated in panel F. N = 53 samples in cohort, n = 6 fistulae with partially epithelialised edge

quantified. H. Reference H&E image of a partially epithelialised fistula tract profiled with Visium ST (n = 34 samples). I. Spatial distribution of NRG1 gene expression in fistula tissue section (n = 34 samples), with reference H&E image in H. J. Spatial distribution of receptor-ligand product score for *NRG1-ERBB2* interaction, with reference H&E image in H. n = 34 samples. K. Heatmap visualising gene expression variation in epithelial cells conditional on which FAS or other mucosal fibroblast cell types are in the local spatial neighbourhood. Aggregated and scaled mean expression is shown. For example, epithelial cells closer to muscularis mucosa express stem cell marker genes such as *LGR5*, while epithelial cells nearby FAS cells upregulate innate immune response genes e.g. *ISG15*. L. Relative gene expression of frizzled receptors *FZD6*, *FZD7* and the proliferation marker *MKI67* as assessed by qPCR in the validation cohort (n = 84). Gene expression data are presented as mean values with error bars representing the SEM. Statistical analysis was performed using an unpaired t-test, with significance set at p < 0.05. Expression levels of *FZD6* and *FZD7* were adjusted for *EPCAM* expression using a generalised linear model. Compared to healthy controls, *FZD6* expression was significantly reduced in internal CD fistula (fold change = −1.82, p = 0.0048), perianal CD fistula (fold change = −1.60, p = 0.04), and diverticular fistula (fold change = −1.79, p = 0.02). In contrast, FZD7 expression was significantly increased in internal CD fistula (fold change = 0.83, p = 0.01), external CD fistula (fold change = 1.23, p = 0.03), and diverticular fistula (fold change = 1.02, p = 0.01). M. Selected gene marker expression measured in Xenium ST cohort (n = 53) visualised at the edge of partially epithelialised fistula edge, with ROI indicated by white box in panel D. N. Selected marker gene expression measured in Visium ST cohort (n = 34) visualised in partially epithelialised fistula, with reference H&E image visualised in panel H.

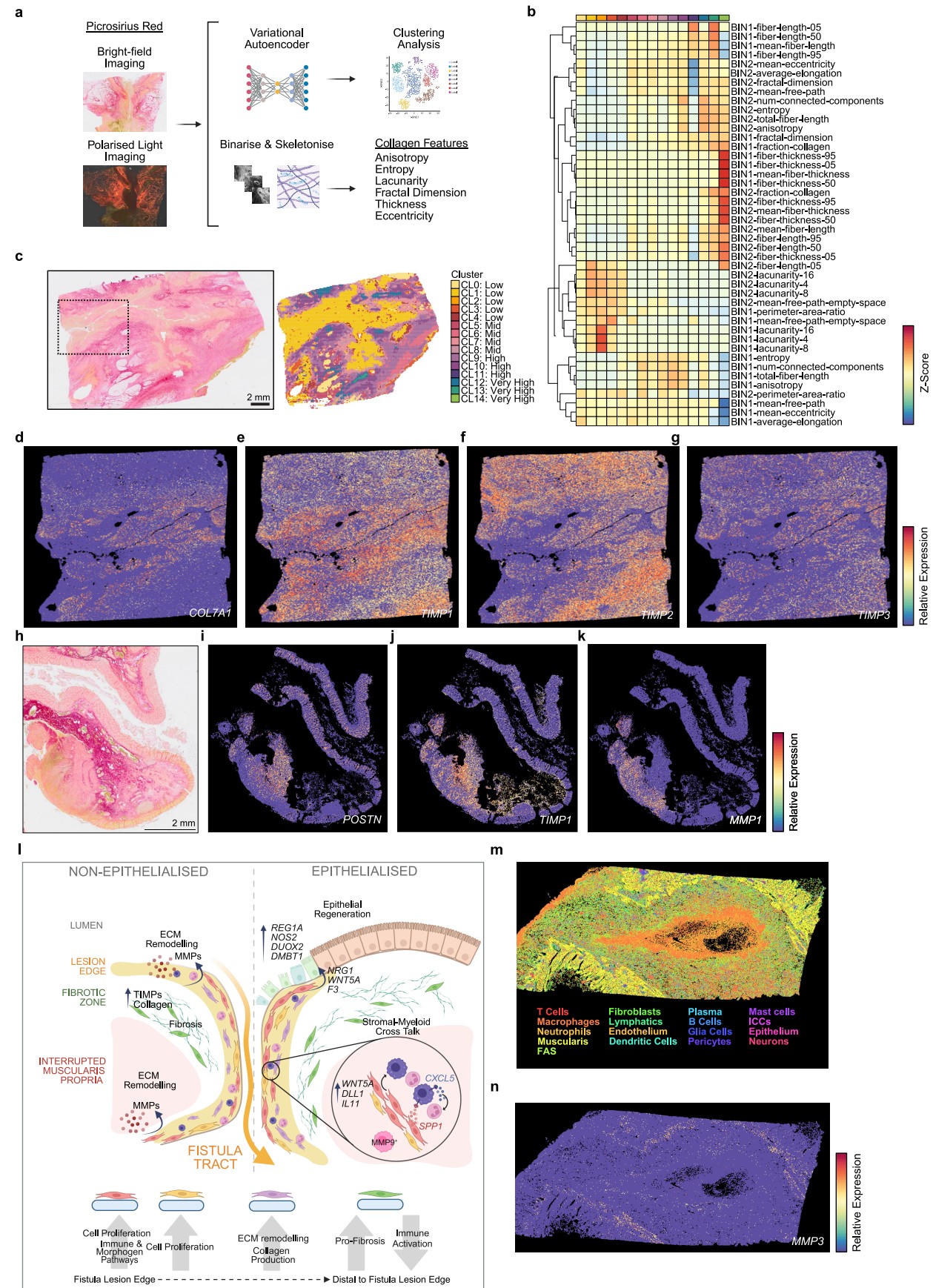

**Extended Data Fig. 12** | See next page for caption.

**Extended Data Fig. 12 | ECM analysis of Crohn's fistula tracts.** Related to Fig. 5. A. A schematic overview of the collagen image analysis approach. Briefly, adjacent cut tissue sections paired with Visium and/or Xenium ST data were stained with picrosirius red and imaged with brightfield imaging and under polarised light. Images were patched into equal-sized patches 250 μm in size, used to train a variational autoencoder and the latent space was used to cluster image patches. In parallel, each image patch was also red-channel binarized at two different thresholds and skeletonised, and for each patch geometric collagen features (e.g. anisotropy) were calculated and mapped to the image-based clustering. Created in BioRender. Group, S. (2025) https://BioRender. com/ytis8zj. B. Heatmap visualising all calculated collagen features across all detected image clusters as described in A. Mean, row-scaled values per cluster are visualised. N = 27 samples. C. Representative tissue section from a non-epithelialised CD fistula stained with picrosirius red (n = 27) (left) and corresponding spatial distribution of detected image clusters as indicated in B. A dashed line box represents an adjacent tissue section region profiled with Xenium ST (n = 53). D. Spatial distribution of gene expression of FAS cell expressed *COL7A1* in a representative non-epithelialised fistula tissue section from ROI indicated in panel C. n = 53 samples in cohort. E. Spatial distribution of gene expression of FAS cell expressed *TIMP1* in a representative non-epithelialised fistula tissue section from ROI indicated in panel C. n = 53 samples in cohort. F. Spatial distribution of gene expression of FAS cell expressed *TIMP2* in a representative non-epithelialised fistula tissue section from ROI indicated in panel C. n = 53 samples in cohort. G. Spatial distribution of gene expression of FAS cell expressed *TIMP3* in a representative non-epithelialised fistula tissue section from ROI indicated in panel C. n = 53 samples in cohort. H. A representative image of a partially epithelialised fistula section stained with picrosirius red (n = 27). I. Spatial distribution of gene expression of FAS cell expressed *POSTN* gene profiled with Xenium ST (n = 53 samples in cohort), with corresponding adjacent picrosirius red section shown in panel H. J. Spatial distribution of gene expression of FAS cell expressed *TIMP1* gene profiled with Xenium ST (n = 53 samples in cohort), with corresponding adjacent picrosirius red section shown in panel H. K. Spatial distribution of gene expression of FAS cell expressed *MMP1* gene profiled with Xenium ST n = 53 samples in cohort), with corresponding adjacent picrosirius red section shown in panel H. L. A schematic overview of key findings presented in this manuscript. Created in BioRender. Group, S. (2025) https://BioRender.com/ ekjjfu1. M. A selected non-epithelialised fistula tissue section profiled by Xenium ST cohort (n = 53) capturing the fistula tract penetrating through muscularis propria layers. Each point represents cell centroids, with cells coloured by broad lineage. Muscle cells are shown in bright yellow, along the edges of the fistula tract. N. Gene expression overlay visualising expression of *MMP3* along the muscularis propria boundaries in a non-epithelialised fistula tract shown in panel M. n = 53 in cohort.

# Reporting Summary

## Statistics

For all statistical analyses, confirm that the following items are present in the figure legend, table legend, main text, or Methods section.

| n/a | Confirmed | |
|---|---|---|
| ☐ | ☒ | The exact sample size (*n*) for each experimental group/condition, given as a discrete number and unit of measurement |
| ☐ | ☒ | A statement on whether measurements were taken from distinct samples or whether the same sample was measured repeatedly |
| ☐ | ☒ | The statistical test(s) used AND whether they are one- or two-sided |
| | | *Only common tests should be described solely by name; describe more complex techniques in the Methods section.* |
| ☐ | ☒ | A description of all covariates tested |
| ☐ | ☒ | A description of any assumptions or corrections, such as tests of normality and adjustment for multiple comparisons |
| ☐ | ☒ | A full description of the statistical parameters including central tendency (e.g. means) or other basic estimates (e.g. regression coefficient) AND variation (e.g. standard deviation) or associated estimates of uncertainty (e.g. confidence intervals) |
| ☐ | ☒ | For null hypothesis testing, the test statistic (e.g. *F*, *t*, *r*) with confidence intervals, effect sizes, degrees of freedom and *P* value noted |
| | | *Give P values as exact values whenever suitable.* |
| ☒ | ☐ | For Bayesian analysis, information on the choice of priors and Markov chain Monte Carlo settings |
| ☒ | ☐ | For hierarchical and complex designs, identification of the appropriate level for tests and full reporting of outcomes |
| ☐ | ☒ | Estimates of effect sizes (e.g. Cohen's *d*, Pearson's *r*), indicating how they were calculated |

*Our web collection on statistics for biologists contains articles on many of the points above.*

## Software and code

Policy information about availability of computer code

| Data collection | BD FACS Diva Software v9.0 BD Biosciences<br>Zeiss Axioscanner (Software: Zen Blue Edition Version 3.3.89.0000) |
|---|---|
| Data analysis | FACS: FlowJo v10.8.1 FlowJo.com<br>PCR, Image data: Graphpad Prism v9.5.1 (528) www.graphpad.com<br>Qupath software (version 0.5.0)<br><br>Illumina bcl2fastq (version 2.20.0.422)<br>FastQC software (version 0.11.9)<br>10x Genomics Cellranger software (version 7.1.0)<br>10X Genomics Spaceranger software (version 2.1.0)<br>cNMF pipeline (version 1.7.0)<br>Cutadapt software (version 4.4)<br>STAR aligner software (version 2.7.11a)<br>subread featureCounts software (version 2.0.6)<br>multiQC software (version 1.14)<br>Picard tools (version 3.2.0)<br>pySCENIC pipeline (version 0.12.2b0)<br>Baysor algorithm (0.6.2) |

R package DropletUtils (version 1.24)
R package scCustomize (version 2.1.2)
R package Seurat (version 5.1.0)
R package Harmony (version 1.2.1)
R package clustree (version 0.5.1)
R package DESeq2 (version 1.44.0)
R package miloR (version 2.0.0)
R package sccomp (version 1.9.5)
R package ggpraph (version 2.2.1)
R package clusterProfiler (version 4.12.6)
R package DBI  (version1.66.0)
R package CARD (version 1.1)
R package scCustomize (version 2.1.2)
R package MAST (version 1.33.0)
R package AUCell(version 1.30.1)

Python (version 3.9)
Python package skimage (version 0.20.0)
Python package numpy (version 1.26.4)
Python package pandas (version 2.1.1)
Python package scikit-learn (version 1.5.2)
Python package matplotlib (version 3.7.1)
Python package seaborn (version 0.13.2)

Code used in the data analysis has been deposited on github https://github.com/agneantanaviciute/cdfistulaspatialtranscriptomics and https://github.com/ChloeHJ/spamsc.

For manuscripts utilizing custom algorithms or software that are central to the research but not yet described in published literature, software must be made available to editors and reviewers. We strongly encourage code deposition in a community repository (e.g. GitHub). See the Nature Portfolio guidelines for submitting code & software for further information.

# Data

Policy information about availability of data

All manuscripts must include a data availability statement. This statement should provide the following information, where applicable:
- Accession codes, unique identifiers, or web links for publicly available datasets
- A description of any restrictions on data availability
- For clinical datasets or third party data, please ensure that the statement adheres to our policy

All raw sequencing data has been deposited on GEO (series GSE284232), accessions GSE283945(Visium ST), GSE305631(bulk RNA-Seq) and GSE284230 (scRNA-Seq). All processed scRNA-Seq, Visium ST and Xenium ST Seurat data RDS object files, image data and scRNAseq and spatial transcriptomics data tables have been deposited as Mendeley Data: doi: 10.17632/tn972brm9s.1 (https://data.mendeley.com/datasets/tn972brm9s/1), doi: 10.17632/mxy6p6wfmy.1 (https://data.mendeley.com/preview/mxy6p6wfmy) and doi: 10.17632/64fkdfcpzb.1 (https://data.mendeley.com/datasets/64fkdfcpzb/1).

Data for scRNA-Seq meta analysis was obtained from GEO: GSE189185, GSE114374, GSE134809, GSE260842, GSE260833, GSE266546; ArrayExpress: E-MTAB-8901; Broad Single Cell Portal SCP259, SCP1884; and Zenodo https://doi.org/10.5281/zenodo.13768607.

Additional scRNA-Seq reference data for cell type deconvolution was downloaded from GEO: GSE201153 and GSE163668.

A browsable spatial data portal web app is available at  https://simmonslab.shinyapps.io/cd-fistula-data-portal/

# Research involving human participants, their data, or biological material

Policy information about studies with human participants or human data. See also policy information about sex, gender (identity/presentation), and sexual orientation and race, ethnicity and racism.

| | |
|---|---|
| Reporting on sex and gender | Samples were balanced across biological sex (as reported clinically) as part of the study design, where possible - scRNAseq (14 F, 19M), Visium ST (15 F, 19M), Xenium ST (28 F, 25M), biomark (36F, 48M). |
| Reporting on race, ethnicity, or other socially relevant groupings | Not applicable, no data on race and ethnicity was collected. |
| Population characteristics | Human ileal and colonic tissue was collected from consenting adult. A detailed sample overview with sex, age and other clinical characteristics is provided in Supplementary Table 1. |
| Recruitment | Patients were recruited to studies in a randomized fashion depending on their presentation to hospital as part of their routine clinical care. Recruitment and collection was by multiple investigators, at multiple sites, over a period of many years, with no obvious biases that could be identified. |
| Ethics oversight | All samples were collected with informed consent as per the principles of Helsinki under the aegis of studies and procedures |

| Ethics oversight | approved by different NHS Research Ethics Committees (RECs). The relevant registered approvals are as follows: REC reference(s): TIP Study: 18/WM/0237 GI Biobank: 16/YH/0247 IBD Biobank: 09/H1204/30 FFPE Tissue samples (OCHRe): 19/SC/0173 FFPE Tissue Samples (Bayreuth): Friedrich Alexander University, Bayreuth, Germany, Ethics number: 23-131 bp |

Note that full information on the approval of the study protocol must also be provided in the manuscript.

# Field-specific reporting

Please select the one below that is the best fit for your research. If you are not sure, read the appropriate sections before making your selection.

☒ Life sciences  ☐ Behavioural & social sciences  ☐ Ecological, evolutionary & environmental sciences

For a reference copy of the document with all sections, see nature.com/documents/nr-reporting-summary-flat.pdf

# Life sciences study design

All studies must disclose on these points even when the disclosure is negative.

| Sample size | No formal sample size calculations were performed for spatial and single cell analyses. Instead, we analyzed all available patient samples meeting inclusion criteria resulting in a large cohort. This sample size exceeds many prior studies of intestinal tissue using these methods and provides sufficient statistical power to detect robust cell state differences and spatial patterns. For bulkRNA-Seq, we used n=5 biological replicates per group, which is in line with accepted practice for bulk RNA-Seq discovery studies, providing sufficient power to detect consistent DEGs while balancing feasibility and resources. |
| Data exclusions | No data were excluded from analysis. |
| Replication | All experiments were performed in triplicates or more in order to ensure reliability and reproducibility of results. Exact quantification for each replicate is described in methods and figure legends. |
| Randomization | Patients attending hospital for their planned care were consented in a random fashion to collect research samples, and classified into groups - healthy or disease - based on histopathological analysis blinded to the research objective. For TWIST1 and OSR2 overexpression experiments, bulk RNA seq, qPCR and imaging, a paired design was used rather than randomised, where samples from the same donor were used across all experimental groups. All statistical analyses of these data were performed in a paired fashion, accounting for donor variability as a covariate in the statistical testing. |
| Blinding | Blinding was not necessary as experimental read-outs were automated (FACS, qPCR, scRNA-Seq, spatial transcriptomics, bulkRNAseq). Histopathology annotations were blinded. |

# Reporting for specific materials, systems and methods

We require information from authors about some types of materials, experimental systems and methods used in many studies. Here, indicate whether each material, system or method listed is relevant to your study. If you are not sure if a list item applies to your research, read the appropriate section before selecting a response.

## Materials & experimental systems

| n/a | Involved in the study |
|---|---|
| ☐ | ☒ Antibodies |
| ☐ | ☒ Eukaryotic cell lines |
| ☒ | ☐ Palaeontology and archaeology |
| ☒ | ☐ Animals and other organisms |
| ☐ | ☒ Clinical data |
| ☒ | ☐ Dual use research of concern |
| ☒ | ☐ Plants |

## Methods

| n/a | Involved in the study |
|---|---|
| ☒ | ☐ ChIP-seq |
| ☐ | ☒ Flow cytometry |
| ☒ | ☐ MRI-based neuroimaging |

# Antibodies

| Antibodies used | Immunohistochemistry: TWIST1 E5G9Y Rabbit IgG RRID:AB_3064916 (pH 6, 1:800) Anti-F3 AMAb91235 Mouse IgG RRID:AB_2665858 (pH 6, 1:250) CD45 30-F11 Rat IgG2b, kappa RRID:AB_467251 (pH 6, 1:50)  Immunoflourescence: Neutrophil Elastase NIMP-R14 Rat IgG2b 1:100 RRID:AB_303154 |

F3 (C0142) CL3905 Mouse IgG1 1:100 RRID:AB_2665858
WNT5A MAB645 Rat IgG2A 1:200 RRID:AB_10571221
Cleaved Caspase 3, 5A1E, Rabbit IgG 1:100 RRID:AB_2928048
CD45, RM1007, Rabbit, IgG, 1:100 RRID:AB_3678516
Periostin, EPR20806, Rabbit, IgG, 10ug/ml RRID:AB_2924310
NaKATPase, Rabbit, IgG, 10ug/ml RRID:AB_3451994
Collagen 7, Abcam: ab198899, Rabbit, IgG, 10ug/ml
Vimentin, EPR3776, Rabbit, IgG, 5 ug/ml RRID:AB_2909595
PRRX2, HPA026808, Rabbit, IgG, 1:100 RRID:AB_10603228
OSR2,HPA052425, Rabbit, IgG, 1:100 RRID:AB_2681826
TWIST1, E5G9Y, Rabbit, IgG, 1:100 RRID:AB_3064916
RUNX2, EPR14334, Rabbit, IgG, 1:100 RRID:AB_2889254
KI67, SP6 Rabbit IgG 1:100 RRID: AB_302459
Pan-Cytokeratin PCK-26 Mouse IgG1 1:100 RRID: AB_305450
Pan-Cadherin EPR1792Y IgG  10 ug/ml RRID: AB_3106921

FACS:
Brilliant Violet 785'" anti-human CD326 (EpCAM) Antibody, 9C4 clone BioLegend Cat# 324237 RRID:AB_2632936 (1:50)
FITC anti-human CD45 Antibody, H130 clone BioLegend Cat# 304005 RRID:AB_314393 (1:100)
Human Trustain FcX (Fc Block) Biolegend Cat# 422302, RRID:AB_2818986 (TruStain FcX was added at 5 µl per 1×10⁶ cells in 100 µl staining volume)
DAPI (4'6-diamidino-2- phenylindole) Solution, BD Pharminogen, Cat#564907 (1:1000)

CITE-seq:
TotalSeqC anti-human CD4 Antibody BioLegend Cat# 344651 RRID:AB_2800921 (1:100)
TotalSeqC  anti-human CD8 Antibody BioLegend Cat# 344753 RRID:AB_2800922(1:100)
TotalSeqC  anti-human CD56 (NCAM) Antibody BioLegend Cat# 362559 RRID:AB_2801002 (1:100)
TotalSeqC  anti-human CD3 Antibody BioLegend Cat# 344849 RRID:AB_2814272 (1:100)
TotalSeqC  anti-human CD45RA Antibody BioLegend Cat# 304163 RRID:AB_2800764 (1:100)
TotalSeqC anti-human CD45RO Antibody BioLegend Cat# 304259 RRID:AB_2800766 (1:100)
TotalSeqC anti-human CD279 (PD-1) Antibody BioLegend Cat# 329963 RRID:AB_2800862 (1:100)
TotalSeqC anti-human CD103 (Integrin aE) Antibody BioLegend Cat# 350233 RRID:AB_2800933 (1:100)
TotalSeqC anti-human Hashtag 1 Antibody BioLegend Cat# 394661 RRID:AB_2801031 (0.75µl per 1x10⁶ cells per 100µl staining volume)
TotalSeqC anti-human Hashtag 2 Antibody BioLegend Cat# 394663 RRID:AB_2801032 (0.75µl per 1x10⁶ cells per 100µl staining volume)
TotalSeqC anti-human Hashtag 3 Antibody BioLegend Cat# 394665 RRID:AB_2801033 (0.75µl per 1x10⁶ cells per 100µl staining volume)
TotalSeqC anti-human Hashtag 4 Antibody BioLegend Cat# 394667 RRID:AB_2801034 (0.75µl per 1x10⁶ cells per 100µl staining volume)
TotalSeqC anti-human Hashtag 5 Antibody BioLegend Cat# 394669 RRID:AB_2801035 (0.75µl per 1x10⁶ cells per 100µl staining volume)
TotalSeqC anti-human Hashtag 6 Antibody BioLegend Cat# 394671 RRID:AB_2820042 (0.75µl per 1x10⁶ cells per 100µl staining volume)
TotalSeqC anti-human Hashtag 7 Antibody BioLegend Cat# 394673 RRID:AB_2820043 (0.75µl per 1x10⁶ cells per 100µl staining volume)

Validation

All commercially available antibodies are commonly used clones with extensive validation, the relevant literature being detailed on the supplier website. A validation statement from BioLegend: Antibody validation is a critical step in the journey towards obtaining consistent reproducibility in science. To ensure they are both specific and sensitive, we validate our antibodies through a variety of methods including:

- Testing on multiple cell and tissue types with a variety of known expression levels.
- Validation in multiple applications as a cross-check for specificity and to provide additional clarity for researchers.
- Comparison to existing antibody clones.
- Using cell treatments to modulate target expression, such as phosphatase treatment to ensure phospho-antibody specificity. (www.biolegend.com/en-gb/bio-bits/highly-specific-validated-antibodies)

Where relevant, we further confirmed specificity by assessing staining in the expected cell types and tissue structures, and by comparing with transcript expression profiles in our datasets. All antibodies were additionally tested on full-thickness intestinal sections, both inflamed and non-inflamed, with staining conditions optimised through serial dilutions and pH adjustments around the supplier's recommended conditions. Specificity was further supported by validating the distribution of stromal markers, for example by confirming absence of staining in epithelial cells.

# Eukaryotic cell lines

Policy information about cell lines and Sex and Gender in Research

Cell line source(s)

HEK293 cells were obtained  from ATCC (#CRL-1573).

Authentication

HEK293 cells were authenticated by the supplier  (ATCC #CRL-1573).

| Mycoplasma contamination | HEK293 cells were tested mycoplasma-free. |
|---|---|
| Commonly misidentified lines<br>(See ICLAC register) | *Name any commonly misidentified cell lines used in the study and provide a rationale for their use.* |

# Clinical data

Policy information about clinical studies

All manuscripts should comply with the ICMJE guidelines for publication of clinical research and a completed CONSORT checklist must be included with all submissions.

| Clinical trial registration | NA |
|---|---|
| Study protocol | NA |
| Data collection | NA |
| Outcomes | NA |

# Plants

| Seed stocks | NA |
|---|---|
| Novel plant genotypes | NA |
| Authentication | NA |

# Flow Cytometry

## Plots

Confirm that:

☒ The axis labels state the marker and fluorochrome used (e.g. CD4-FITC).

☒ The axis scales are clearly visible. Include numbers along axes only for bottom left plot of group (a 'group' is an analysis of identical markers).

☒ All plots are contour plots with outliers or pseudocolor plots.

☒ A numerical value for number of cells or percentage (with statistics) is provided.

## Methodology

| Sample preparation | For flow cytometry sorting samples were processed identically to those described in methods section. Following this they were then washed with staining buffer (PBS with 2% BSA and 0.01%Tween) and were then stained with appropriate antibodies (Flow cytometry/CITE-seq) at pre-optimized concentrations for 30 minutes. Samples were then washed and sorted directly as described. Prior to running samples compensations were calculated with an unstained cellular control and compensation beads (BD). |
|---|---|
| Instrument | BD FACS Aria Illu, BD FACS Fusion (Sorting) |
| Software | BD FACS Diva Software v9.0, with Quantification performed using FlowJo v10.8.1 |
| Cell population abundance | CD45+, CD45-EPCAM+ or CD45-EPCAM- abundance: 10-90% of parent population, depending on degree of inflammation and tissue source material.<br>As all sorted cells were subject to single cell analysis, we were able to validate transcriptionally that there was no unexpected cellular contamination. |
| Gating strategy | CD45+Sort: FSC-SSC-> Singlets -> Live Dead -> EPCAM-CD45+<br>CD45-EPCAM- Sort: FSC-SSC-> Singlets -> Live Dead -> EPCAM-CD45- |

☒ Tick this box to confirm that a figure exemplifying the gating strategy is provided in the Supplementary Information.

