## [Peer Review File · Nature]

Spatial Fibroblast Niches Defining Crohn's Fistulae

Corresponding Author: Professor Alison Simmons

Version 0:

Reviewer comments:

Referee #1

(Remarks to the Author)

In their manuscript titled "Spatially Resolved Insights Into Fistulating Crohn's Disease Pathogenesis", McGregor et al utilized scRNAseq and high resolution spatial profiling to characterize a robust set of fistula samples from inflammatory bowel disease patients. The goal of the work was to define the cellular and molecular interactions that typify and potentially drive the formation and maintenance of the painful and debilitating Crohn's fistula. Their tissue samples included a total of 67 fistulas from 43 patients; most of which originated from patients with Crohn's disease and their analysis included scRNAseq on 33 full-thickness sections from 22 donors (129.2K cells across immune, epithelial and stromal lineages and identified 17 broad cell populations), Visium on 34 sections (93K spots sequenced) and Xenium on 47 sections (6.3M cells imaged). Based on this unprecedented dataset, this entirely observational but still exciting and technically elegant study succeeds in illuminating cell types, subsets, states, locations and interaction partners as well as growth factor signaling pathways that are dysregulated in fistulas, with a particular focus on fistula-associated stromal (FAS) cells. Interestingly, FAS cells appear to be similar to previously characterized inflammatory or activated fibroblast populations that are associated with ulceration in IBD. Using single cell resolution spatial profiling, the authors go further to define the localization patterns of FAS subsets across fistula topography and speculate on their developmental origins as well as their contributions to the generation and maintenance of fistulae. The state-of-the-art spatial characterization of CD fistulae presented in this body of work certainly provides novel and foundational observations to expand and accelerate the field. However, the manuscript provides minimal functional validation or mechanistic insight and the comparative analysis of FAS with previously reported fibroblast populations was incomplete. Furthermore, the authors overstate claims throughout the paper. To be further considered for publication the following points would need to be addressed:

Major Points:

It is unclear why the manuscript focuses almost entirely on fibroblasts rather than on all cell types given the authors sequenced and analyzed epithelial cells and immune cells in addition to stromal cells. If the authors opt to maintain a fibroblast focus then the title of the manuscript should be changed accordingly.

The subheading "scRNAseq identifies submucosal fibroblast drivers of remodeling and fibrosis in Crohn's fistulae" is one of many examples where the authors overstate their findings. Without providing any functional or mechanistic evidence the writing will need to be significantly modified to temper statements and claims.

Direct and deeper comparisons with recent key studies (e.g. Levantovsky, Med, 2024; Mukherjee, Gastroenterology; 2023) should be carried out where possible and discussed.

Previous reports have shown exquisite differences in cell composition and states along different regions of the GI tract (e.g. Anders, Sci Reports, 2021; Elmentaite, Nature, 2021; Burclaff, CMGH, 2022). The manuscript at hand profiles a diverse set of fistula samples along the GI tract, originating from different diseases (IBD, diverticulitis) and mention that "principal component analysis of both Visium and Xenium cohorts confirmed strong samples segregation by fistula subtypes, with epithelialisation and location as the primary drivers of variability". While there is merit to a broad sampling approach, using the dataset to define region-specific differences may nevertheless yield important insights.

The manuscript states that control fistulae from diverticulitis samples were similar to non-perianal fistulae from CD samples. This is surprising as diverticulitis and IBD have different disease manifestations and the authors showed how different fistula samples separate based on location and epithelialization. Is this statement suggesting that when location and epithelialization status are matched, fistula between IBD and diverticulitis samples are similar?

In Figure 3E, the authors highlight multiple subsets of the FAS state based on location using spatial xenium data. However, it is unclear what individual genes or signatures were used to define these states. The authors should provide greater clarity on their method for identifying FAS subtypes, gene signatures, and the degree of similarity and difference between FAS subsets.

Figure 4H attempts to contrast FAS with previously described inflammatory fibroblasts. However, multiple genes on either side of the heatmap have been previously attributed to inflammatory fibroblasts found in IBD. As such, it is unclear what signature was used here to define the inflammatory fibroblast state. To better understand how FAS from CD fistulae relate to inflammatory fibroblasts from other IBD tissues, the authors should perform a meta-analysis comparing FAS with gene signatures of inflammatory fibroblasts from multiple publications (including but not limited to: Levantovsky et al, *Med* 2024, Thomas et al., *Nat Immunol* 2024; Mukherjee et al, *Gastroenterology* 2023, Kong et al, *Immunity* 2023; Friedrich et al, *Nat Med* 2021; Martin et al *Cell* 2019, Smilie et al *Cell* 2020). Markers used by the authors to define FAS have previously been attributed to inflammatory fibroblasts in IBD (e.g. IL24, IL11, CXCL1, CXCL10, CXCL2, CXCL5, CHI3L1, POSTN, MMP3, COL7A1). Further comparative studies between FAS, FAS-subsets, and previously characterized FB states in the literature are needed to highlight what (if any) markers, genes or signatures are truly unique to FAS (if any).

If a FAS-specific gene signature can be computed and validated then the authors should ascertain whether FAS are present in diverticulitis and ulcerative colitis (full-thickness sections or pinch biopsies). How do FAS compare across diseases?

In Figure 4H, several genes highlighted in the differential expression analysis are not fibroblast genes (e.g. Jchain is expressed by B cells and Fcgr3b is expressed by myeloid cells). Visium and Xenium data can be limited by segmentation and ability to resolve individual cells. While segmentation works well for large and well defined cell types such as epithelial cells, it is more complex with stromal and immune cell subtypes in ulceration regions or fistula where cells can be packed closely without clear borders. More convincing segmentation and QC metrics should be provided along with orthogonal validation using marker genes of pertinent populations.

The authors analyzed gene regulatory networks to identify transcription factors (TFs) that are specific to FAS and highlight several TFs that are important in tissue patterning. However, several of the examples have unknown or poorly studied functions in the gut (examples brought up in text are MSX2 involved in limb patterning, OSR2 involved in cleft palate, and RUNX2 involved in skeletal morphogenesis). Next the authors claim that together these TFs regulate developmental patterning, tube structure formation, and morphogenesis that suggest a role for FAS in fistula formation. Mechanistic work, such as proliferation, migration, differentiation and/or activation experiments to study the function of the highlighted TFs in gut fibroblasts would bolster the authors' claims and strengthen the manuscript.

The authors state "to validate these FAS gene expression patterns, we assessed TWIST1 expression in a validation cohort using IHC where its expression was increased and localised to the deeper stromal compartment of fistulating CD tissue".

However, it appears that TWIST1 was stained by itself making it impossible to know if FAS cells are TWIST+. As such, the tissue should be co-stained with additional markers of FAS in order to quantitatively assess co-localization with TWIST. Furthermore, in vitro mechanistic studies evaluating TWIST1 function in human primary gut fibroblasts would go a long way toward validating the authors' hypothesis and claims.

The authors state that FAS fibroblasts are "imprinted with Stromal 3-like programs which are fibroblasts typically associated with submucosa and deeper layer tissues" and state that they "confirmed this via a meta-analysis of single cell data". To make this claim, McGregor et al relied on a 5-gene signature where 4/5 genes were shared in their FAS fibroblasts.

Furthermore, multiple genes in stromal 1, 2, and 4 signatures were conserved in FAS fibroblasts including CTSC, TCF21, F3, WNT5A, and TNFSF10. While the hypothesis that submucosal fibroblasts give rise to fistula associated fibroblasts is interesting, the evidence provided is insufficient to support any claim. The manuscript should be adjusted here and in other places to clarify the speculative nature of the interpretations and conclusions.

Minor Points

Add page and line numbers, and label each figure (Fig 1, 2, 3, ...) for ease of review.

Add a cartoon to schematize the new central findings relative to the histopathology of fistula and other hallmark pathological features of CD and UC.

(Remarks on code availability)

Referee #2

(Remarks to the Author)

Summary:

This study links new technologies of spatial transcriptomics to the very Crohn's characteristic, formation of fistulous tracts, a highly morbid complication. It nicely summarizes (top of Fig2) the complex variables which distinguish the various fistulous tracts from a pathophysiologic, gut location (e.g. ileum vs. rectum) perspective.

Strengths of the study include

- Inclusion of large numbers of cases and fistulae, specifically 62 fistulae from 42 patients. They include 11 of their fistulae from non-CD. However, this major strength is somewhat lessened by the overall qualitative feel of their current data presentation. (See 'Weaknesses, suggestions and questions', point #3, below)

- A second major strength is their inclusion of 21 tissue sections analyzed by both Xenium and Visium platforms. However, here again, it would be nice to extend the analysis to include various levels of clustering (e.g. Oliver, *Nature* 2024) to 'push' how comprehensive their custom Xenium panel can recreate the whole transcriptome of Visium. These two platforms trade

off transcriptome comprehensiveness (Visium) against likely superior spatial resolution (Xenium).

Use of other IBD datasets and even diabetic wound healing to replicate their inference wrt Stromal cluster 3.

- Careful validation by 2 senior pathologists and inclusion of very difficult to attain samples (e.g. Fig 4, 5), including 33 full thickness ileal samples from 22 donors.

Their strongest inference deals with their novel classification of fistula-associated zones, between FAS-LAZ (lumen adjacent, active remodeling, neutrophils, macrophages, morphogen signals) vs. FAS-ALC (active lesion core) vs. FAS-FOZ (fibrotic, stabilizes the tract) vs. FAS-LOC (lymphoid organizer). They close the paper with validating this spatially using Picosirius red, making inference wrt collagen type and density.

Weaknesses, suggestions and questions

This is an incredibly ambitious, descriptive effort, whose strength is primarily as a data resource as opposed to biologic inference/advance. Descriptive is not necessarily problematic as no doubt the IBD community (assuming the data are shared with careful annotation/linking of the clinical meta-data) will find these data incredibly informative.

1. A weakness, which the authors thoughtfully list, is that there are no time-course analyses. To a substantial degree, this reflects the incredible difficulty of predicting when fistulae will develop; it is highly stochastic and random in nature. However, this undercuts a bit the late-stage/secondary effect significance of classifying FAS layers. Fistulae likely start with a failure of epithelial healing, which the authors try to address with some of their within and between patient analyses involving epithelialization vs. non-epithelialization of the fistulous tracts; however no major inference/advance is made and in my reading, sampling of the leading edge of the tracts (perhaps the most 'pathophysiologic location'?) does not occur.
2. Epithelialization of the tracts is a secondary (and likely not beneficial) adaptation given the fact that curettage of the tracts (for perianal fistulae) improves fistulae closure (with concomitant biologic usage). The authors should address these factors vis a vis their FAS-classification. (Consider also that mesenchymal stem cell transplants post-curettage showed no benefit over curettage alone in a major European trial).
3. The major area for improvement would be to apply data analyses across samples that fully leverages their impressive sample size. Much of their spatial analyses are within sample in nature. Strongly consider applying SOPA (Blampey et al., Nature Comm 2024) to fully leverage cross-walking between H&E, Xenium, and Visium within and between samples more rigorously. Sopa incorporates Baysor (transcript-based clustering, which they use) as well as other annotation tools downstream.
4. In my reading of the methods, cell segmentation was performed based upon an early Xenium platform method of DAPI staining, followed by software-based extension from the nucleus. Were later cell segmentation markers provided by 10X used? The resolution of the Xenium display items is a bit low, and certainly in a couple of places, overstated with respect to being subcellular.
5. (Minor) In fibroblast sub-analyses, was re-normalization performed after subsetting? Fig 3D: list type of fistulae in legend.
6. (Moderate) Fig 4 legend and result title. Overstatement to say 'origin'. Not clear that once in time observation can rigorously infer cell ontogenies.
7. Consider relegating Fig 1 (H&E, Basic histology) to Supplementary especially given that Fig 2 starts out with such a nice clinical summary of the extremely complex sampling, ileal vs. colonic vs. rectal. Given the present extensive supplementary H&E, not clear why Fig 1 tissues were included to start the manuscript.
8. (Moderate) What is the premise for the controls? While logical to include them (they would likely be criticized if they didn't) it's not clear what the biologic advance, if any, is. In multiple places, Crohn's specific clusters are forwarded, but without clear statistical enrichment, especially given lower sample/cell sizes for the controls. Idiopathic (non-Crohn's) cryptoglandular perianal fistulae are lower (anus region) than most Crohn's fistulae, so it is not clear what the most significant pathophysiologic advance(s) and comparisons are. Same general points wrt diverticular fistulae.
9. (Moderate-major) Consider selective protein markers from adjacent sections, same blocks for some cell biology advances (apoptosis, phospho markers) to further leverage spatial-based inference, advance biologic inference. Do the authors feel that significant advances from their CITE-Seq data were attained?
10. (Minor) Number lines and pages.

(Remarks on code availability)

Referee #3

(Remarks to the Author)

In this paper, McGregor et al. present a comprehensive and highly detailed analysis of the cellular landscapes of fistulas using diverse spatial transcriptomics approaches and single cell RNAseq. These devastating lesions, common in Crohn's and other diseases, have not been systematically characterized at the spatial transcriptomics level. The authors use state of the art techniques and analyses and provide a fantastic resource that will be highly informative to the community. The authors identify several Fistula-associated stromal (FAS) subtypes. They utilize their transcriptomics analyses to identify key transcription factors, such as TWIST1, that could drive these stromal programs. Using spatial analysis (Xenium) they identify specific zonation patterns of distinct FAS subtypes. They also describe molecular and cellular differences between fistulas and ulcers and between epithelialized vs. non-epithelialized fistula tracts, identified metaplastic epithelial cells, and correlated the molecular signatures with morphological features of the ECM. The data and analyses are high quality. I have a few comments that should be addressed in the revision:

- The authors nicely present zonation of distinct cell types, however they should complement this analysis with an unbiased quantitation of the expression of all genes as a function of the distance from the luminal border of the fistula. This could be

done on both the Visium and Xenium datasets in the subset of samples where the structure is well delineated.

- Morphogenic pathway analyses – this part is confusing, WNT2 is a canonical WNT ligand whereas WNT5A is non-canonical, in the small intestine WNT2 is located at the crypt and is thought to promote proliferation whereas WNT5A localizes at the tips of villi in both mouse (<https://www.nature.com/articles/s41467-020-15714-x>) and human (<https://www.nature.com/articles/s41586-024-07793-3>). This section makes some alternative claims regarding the roles of WNT2 and WNT5A that need to be either revised according to the previous literature or supported by cited references.
- The authors should add quality control metrics for the Visium experiments – for example how many UMIs on average were obtained per spot.
- Figure 1 – define MAT in caption of C rather than E (where it first appears).
- Figure 3 – define the FAS subsets in the figure caption.
- EDF3E(ii-iii) – Quantify the abundance of FAS fibroblasts in the different diabetic study groups (e.g. violinplot of summed expression of FAS markers). The UMAP suggests that there is indeed an enrichment in the DFU but this requires formal quantification.
- EDF5A – mark the histological landmarks (lumen, inner layers). Add colorbar (what is green/purple) and scale bars.
- Figure 4F – define ‘Credible interval of the slope’.
- Provide references for the previously shown CXCL13 adjacency to B cells and CXCL1 and CXCL2 adjacency to neutrophils.
- Figure S3E,F – add percent variability captured by the first two PCs.
- Data was not available (the doi link does not lead to data).
- The supplementary data containing H&E annotated images is highly impressive and informative, enabling orientation within the complex dataset presented.

(Remarks on code availability)

The code and data links were broken.

Version 1:

Reviewer comments:

Referee #1

(Remarks to the Author)

The authors have largely addressed the concerns raised by this reviewer. My only final comment is that the manuscript title should be changed to avoid the suggestion of causality. The abstract, which reflects the major observations described in this outstanding and novel resource, does not suggest causality.

(Remarks on code availability)

Referee #2

(Remarks to the Author)

This resubmission of a spatial transcriptomics effort to advance understanding of fistulating Crohn’s disease has added a substantial amount of new data and has developed a much more logical and compelling progression of main Figures. By so doing, the present submission advances understanding of this unique trait (i.e. fistulating Crohn’s disease); while major questions regarding pathogenesis and treatment of fistulating Crohn’s disease remain, the present manuscript presents a compelling model of pathogenesis, beyond merely serving as a valuable single cell resource for the IBD field generally. In particular, an enormous amount of work bringing together other datasets’ fibroblasts/stromal cells is provided (Supplementary Table 2), which will be extremely useful to the community. However, I question a bit the final column, which places the world’s literature through the somewhat qualitative final column lens (S1-S4, Kinchen 2018), instead of using more quantitative measures such as majority voting in CellTypist; the most valuable data, irregardless, is the primary single cell data, so not necessary to re-analyze/edit Supplementary Table 2, unless the final column can be easily adjusted/quantitated. Much more effective ‘zoom-out’ and ‘zoom-in’ (regions of interest, ROI) images are provided to fully leverage the much finer cell resolution attainable by Xenium. Substantial mechanistic advances linking *OSR2* (newly linking) and *TWIST1* over-expression studies in primary fibroblasts are now provided. Key multicolor fluorescence protein-based validation are now provided for highlighted/emphasized pathways. Regarding their segmentation limitations, I think their responses are transparent and perfectly acceptable, as there are simply profound limitations with the state of cell segmentation now; because of the power of transcriptome scaling, we are finding key transcript-based domain interactions not necessarily solely through the cell segmentation → to transcriptome filter requirement most effective; in other words, biologically meaningful transcript-transcript domains/niches can be reported which are not necessarily restricted to complete cell-cell resolution via segmentation. Below please find some questions and suggested edits/improvements

Fig 1: succinct, summary of cohorts; includes both ileal and perianal fistulae. Text of results concisely summarizes the major sources of variation for the complex sampling strategy. Effective tri-column comparison of scRNA-Seq, ST-Visium; and ST-Xenium.

(moderate) add cell totals across the 3 modalities, possibly via stacked bar graph/supplementary table across samples; the field would find comparative cell estimates between dissociated cell vs. intact tissue with early fixation (Xenium; Visium)

helpful, IMO, recognizing the marked differences in estimated cell totals (100K vs. 8-9 million). Within early fixation approaches comparison between Visium and Xenium should be broadly similar and an overt, high level comparison across individual samples (with the source of variance wrt sampling variability) would be helpful to provide a sense of sample-based rigor. Broad cell count modalities (i.e. stromal, epithelial, immune) would provide key rigor validation for their inferences, assuring that results are consistently observed across samples and perhaps, across platforms.

Fig 2.

(moderate) I find the Kinchen-defined stromal numbers confusing, compounded further by confusing designations stromal 3 (4), stromal 3 (5), etc. It is clear that the fistula-associated stromal (FAS) cells by UMAP are quite separate, and the major goal that this is moving toward is genes with enhanced expression in FAS (Fig 3C). Suggest either conflating these non-FAS designations by sampling source or provide representative gene names in Fig2A.

(minor) line 168; diverticular; 'broadly similar' Either delete or provide greater statistical precision for this statement.

Fig 3. Whole transcriptome (Visium) and transcription factor analyses. Fistula development using TFs, fibroblast functions. Nice co-localization of COL7, Ki67 and POSTN by immunofluorescence. Rigorous inference wrt increased OSR2 (n = 84) expression, justifying subsequent over-expression studies. Over-expression by lentiviral transduction implicates a primary result of OSR2 as driving FAS formation. Nice linkage by RT-PCR of OSR2 over-expression driving an increase in COL7A. Question: in my reading of the Picrosirius red literature, it measures primarily COL1 and COL3 family members. How would the authors link/interpret results from Figure 6 (Picrosirius Red) to the OSR2-COL7A linkage? Are COL7 members typically affiliated with basement membrane/epithelia in their data?

Fig 4. The authors are attempting to find similarities and differences between epithelial ulcers and FAS. The underlying premise, which the authors acknowledge is not established, is the extent to which fistulas arise from ulcers. This is potentially problematic, as the pinpoint mucosal fistulous openings are often qualitatively/visually differ from deep ulcers seen in ileal Crohn's; however, given the study's expansion of the ulcer cohort using the larger (n = 5000 Xenium) new dataset, this new comparison seems appropriate and valuable.

Question/moderate to major suggestions: I could not track down in Figure 4A the corresponding H&E. Furthermore the blue colors between muscularis (mucosae??) vs. macrophages are hard to distinguish. (line 1740, legend). It is quite common for the muscularis mucosae to hypertrophy in chronically inflamed regions. Are you sure that the 'submucosa' designation in main Figure 4A is correct? Possible to add the corresponding H&E?

(minor-moderate) Main 4H; provide confidence estimates of ulcer to fistula

(Minor) Line 45, abstract, I would favor deleting the term 'putative precursor' as this is a bit too speculative, in my opinion (minor-moderate) in the cNMF analyses, I could not find a parameter/number of gene expression programs to error rates in order to justify the selected number of parameters/expression programs. Can this be provided in supplementary?

Fig 5. Epithelialization within fistulous tracts

Question: Because they observe squamous only with the cutaneous/externally directed fistulae, are they able with the Visium/whole transcriptome transcription factor expression identify differentiating factors between classic squamous vs. columnar epithelial cell differentiation? Do they have any dentate line (e.g. from their non-Crohn's anorectal fistulae) based data?

Fig 6. Tract evolution and stabilization. Col 1 and Col 3 vs. Col7

Mild-moderate suggestion. Provide supplementary table of features and clusters of Picrosirium red staining

(minor) can you link by color legend the clusters (Fig 6C) with designations in Fig 6D so that we can visually determine what the significant differences are?

Clarifying suggestions:

It took me awhile to understand the distinction between Extended data (attached to main Figure) vs. Supplementary figures; not necessarily ordered within the text (?). This challenge was compounded by the fact that (at least when I expanded the compressed folder) the Supplementary Figures were neither in order nor titled. Regarding the main result text, in some cases, it might make sense for the Supplementary Figures to go out of order, but with this massive amount of data, I had trouble reviewing the Supplementary Figures; this challenge was doubly difficult because for many of the main text results, large stretches of text-based results exclusively refer to Supplementary Figures.

Broad impact suggestions

I would have generally favored casting a broader net wrt adaptive immunity results presentation but I certainly understand the authors' myeloid-epithelial-stromal focus. Given what will inevitably be a very high interest by adaptive cell-focused immunologists (e.g. FAS-LOC zones), it is essential that the data be easily shared/downloaded and that the matching code generating the figures be provided.

Possible to provide H&E and other visual tools through an image repository? Linking to the ST data where illustrative and possible?

(Remarks on code availability)

Referee #3

(Remarks to the Author)

The authors have addressed all of my comments with new analyses and text. I recommend publication of this interesting and important work.

(Remarks on code availability)

RESPONSE TO REVIEWERS for Nature manuscript 2024-12-27425

We thank all the reviewers for their helpful and thoughtful review of this work. We have now revised the manuscript incorporating all the changes requested with the inclusion of substantial new data. A summary of key new data/revisions are as follows:

- New data:
 - 5K Xenium spatial transcriptomics from fistulae, ulcers and healthy controls
 - Additional Xenium sections from diverticular fistulae
 - Cell dive multiplexed protein marker data
 - TWIST1 and CD45 IHC co-stain
 - Lentiviral transduction of primary intestinal fibroblasts with TWIST1 and OSR2 and functional analysis by:
 - qPCR, morphology images, bulk RNA-Seq, protein expression of COL7A, α -SMA, F-actin
- Public data:
 - Meta analysis of publicly available data expanded to 487 patient samples
 - Additional table summarising marker genes linked to pathogenic fibroblasts reported for CD and UC and their relationship to our meta analysis.
- Additional analyses:
 - QC metrics from spatial data
 - Cell segmentation QC metrics from spatial data
 - Distance-based gradients from spatial data
 - Extended tissue location matched comparisons between CD fistulae and diverticular disease fistulae
 - Factor analysis, cross-comparing single cell meta analysis and spatial datasets
 - Cell abundance analysis of diabetic foot ulcer dataset
- Updated analyses:
 - Clustering, visualisation, QC, PCA, cell type abundance, differential expression analyses etc. all updated to include additional diverticular control samples that were integrated into the original data. This does not change the original conclusions.
- Reviewer only data:
 - Integrative analysis of visium and xenium – sections are not immediately consecutive; therefore image registration is not working well enough to include in the manuscript
 - Wound healing assay preliminary results
- Text changes:
 - The first section describing the patient cohort has been shortened, and original **Figure 1** has been moved to **Extended Data Figure 1**.
 - The meta analysis section has been updated with new results under its own header

- Revised section titles to not imply mechanistic findings where appropriate
- Revised the text in the morphogen section
- Additional section added in the text describing multiplexed IF analysis
- Additional section added in the text describing in vitro functional experiments
- Additional section added in the text describing additional ulcer vs fistula analysis
- Other smaller text changes to address individual comments or accommodate new data

Referee #1 (Remarks to the Author):

In their manuscript titled "Spatially Resolved Insights Into Fistulating Crohn's Disease Pathogenesis", McGregor et al utilized scRNAseq and high resolution spatial profiling to characterize a robust set of fistula samples from inflammatory bowel disease patients. The goal of the work was to define the cellular and molecular interactions that typify and potentially drive the formation and maintenance of the painful and debilitating Crohn's fistula. Their tissue samples included a total of 67 fistulas from 43 patients; most of which originated from patients with Crohn's disease and their analysis included scRNAseq on 33 full-thickness sections from 22 donors (129.2K cells across immune, epithelial and stromal lineages and identified 17 broad cell populations), Visium on 34 sections (93K spots sequenced) and Xenium on 47 sections (6.3M cells imaged). Based on this unprecedented dataset, this entirely observational but still exciting and technically elegant study succeeds in illuminating cell types, subsets, states, locations and interaction partners as well as growth factor signaling pathways that are dysregulated in fistulas, with a particular focus on fistula-associated stromal (FAS) cells. Interestingly, FAS cells appear to be similar to previously characterized inflammatory or activated fibroblast populations that are associated with ulceration in IBD. Using single cell resolution spatial profiling, the authors go further to define the localization patterns of FAS subsets across fistula topography and speculate on their developmental origins as well as their contributions to the generation and maintenance of fistulae. The state-of-the-art spatial characterization of CD fistulae presented in this body of work certainly provides novel and foundational observations to expand and accelerate the field. However, the manuscript provides minimal functional validation or mechanistic insight and the comparative analysis of FAS with previously reported fibroblast populations was incomplete. Furthermore, the authors overstate claims throughout the paper. To be further considered for publication the following points would need to be addressed:

Major Points:

It is unclear why the manuscript focuses almost entirely on fibroblasts rather than on all cell types given the authors sequenced and analyzed epithelial cells and immune cells in addition to stromal cells. If the authors opt to maintain a fibroblast focus then the title of the manuscript should be changed accordingly.

While we agree that the primary focus of the manuscript is on the role of fibroblasts, we present data from all compartments, as well as making all the data openly available as a major resource for future studies. We have carried out in depth clustering and cell subtype annotations for all of the key cell type lineages in all of the datasets presented in the manuscript. Indeed, a significant amount of space is already dedicated to the role of epithelial cells in fistula tract re-epithelisation, as well as macrophage states that are linked to fistula tracts. We focused on fibroblasts and the cells they interact with as these cells consistently showed the most over-represented signal in the data. As such we have changed the title of the manuscript as recommended.

The subheading “scRNAseq identifies submucosal fibroblast drivers of remodeling and fibrosis in Crohn’s fistulae’ is one of many examples where the authors overstate their findings. Without providing any functional or mechanistic evidence the writing will need to be significantly modified to temper statements and claims.

We have revised the subheadings and wording throughout the manuscript to accurately represent our findings. Furthermore, we have also added additional functional experiments—please see details below.

Direct and deeper comparisons with recent key studies (e.g. Levantovsky, *Med*, 2024; Mukherjee, *Gastroenterology*; 2023) should be carried out where possible and discussed.

We have now undertaken a large scale meta analysis of 11 scRNA-Seq studies from CD and UC, having collected and extracted fibroblast data from 487 patients. The results are presented in **Extended Data Figure 4** and **Supplementary Figure S7** and in a new expanded section of text. We were unable to obtain the Mukherjee *et al* dataset, as the dataset is not publicly available – and it was not possible to align our institutions with regards to the terms of a data sharing agreement. Therefore, the comparison in the manuscript with these data remains qualitative. To address this more thoroughly, as well as other cases where we could not access datasets used in publications, we have created a Supplementary Table contrasting our meta-analysis results with the markers of inflammatory fibroblasts described in these studies. Please see a more in-depth response below.

Previous reports have shown exquisite differences in cell composition and states along different regions of the GI tract (e.g. Anders, *Sci Reports*, 2021; Elmentaite, *Nature*, 2021; Burclaff, *CMGH*, 2022). The manuscript at hand profiles a diverse set of fistula samples along

the GI tract, originating from different diseases (IBD, diverticulitis) and mention that “principal component analysis of both Visium and Xenium cohorts confirmed strong samples segregation by fistula subtypes, with epithelisation and location as the primary drivers of variability”. While there is merit to a broad sampling approach, using the dataset to define region-specific differences may nevertheless yield important insights.

We thank the reviewer for the important suggestion and indeed acknowledge the importance of regional differences in the intestine. These considerations led us to include control samples from small and large intestine in our analysis. In the revised manuscript, we have included additional analyses comparing cell type abundance differences between fistulae found in different locations, controlling for epithelialisation status. These results are provided in **Supplementary Figure 9**. While regional differences provide a very interesting area to explore, unfortunately due to space constraints in the manuscript, we were unable to dedicate further space to extended discussion on this topic.

The manuscript states that control fistulae from diverticulitis samples were similar to non-perianal fistulae from CD samples. This is surprising as diverticulitis and IBD have different disease manifestations and the authors showed how different fistula samples separate based on location and epithelialization. Is this statement suggesting that when location and epithelialization status are matched, fistula between IBD and diverticulitis samples are similar?

To help address this question, we have expanded our original Xenium dataset to include another 6 diverticular disease fistula samples from 5 donors, as we wanted to ensure that any analyses were sufficiently well powered. This brings up the total of diverticular fistula control samples to 11 samples from 10 donors. We have sampled both epithelialised and non-epithelialised areas of the fistula tract (please see updated **Figure 1** for sample overview). Shown below is one additional example of our expanded cohort where we have sampled both epithelialised and non-epithelialised parts of the tract, also capturing the area where the fistula tract penetrates through muscularis propria:

Reviewer Only Figure 1. A representative example of a diverticular disease fistula sample included in expanded cohort, where both the epithelialized and non-epithelialized part of the fistula tract is sampled, as indicated with the red outline in H&E image of the sample in panel A. Panel B visualised cell types recovered in the data, while panel C shows strong, localised expression of *WNT5A* along the edge of the fistula tract, as also seen in CD fistulae.

We have integrated these samples into our original dataset and updated all downstream analyses. Updated PCA analysis, carried out on pseudobulk level aggregation for each section and presented in **Supplementary Figure 4**, supports our previous conclusions that the overall main axes of variation partition our fistula samples primarily based on epithelialisation status and location. To investigate further, we carried out pseudobulk differential expression analysis of all colonic fistulae. The reason for this restriction is that while there are substantial differences between for instance, peri-anal CD fistulae and diverticular fistulae, location is a perfect confounder between these pathologies. When comparing colonic fistulae only, we additionally controlled for epithelialisation status in the model formula. This way we have identified only a handful differentially expressed genes between diverticular and external entero/colocutaneous CD fistulae and diverticular vs internal CD fistulae, while a very similar profile of gene expression changes to that seen in CD fistulae was identified when comparing diverticular fistulae with healthy colon. These results are presented in **Supplementary Figure 9**.

As the above remained a high level, pseudobulk analysis, we further carried out differential abundance testing of all cell sub-populations identified in our dataset. In the original manuscript, we had presented results comparing only diverticular fistulae with healthy control samples. We have now updated this analysis to include the additional samples imaged and further include analysis comparing internal and external CD fistulae with diverticular controls. As before, we have controlled for epithelialisation status in the model formula. These results are presented in **Supplementary Figure 9**. For instance, when

comparing diverticular fistula with external, non-peri anal CD fistula, germinal center B-cells were more prevalent in diverticular fistulae. However, overall differences were small and FAS populations, as well as SPP1+ macrophages and other CD-fistula enriched cell types were also represented in diverticular fistulae.

Taken together, this suggests that while there are differences between diverticular disease and other fistula types, many overall cellular stage changes appear to be similar by this analysis indicative of similar molecular processes operative regardless of initiating aetiology.

Finally, while we have added the additional data to the manuscript, we have limited the discussion and further in depth analyses on this topic due to space constraints.

In Figure 3E, the authors highlight multiple subsets of the FAS state based on location using spatial xenium data. However, it is unclear what individual genes or signatures were used to define these states. The authors should provide greater clarity on their method for identifying FAS subtypes, gene signatures, and the degree of similarity and difference between FAS subsets.

FAS subsets were defined based on unbiased transcriptome clustering analysis, without specifically adopting spatial clustering methods. Key markers from this clustering analysis are shown in **Supplementary Figure 10**. The subsets uncovered this way naturally segregated into different spatial regions surrounding within the fistula tracts. We have not included UMAP embeddings of all sub-clustering analyses for different lineage cells due to space constraints.

In the revised manuscript, we have included an additional, Xenium 5K-plex *in situ* dataset. While the primary purpose of this dataset was to further investigate FAS-like cells at the bases of CD ulcers, we have expanded the manuscript with additional figures (**Figure 4, Supplementary Figure 17, 18**) from fibroblast cells from this dataset, as the larger gene panel enabled discovery of additional markers. We have also included a negative matrix factorisation analysis of fibroblast subsets in this dataset to further clarify distinct versus overlapping co-expression patterns within FAS cell subtypes. Our original findings replicated well in this dataset, revealing the same spatial layering of FAS cells.

Figure 4H attempts to contrast FAS with previously described inflammatory fibroblasts. However, multiple genes on either side of the heatmap have been previously attributed to inflammatory fibroblasts found in IBD. As such, it is unclear what signature was used here to define the inflammatory fibroblast state. To better understand how FAS from CD fistulae relate to inflammatory fibroblasts from other IBD tissues, the authors should perform a meta-analysis comparing FAS with gene signatures of inflammatory fibroblasts from multiple publications (including but not limited to: Levantovsky et al, *Med* 2024, Thomas et al., *Nat Immunol* 2024; Mukherjee et al, *Gastroenterology* 2023, Kong et al, *Immunity* 2023;

Friedrich et al, Nat Med 2021; Martin et al Cell 2019, Smilie et al Cell 2020). Markers used by the authors to define FAS have previously been attributed to inflammatory fibroblasts in IBD (e.g. IL24, IL11, CXCL1, CXCL10, CXCL2, CXCL5, CHI3L1, POSTN, MMP3, COL7A1). Further comparative studies between FAS, FAS-subsets, and previously characterized FB states in the literature are needed to highlight what (if any) markers, genes or signatures are truly unique to FAS (if any).

We have first expanded our scRNA-Seq meta analysis to include data from several additional studies. Unfortunately, we were unable to obtain the data from Mukherjee *et al.* as explained previously. Nonetheless we have included fibroblast data from 487 patient samples, spanning small and large intestine; biopsy and full-thickness derived surgical resections.

17 fibroblast sub-clusters were identified via our revised integrative analysis, broadly classified into submucosal, follicular, mucosal or mucosal telocyte/Stromal 2 lineages. As expected, submucosal clusters were more widely represented in full-thickness samples, while mucosal fibroblasts dominated studies using biopsy sampling.

We identified several clusters which were enriched in IBD – either in active UC, active CD or sampled from CD fistulae. “S4” cluster aligned with follicular reticular-like cells described by Kinchen et al 2018 in UC and Elmentaite et al, 2020 in pediatric CD. “S1 CXCL10+” cluster was distinct from “S4” and instead matched CXCL10+CCL19+ fibroblasts described by Korsunsky et al, 2022 in UC, despite sharing several marker genes. Some clusters closely matched fibroblast subsets described in the literature as disease-associated, but these were not significant in our meta analysis. For instance, Humprey’s *et al* 2024 describe LUM+ fibroblasts, but the key markers align most closely with homeostatic mucosal and submucosal subsets. Mukherjee et al describe MMP+/WNT5+ fibroblasts, which are consistent with disease enriched S1-CHI3L1+ and S2-CHI3L1+ clusters; however, they also describe CXCL14/ADAMDEC+ and CXCL14/F3/PDGFR+ subsets as disease-linked, which correspond to baseline mucosal fibroblasts and telocytes respectively. Inflammation-Associated Fibroblasts described in UC by Smilie et al, 2019 corresponded to S2-CHI3L1+ cluster in our meta-analysis.

There were four clusters that shared marker genes between previously described IBD linked subsets and with FAS cells from our study. We called these cells S1-CHI3L1+, S2-CHI3L1+, S3-CHI3L2+ and S3-FAP+. The latter two clusters were enriched in fistulating disease samples when compared to both active UC and non-fistulating CD, while the former two were often identified in biopsy-only studies in both UC and CD, in line with their underlying mucosal (S1 or S2) versus submucosal (S3) identities. **Analysing the gene co-expression displayed by these fibroblast states (Extended Data Figure 4), we observed while they shared overlapping markers, they could be differentiated by co-expression signatures.** For example, all four subsets expressed CXCL8+. However, MMP1 and MMP3, were expressed by S1-CHI3L1+, S2-CHI3L1+ and S3-CHI3L2+, but not S3-FAP+ cells. FAP was expressed by S3-

CHI3L2+ and S3-FAP+ cells. Of note several of the studies that we re-analysed likely did not sample sufficient numbers of fibroblasts to “cluster out” these subsets, and often a reported cluster was split between multiple populations in our analysis.

To understand the differences between FAS and previously described IBD linked fibroblast states better, we undertook factor analysis using consensus non-negative matrix factorisation (cNMF), to identify gene co-expression networks governing cell type identity versus gene activity programs shared by differing fibroblast cell states. While this analysis identified core mucosal, submucosal and telocyte factors as expected, we identified 8 factors which were correlated to IBD-associated fibroblast states.

Factor 14, for instance, captured a core interferon response signature mostly active in S1-CXCL10+ cells. Factor 10 encompassed genes (TNFRSF11B, PHLDA1, CHI3L1, TIMP1, IL1R1, AREG) linked to “Activated fibroblasts” (Mennillo et al, 2024), “inflammation-associated fibroblasts” (Smillie et al, 2019), and “THY1+FAP+PDPN+ fibroblasts” (Thomas et al, 2024). This was active in S1-CHI3L1+ and S2-CHI3L1+ subsets, but not S3-FAP+ nor S3-CHI3L2+. As the authors from Thomas et al, 2024 provided cell annotations, we were able to visualise this observation matched with cell mapping – see figure below, cells in purple (cells from other studies are shown in grey/NA):

Reviewer Only Figure 2. Fibroblast meta analysis UMAP visualisation, with cells from Thomas *et al* study coloured, and all other cells are in grey. Original study cell annotations are forwarded, visualising mapping of THY1+FAP+PDP+ fibroblasts onto mucosal fibroblast cells, outside the main FAS cluster.

The cell state S3-CHI3L2+ akin to the FAS cells described in our study was dominated by four factors – factor 25 (CXCL13, MMP13, CHI3L2, CHI3L1), factor 24 (IL11, IL24, MMP1, MMP3, COL7A1), factor 22 (CXCL8, CXCL5, MMP9, MT1X) and factor 15 (SPARC, FAP, MMP2, various

collagens). Factor 15 was also dominant in S3-FAP3+ cells, while also active in S1-CHI3L1+ and S2-CHI3L1+ cells. Similarly, factor 24 was active in S1-CHI3L1+ and S2-CHI3L1+ cells but not S3-FAP+ cells. Factors 22 and 25 were more restricted to S3-CHI3L2+ cells. Taken together, this suggests that there are multiple gene co-expression networks that become active in inflammation, fibrosis or fistulating disease and that different combinations of these programs contribute to the fibroblast states observed in specific disease contexts.

We used these factors to score fibroblast cell clusters identified in our new spatial analysis (using Xenium 5k-plex) and also carried out a reciprocal analysis where we scored the fibroblast factors identified from spatial data onto our meta analysis. We found FAS-FOZ cells share programs with S3-FAP+ cells, while different S3-CHI3L2+ cell factors correspond to FAS-ALC (e.g. Factor 25) and FAS-LAZ (e.g. Factor 24) cells. Therefore fibroblast profiles detected as one scRNA-Seq cluster by our study and others, actually form spatially-variable functional gradients with respect to fistula lesion edges, as detected by spatial analysis.

We have summarised the previously reported IBD linked fibroblast subtypes from existing literature, disease associations and their key markers in **Supplementary Data Table 2**, while all the meta analyses described above are presented in **Extended Data Figure 4**.

If a FAS-specific gene signature can be computed and validated then the authors should ascertain whether FAS are present in diverticulitis and ulcerative colitis (full-thickness sections or pinch biopsies). How do FAS compare across diseases?

As discussed we included UC data in our scRNA-Seq meta analysis. Differential abundance analysis has highlighted that fistula-specific clusters, although sharing features with similar cells from UC patients, were detected at a very low frequency in UC compared to CD. In the revised manuscript, we have included this analysis in **Extended Data Figure 4 and Supplementary Figure 7**.

Furthermore, we have re-analysed UC, Checkpoint-Inhibitor Induced colitis and healthy tissue sections from a previously published Visium Spatial Transcriptomics dataset from Gupta *et al*, 2024, which includes both biopsy and resection UC samples. Calculating FAS cell factor activity in these sections showed overall much more limited activity of these modules. These results are presented in **Supplementary Figure 7**.

Regarding diverticulitis, while we have expanded our initial cohort by including additional diverticular disease fistula samples, as outlined above, we are unable to access material from non-fistulating diverticulitis and therefore were unable to include these in further analysis to assess whether FAS cell signatures may be present prior to fistula formation. We are also not aware of any previously published, publicly accessible scRNA-Seq datasets with diverticulitis to include in our meta analysis.

In Figure 4H, several genes highlighted in the differential expression analysis are not fibroblast genes (e.g. *Jchain* is expressed by B cells and *Fcgr3b* is expressed by myeloid cells). Visium and Xenium data can be limited by segmentation and ability to resolve individual cells. While segmentation works well for large and well defined cell types such as epithelial cells, it is more complex with stromal and immune cell subtypes in ulceration regions or fistula where cells can be packed closely without clear borders. More convincing segmentation and QC metrics should be provided along with orthogonal validation using marker genes of pertinent populations.

Cell segmentation unfortunately remains an incompletely solved problem in computational analysis of spatial transcriptomics data, despite many recent advances and new methods. It remains incredibly difficult to entirely eliminate small amounts of contaminating signal from nearby cells due to errors in cell segmentation, in particular in regions with very densely packed cells, such as in highly inflamed areas. It is indeed true that for example *JCHAIN* is detected as a differentially expressed gene in this context due to a greater abundance of B/Plasma cells in the immediate neighbourhoods of FAS cells in CD ulcers, further exacerbated by the fact that *JCHAIN* transcript is very abundant. This makes biological sense, as ulcers are mucosal, where we would expect an increase in plasma cells to begin with.

While the field is evolving, we have followed the current best practices by re-segmenting all *in situ* ST data with transcript-density based method cell segmentation to correct these type of artefacts and minimise cell segmentation “doublets” as much as possible. Indeed, in benchmarks of cell segmentation algorithms for *in situ* data, Baysor (our approach) has been demonstrated to achieve high (sometimes highest, depending on evaluation strategies) transcriptome-based metrics for cell segmentation, and is known to skew towards over- rather than under- segmenting cells, and therefore is one of the best current methods to avoid cell segmentation “doublets”. In the revised manuscript, we have included an additional QC figure to show a large improvement in cell segmentation that is achieved when applying transcript density based methods to our data when compared to Xenium on board nuclei-expansion only. These results are presented in **Supplementary Figure 1**.

In the revised manuscript, we have generated additional *in situ* ST data using new, 5100-plex Xenium panel. In this dataset, we used a cell boundary staining approach. While these data already provide much cleaner cell partitioning than nuclei-expansion based methods, it is nonetheless clear that boundary-based cell segmentation remains imperfect, and transcript-density based cell segmentation, using a cell boundary based cell segmentation as a prior, which effectively combines the two approaches, produces the best results. Therefore, throughout the manuscript, this combination approach has been used for all of our *in situ* ST datasets. This is effectively the same approach as has been described with published pipelines such as SOPA – however, this method was not yet published when we started this

project and therefore internally we have implemented what is effectively the same solution as SOPA pipeline.

In the revised manuscript, we have also expanded the methods section to highlight that this process still remains imperfect. However, all data presented in this manuscript will be publicly available, and as new methods emerge, we expect further re-analyses by us and the wider research community may be able to refine the results presented here and find new insights.

The authors analyzed gene regulatory networks to identify transcription factors (TFs) that are specific to FAS and highlight several TFs that are important in tissue patterning. However, several of the examples have unknown or poorly studied functions in the gut (examples brought up in text are MSX2 involved in limb patterning, OSR2 involved in cleft palate, and RUNX2 involved in skeletal morphogenesis). Next the authors claim that together these TFs regulate developmental patterning, tube structure formation, and morphogenesis that suggest a role for FAS in fistula formation. Mechanistic work, such as proliferation, migration, differentiation and/or activation experiments to study the function of the highlighted TFs in gut fibroblasts would bolster the authors' claims and strengthen the manuscript.

We agree with the reviewer and have included new data exploring the role of two FAS expressed TFs, TWIST1 and OSR2. TWIST1 has been previously linked to fibrosis functions in different tissue contexts¹⁻⁵, while the role of OSR2 has not been previously well studied in the context of fibroblasts (or the gut).

We used lentiviral transduction to over-express these TFs in primary human intestinal fibroblasts from five different donors. We could confirm successful TWIST1 and OSR2 over-expression with qPCR.

To ascertain which, if any, FAS cell functions TWIST1 and OSR2 might control, we further measured by qPCR selected cytokines, chemokines, morphogens, matrix-metalloproteases and collagens identified as expressed by FAS cells in our analyses. This analysis suggested that over-expression of OSR2 also induced expression of COL7A1, a key collagen found at the very edge of fistula tracts, as well as other markers of FAS cells, such as MME, but not cytokine or chemokine expression. Thus, it is likely that immune-signalling related functions of FAS cells are not controlled by OSR2.

To investigate this phenotype further, we bulk RNA-sequenced these cells and control samples. TWIST1 over-expression consistently induced many ECM remodelling related pathways and downregulated integrin and cell adhesion pathways. OSR2 overexpression on the other hand, showed more marked changes, including induction of a set of developmental and patterning gene programs.

Furthermore, OSR2 expression resulted in a markedly different cellular morphology of fibroblasts. Cells appeared more rounded and less spindle-like as control fibroblasts. We also observed reduced actin filament staining with Phalloidin, and more specifically, reduced expression of α -SMA. On the other hand, COL7A protein expression, in line with RNA expression data, was increased. These data are presented in **Figure 3** and **Extended Data Figure 7**.

We also explored whether TWIST1 or OSR2 altered the migratory capacity of primary colonic fibroblasts by performing wound healing assays using the ibidi Culture-Insert 2 Well 24 system across patient-derived lines and genotypes over 48–96 hours. SPY-555 DNA dye (Spirochrome) was used to label nuclei for tracking cell movement, and SiR-Actin (Spirochrome) was used to visualise cell boundaries and assess wound closure.

Reviewer Only Figure 3. Fibroblast migration assay imaging results from a single donor.

Although this approach had the potential to yield informative results, several technical limitations impaired interpretability:

1. *Inconsistent migratory behaviour:* Fibroblasts exhibited variable and often limited migration across experimental runs. The cells occasionally formed dense clusters, which appeared to be dependent on cell number and seeding density.
2. *Signal instability during acquisition:* Fluorescent signal intensity declined substantially over time. Loss of nuclear signal at later time points prevented reliable cell tracking, while fluctuations in actin signal hindered delineation of the leading edge. Although medium replenishment can partially mitigate this issue, signal drift will remain a significant challenge.

3. *Difficulties with signal segmentation:* Cells frequently exhibited overlapping growth upon reaching confluence, which complicated segmentation of nuclear and cytoskeletal signals for accurate quantification.
4. *Lack of proliferation control:* The current experimental setup did not allow normalisation of migratory behaviour to account for differences in cell proliferation, further confounding interpretation.
5. *Post-transduction stress response:* In some cases, fibroblasts displayed abnormal DNA staining patterns (e.g. fractured nuclei, cytoplasmic signal), likely reflecting residual stress from lentiviral transduction. While extended culture might restore normal behaviour, this was not feasible within the timeframe of the study.

Given these limitations, we concluded that substantial optimisation would be required to implement this assay effectively and therefore excluded it from the revised manuscript.

The authors state “to validate these FAS gene expression patterns, we assessed TWIST1 expression in a validation cohort using IHC where its expression was increased and localised to the deeper stromal compartment of fistulating CD tissue”. However, it appears that TWIST1 was stained by itself making it impossible to know if FAS cells are TWIST+. As such, the tissue should be co-stained with additional markers of FAS in order to quantitatively assess co-localization with TWIST. Furthermore, *in vitro* mechanistic studies evaluating TWIST1 function in human primary gut fibroblasts would go a long way toward validating the authors’ hypothesis and claims.

We have now included a tissue co-stain with CD45 to show that TWIST1 staining in these tissue sections is found in the stromal and not the immune compartment **Supplementary Figure 6**.

We have evaluated TWIST1 function *in vitro* in intestinal fibroblasts, as above.

The authors state that FAS fibroblasts are “imprinted with Stromal 3-like programs which are fibroblasts typically associated with submucosa and deeper layer tissues” and state that they “confirmed this via a meta-analysis of single cell data”. To make this claim, McGregor et al relied on a 5-gene signature where 4/5 genes were shared in their FAS fibroblasts. Furthermore, multiple genes in stromal 1, 2, and 4 signatures were conserved in FAS fibroblasts including CTSC, TCF21, F3, WNT5A, and TNFSF10. While the hypothesis that submucosal fibroblasts give rise to fistula associated fibroblasts is interesting, the evidence provided is insufficient to support any claim. The manuscript should be adjusted here and in other places to clarify the speculative nature of the interpretations and conclusions.

We agree that definitive proof would require lineage tracing experiments, not feasible within the scope of this work. As such, we have qualified the results section in the manuscript to highlight this limitation.

However, based on the scRNA-Seq meta-analysis (**Supplementary Figure 7, Extended Data Figure 4**), we have shown that cNMF analysis factors that define mucosal fibroblast identity (S1 and S2) are not found in FAS cells, but rather in mucosal cells found in UC and CD that share some transcriptional programs with FAS cells. This may indeed suggest a shared ontogeny, and FAS cells could be a more “advanced” state of these cells; Conversely, it is also possible that these shared activity programs just become active in different lineage cells.

Minor Points

Add page and line numbers, and label each figure (Fig 1, 2, 3, ...) for ease of review.

Apologies for the oversight, page and figures have been appropriately labelled in the resubmitted manuscript.

Add a cartoon to schematize the new central findings relative to the histopathology of fistula and other hallmark pathological features of CD and UC.

We have added a schematic summary to **Extended Data Figure 10** summarising key findings on the manuscript.

Referee #2 (Remarks to the Author):

Summary:

This study links new technologies of spatial transcriptomics to the very Crohn’s characteristic, formation of fistulous tracts, a highly morbid complication. It nicely summarizes (top of Fig2) the complex variables which distinguish the various fistulous tracts from a pathophysiologic, gut location (e.g. ileum vs. rectum) perspective.

Strengths of the study include

- Inclusion of large numbers of cases and fistulae, specifically 62 fistulae from 42 patients. They include 11 of their fistulae from non-CD. However, this major strength is somewhat lessened by the overall qualitative feel of their current data presentation. (See ‘Weaknesses, suggestions and questions’, point #3, below)

- A second major strength is their inclusion of 21 tissue sections analyzed by both Xenium and Visium platforms. However, here again, it would be nice to extend the analysis to include various levels of clustering (e.g. Oliver, Nature 2024) to ‘push’ how comprehensive their custom Xenium panel can recreate the whole transcriptome of Visium. These two platforms trade off transcriptome comprehensiveness (Visium) against likely superior spatial resolution (Xenium).

Use of other IBD datasets and even diabetic wound healing to replicate their inference wrt Stromal cluster 3.

- Careful validation by 2 senior pathologists and inclusion of very difficult to attain samples (e.g. Fig 4, 5), including 33 full thickness ileal samples from 22 donors.

Their strongest inference deals with their novel classification of fistula-associated zones, between FAS-LAZ (lumen adjacent, active remodeling, neutrophils, macrophages, morphogen signals) vs. FAS-ALC (active lesion core) vs. FAS-FOZ (fibrotic, stabilizes the tract) vs. FAS-LOC (lymphoid organizer). They close the paper with validating this spatially using Picosirius red, making inference wrt collagen type and density.

Weaknesses, suggestions and questions

This is an incredibly ambitious, descriptive effort, whose strength is primarily as a data resource as opposed to biologic inference/advance. Descriptive is not necessarily problematic as no doubt the IBD community (assuming the data are shared with careful annotation/linking of the clinical meta-data) will find these data incredibly informative.

1. A weakness, which the authors thoughtfully list, is that there are no time-course analyses. To a substantial degree, this reflects the incredible difficulty of predicting when fistulae will develop; it is highly stochastic and random in nature. However, this undercuts a bit the late-stage/secondary effect significance of classifying FAS layers. Fistulae likely start with a failure of epithelial healing, which the authors try to address with some of their within and between patient analyses involving epithelialization vs. non-epithelialization of the fistulous tracts; however no major inference/advance is made and in my reading, sampling of the leading edge of the tracts (perhaps the most 'pathophysiologic location'?) does not occur.

We agree that it is indeed very difficult to "stage" fistulae in terms of progression and to sample areas where the fistula are just beginning to emerge. Indeed, due to extremely heterogeneous presentations and large-scale morphogenic remodelling of the affected tissues, it is not trivial identifying just fistula tract itself, its location, direction and other tissue landmarks. In our original cohort, we had the opportunity to obtain one sample from an early fistula/fissuring ulcer, where the lesion had progressed past the muscularis mucosa into the submucosa, but the leading edge had not yet reached muscularis propria. In our experience, this is a very rare specimen – we were able to identify only one case with such clear presentation after thorough histological analysis of our entire tissue bank.

To further explore the origins of fistula tracts in our revised manuscript we undertook an in depth analysis of Crohn's ulcers, from which fistulae are thought to emerge. We present a more in depth analysis of CD ulcers and fistula tracts incorporating an additional *in situ* spatial transcriptomics dataset using a high-plex (Xenium 5000 plex + 100 custom FAS targets) panel. In this dataset, we specifically micro-dissected tissue from CD surgical samples around ulcers, sampling a further cohort of n=7 ulcers, n=7 healthy controls and

n=5 CD fistulae (~ 1.8 million post-QC cells imaged). This approach enabled us to specifically target these lesions directly, enriching for cells involved in pathology, while at the same time a large target panel provides opportunity for less biased discovery.

Using these data, we could conclude that:

- FAS-like cells are present in the bases on all ulcers we sampled but at much lower abundance to that observed in fistula tracts, suggesting their presence alone is not sufficient for fistula formation
- Comparing the micro-environments between FAS cells in fistulae and ulcers, we found many key differences. These are presented in **Figure 4** and **Extended Data Figure 8**.
- Undertaking differential expression analysis of cells matched from appropriate lesion locations (e.g. active ulcer lesion area with fistula edge, fibrotic outer zone with deeper tissue areas in CD), we found that there are strong differences in these fibroblasts. In fistulae, fibroblasts induced additional ECM remodelling programs and perhaps most interestingly, patterning-related TFs (SNAI1, SNAI2, PRRX1, SOX4 and HOXA10). This suggests that although greatly similar to wound-healing phenotypes of fibroblasts in ulcers, FAS cells additionally acquire specific invasive properties.
- Together with the new data from an *in vitro* study of the effects of TWIST1 and OSR2 in intestinal fibroblasts, where we found that OSR2 induces some of these developmental programs, we can begin to map the sequence of steps in how these cells become dysregulated.

2. Epithelialization of the tracts is a secondary (and likely not beneficial) adaptation given the fact that curettage of the tracts (for perianal fistulae) improves fistulae closure (with concomitant biologic usage). The authors should address these factors vis a vis their FAS-classification. (Consider also that mesenchymal stem cell transplants post-curettage showed no benefit over curettage alone in a major European trial).

We agree that re-epithelialisation of fistula tracts is not a desirable outcome. Indeed, our motivation for sampling of fistula tract areas where epithelialisation is incomplete/in progress has been to better understand the process of epithelialisation over the surface of fistula tract. We find FAS cells exhibit very localised NRG1, IL24 and WNT signalling directly adjacent to sites of re-epithelialisation supportive of a role for FAS cells in driving this process. We have revised the manuscript to further cite relevant literature as suggested.

3. The major area for improvement would be to apply data analyses across samples that fully leverages their impressive sample size. Much of their spatial analyses are within sample in nature. Strongly consider applying SOPA (Blampey et al., Nature Comm 2024) to fully leverage cross-walking between H&E, Xenium, and Visium within and between samples

more rigorously. Sopa incorporates Baysor (transcript-based clustering, which they use) as well as other annotation tools downstream.

All of the analyses presented in the manuscript are not within sample, but from fully integrated data. This includes:

- All cell-level integration, dimensionality reduction, clustering, sub-clustering and cell type annotation
- All niche analysis and domain detection was carried out on integrated data to arrive at joint, cross-sample niches
- All co-localisation and local neighbourhood analyses and associated statistics have been carried out on integrated data
- All factor analyses have been carried out on integrated data
- All differential expression and differential abundance analyses were carried out on integrated data, blocking for sample-level or other co-variates where required
- Fistula-edge distance-based calculations in the revised manuscript were carried out across all sections - the standard error around the density plots for example, summarise sample-to-sample variation

Unfortunately, due to a large number of tissue samples profiled, we can only display selected samples for visualisation as representative of the cohort. All samples are made available in supplementary data as integrated data objects, and their histology summarised in Supplementary Data. We have made this clearer in the updated manuscript methods section.

Unfortunately, SOPA was not yet published at the time we started our project – however, we have implemented and arrived at a very similar in-house pipeline for many aspects of the data analysis, including as the reviewer notes, using baysor for cell segmentation. Re-running the entire analysis using SOPA implementation is unlikely to yield any differences, as our workflow is already very similar.

Regarding the suggestion to integrate Xenium and Visium samples, we have attempted an integrated analysis using STAlign⁶, which enables mapping of Xenium data onto Visium space through large deformation diffeomorphic metric mapping. However, while we have a substantial overlap in samples between the two cohorts, the tissue sections themselves are not immediately adjacent – several slices were taken for various optimisations from the same block and therefore the mapping is very distorted as tissue morphology at smaller scales changes a lot. The example below shown in **Reviewer Only Figure 4** is one of our best aligned tissue slices between two platforms, and while general expression patterns hold, the xenium projection is quite distorted. We do not believe these alignments are of sufficiently high quality to be taken forward for downstream analysis and therefore we have opted to not include these data in the revised manuscript.

Review Only Figure 4. A. Original xenium data from a sample slide paired with Visium data shown in B. C. H&E image from Visium data shown in B. D. Xenium data aligned to Visium section using ST align. E. Example gene expression of CXCL8 gene, which shows strong central localised expression in both Xenium (E) and Visium (F) datasets. G. STAligned Xenium expression of CXCL8, showing spatially distorted expression with respect to original datasets.

4. In my reading of the methods, cell segmentation was performed based upon an early Xenium platform method of DAPI staining, followed by software-based extension from the nucleus. Were later cell segmentation markers provided by 10X used? The resolution of the Xenium display items is a bit low, and certainly in a couple of places, overstated with respect to being subcellular.

The original 480-plex sample cohort is indeed imaged using only DAPI staining for cell segmentation. When we started the project, cell segmentation boundary staining was not yet commercially available, and for the sake of consistency within the cohort and to minimise batch effects, we did not change our protocol throughout. It is fairly well established that nuclei-based expansion results in overall poor cell segmentation, creating a lot of transcript mixing and artificial “segmentation doublets”. However, we have taken extra steps, as described in the methods section, to post-process the data to improve cell

segmentation using a transcript density based method, baysor. We have tested and optimised various parameters (prior weights on nuclei segmentation, cell size, cluster numbers etc) to achieve the best segmentation metrics for our dataset (cluster silhouette score, mutually exclusive marker co-expression rate). Our pipeline shows a lot of improvement over nuclei segmentation only – these data have now been included in the revised manuscript in **Supplementary Figure 1**.

Finally, in the revised manuscript, we have included an additional Xenium 5k-plex dataset. As these data were not intended for direct comparison with the previous dataset, we have included the cell segmentation markers in the assay. However, we found that purely image marker based cell segmentation remains inaccurate, and we further post-processed these data with baysor.

However, cell segmentation in spatial transcriptomics remains an incompletely solved problem. We expect our datasets could be reanalysed in the future as the field evolves for more accurate inference. We have since tested several emerging new methods (e.g. cellpose, segger, proseg) in house and have not been able to find one that substantially improves upon our original combination approach of boundary staining and transcript-density based post-processing.

5. (Minor) In fibroblast sub-analyses, was re-normalization performed after subsetting?

Yes, all data were reprocessed after subset isolation for both single cell and spatial workflows.

Fig 3D: list type of fistulae in legend.

The legend has been corrected appropriately.

6. (Moderate) Fig 4 legend and result title. Overstatement to say 'origin'. Not clear that once in time observation can rigorously infer cell ontogenies.

This has been revised and the overall section in the results revised, including with the addition of new spatial dataset.

7. Consider relegating Fig 1(H&E, Basic histology) to Supplementary especially given that Fig 2 starts out with such a nice clinical summary of the extremely complex sampling, ileal vs. colonic vs. rectal. Given the present extensive supplementary H&E, not clear why Fig 1 tissues were included to start the manuscript.

We have taken this advice on board and moved Figure 1 to Extended Data instead. We have also shortened the results section in the main text accordingly. Given the new data included in the revised manuscript, this was necessary to also make space for additional findings.

8. (Moderate) What is the premise for the controls? While logical to include them (they would likely be criticized if they didn't) it's not clear what the biologic advance, if any, is. In multiple places, Crohn's specific clusters are forwarded, but without clear statistical enrichment, especially given lower sample/cell sizes for the controls. Idiopathic (non-Crohn's) cryptoglandular perianal fistulae are lower (anus region) than most Crohn's fistulae, so it is not clear what the most significant pathophysiologic advance(s) and comparisons are. Same general points wrt diverticular fistulae.

We thank the reviewer for raising this important point. Our rationale for including multiple control types was to strengthen comparative inference, despite differences in sampling sizes across cohorts. At the outset of the project, it was unclear to us the extent of heterogeneity and divergence between different sample groups we could expect to observe, as no similar dataset exists. However, we agree that control sample sizes were much smaller compared to the CD fistulae samples and therefore we could not be confident in our conclusions.

Therefore, in the revised manuscript we have:

1. **Expanded diverticular disease controls:** We have now incorporated additional diverticular fistula samples into our comparative analyses (see also response to Reviewer 1), to increase statistical power and further probe whether the fibroblast programs we describe are unique to Crohn's-associated fistulae or shared across distinct aetiologies of fistulising disease.
2. **Additional dataset from inflammatory CD, targeting CD ulcers specifically:** We have expanded our Crohn's disease control cohort, particularly focusing on inflamed ulcers using a high plex 5100 gene panel more focused on discovery. This allowed us to more precisely examine FAS fibroblasts at ulcer bases and those in fistulising lesions. These comparisons provide additional support for a model in which FAS cells emerge in deep ulceration and persist through fistula formation.
3. **Non-Crohn's perianal fistulae:** We acknowledge the small size of this cohort, which was limited to our Visium data. While underpowered for statistical comparisons, we included these samples to provide full transparency and share all available datasets with the research community. However, we have refrained from over-interpreting these data and have not drawn strong biological conclusions from this subset.

9. (Moderate-major) Consider selective protein markers from adjacent sections, same blocks for some cell biology advances (apoptosis, phospho markers) to further leverage spatial-based inference, advance biologic inference. Do the authors feel that significant advances from their CITE-Seq data were attained?

We appreciate the reviewer's suggestion to deepen spatial inference through targeted protein-level analyses. To this end, we undertook multiplex immunofluorescence (IF)

imaging on sections from the same tissue blocks used for transcriptomic profiling, including n=7 fistulae, n=3 inflammatory Crohn's control sections, and n=3 healthy ileum samples.

Our panel included:

- **Lineage and morphological markers:** CD45, Vimentin, PanCK, Neutrophil Elastase, E-cadherin, Pan-cadherin, Na⁺/K⁺-ATPase, DAPI, RPS6
- **ECM and FAS markers:** COL7A1 (lesion-specific), POSTN (fibrotic zone)
- **Functional readouts:** Cleaved CASP3 (apoptosis), KI67 (proliferation)
- **Pathway and FAS identity markers:** WNT5A, F3, and transcription factors TWIST1, PRRX2, RUNX2, OSR2

Although time constraints limited full antibody optimisations, particularly for WNT5A and the transcription factors, which showed variable or non-specific staining, we obtained several informative findings:

- COL7A1 and POSTN marked spatially distinct ECM microenvironments within fistula tracts, with COL7A1 localising to active lesion edges.
- F3 protein was deposited in discrete perilesional zones, consistent with the spatial localisation of FAS fibroblasts identified in our transcriptomic data. Remarkably, while cellular expression (protein and RNA) was more widespread, extracellular deposition was very precisely controlled and restricted to lesion edges.

These findings substantiate transcriptomic inferences and suggest that key aspects of FAS cell behaviour – that is, ECM remodelling, epithelial interactions, and possible roles in lesion organisation-can be captured and spatially contextualised at the protein level. These results have been included in the manuscript in updated results section and in **Figure 3** and **Extended Data Figure 6**.

10. (Minor) Number lines and pages.

Apologies for this oversight, this has been corrected in resubmitted manuscript.

Referee #3 (Remarks to the Author):

In this paper, McGregor et al. present a comprehensive and highly detailed analysis of the cellular landscapes of fistulas using diverse spatial transcriptomics approaches and single cell RNAseq. These devastating lesions, common in Crohn's and other diseases, have not been systematically characterized at the spatial transcriptomics level. The authors use state of the art techniques and analyses and provide a fantastic resource that will be highly informative

to the community. The authors identify several Fistula-associated stromal (FAS) subtypes. They utilize their transcriptomics analyses to identify key transcription factors, such as TWIST1, that could drive these stromal programs. Using spatial analysis (Xenium) they identify specific zonation patterns of distinct FAS subtypes. They also describe molecular and cellular differences between fistulas and ulcers and between epithelialized vs. non-epithelialized fistula tracts, identified metaplastic epithelial cells, and correlated the molecular signatures with morphological features of the ECM. The data and analyses are high quality. I have a few comments that should be addressed in the revision:

- The authors nicely present zonation of distinct cell types, however they should complement this analysis with an unbiased quantitation of the expression of all genes as a function of the distance from the luminal border of the fistula. This could be done on both the Visium and Xenium datasets in the subset of samples where the structure is well delineated.

This is a great suggestion from the reviewer. We have now included this analysis in the updated manuscript in **Figure 2** and **Extended Data Figure 5 and 9**, where we examined both gene expression gradients as well as FAS and macrophage cell subset distributions along these axes. Additionally, we have further analysed the gradient of epithelial cell type distributions in partially epithelialized fistula, as we go further away from the leading edge, in order to quantify precisely where the regenerative zone ends and how soon new homeostasis is established away from the leading, regenerative edge.

- Morphogenic pathway analyses – this part is confusing, WNT2 is a canonical WNT ligand whereas WNT5A is non-canonical, in the small intestine WNT2 is located at the crypt and is thought to promote proliferation whereas WNT5A localizes at the tips of villi in both mouse (<https://www.nature.com/articles/s41467-020-15714-x>) and human (<https://www.nature.com/articles/s41586-024-07793-3>). This section makes some alternative claims regarding the roles of WNT2 and WNT5A that need to be either revised according to the previous literature or supported by cited references.

We have revised and clarified this section in the results section accordingly.

- The authors should add quality control metrics for the Visium experiments – for example how many UMIs on average were obtained per spot.

We have included quality control metrics from Visium experiments in **Supplementary Figure 1**. We have added UMI counts, but specifically excluded other common metrics like mitochondrial percentage. The dataset is generated using FFPE probe-based Visium protocol and probes for highly abundant and mitochondrial genes are deliberately excluded from the assay and therefore cannot be used for QC as counts are effectively zero throughout.

- Figure 1 – define MAT in caption of C rather than E (where it first appears).

We have expanded figure legend accordingly.

- Figure 3 – define the FAS subsets in the figure caption.

We have expanded figure legend accordingly.

- EDF3E(ii-iii) – Quantify the abundance of FAS fibroblasts in the different diabetic study groups (e.g. violinplot of summed expression of FAS markers). The UMAP suggests that there is indeed an enrichment in the DFU but this requires formal quantification.

We have carried out differential abundance analysis using miloR, which does not require formal clustering analysis of the data, comparing healers vs non-healers, and indeed there is a significant enrichment of these cells in healing ulcers. We have also scored these cells using factors derived from cNMF analysis in order to standardise these analyses and make them more comparable. Furthermore, we have included additional violin plots per patient group in order to better visualise these differences. The updated results are presented in **Supplementary Figure 7**.

- EDF5A – mark the histological landmarks (lumen, inner layers). Add colorbar (what is green/purple) and scale bars.

We have added missing scale and colour bars to the figure and annotated the sections for clarity.

- Figure 4F – define ‘Credible interval of the slope’.

We have defined this in the figure legend. In broad terms, this is equivalent to effect size/log fold change.

- Provide references for the previously shown CXCL13 adjacency to B cells and CXCL1 and CXCL2 adjacency to neutrophils.

- We have added further references supporting chemotactic functions of these ligands.

- Figure S3E,F – add percent variability captured by the first two PCs.

The panels have been updated.

- Data was not available (the doi link does not lead to data).

We have updated the data links to contain URLs as well as DOIs. In our testing, these appear to be working.

- The supplementary data containing H&E annotated images is highly impressive and informative, enabling orientation within the complex dataset presented.

We thank the reviewer for their comment and believe the assembled images together with histological annotation will be a useful resource for the community.

References:

1. Wu, Q. *et al.* Twist1 regulates macrophage plasticity to promote renal fibrosis through galectin-3. *Cell Mol Life Sci* **79**, 137 (2022).
2. Zhang, Y. *et al.* TWIST1+FAP+ fibroblasts in the pathogenesis of intestinal fibrosis in Crohn's disease. *J Clin Invest* **134**, (2024).
3. Bridges, R. S. *et al.* Gene expression profiling of pulmonary fibrosis identifies Twist1 as an antiapoptotic molecular 'rectifier' of growth factor signaling. *American Journal of Pathology* **175**, 2351–2361 (2009).
4. Sun, L. *et al.* Transcription factor Twist1 drives fibroblast activation to promote kidney fibrosis via signaling proteins Prrx1/TNC. *Kidney Int* **106**, 840–855 (2024).
5. Ning, X., Zhang, K., Wu, Q., Liu, M. & Sun, S. Emerging role of Twist1 in fibrotic diseases. *J Cell Mol Med* **22**, 1383 (2018).
6. Clifton, K. *et al.* STalign: Alignment of spatial transcriptomics data using diffeomorphic metric mapping. *Nature Communications* 2023 14:1 **14**, 1–14 (2023).

Response to reviewers for 2024-12-27425A

We thank the reviewers for their thoughtful comments throughout the review process, which have helped to greatly improve our manuscript. Please see further responses to individual comments below.

Referees' comments:

Referee #1 (Remarks to the Author):

The authors have largely addressed the concerns raised by this reviewer. My only final comment is that the manuscript title should be changed to avoid the suggestion of causality. The abstract, which reflects the major observations described in this outstanding and novel resource, does not suggest causality.

We have now re-worded the title of the manuscript to avoid invoking causality.

Referee #2 (Remarks to the Author):

This resubmission of a spatial transcriptomics effort to advance understanding of fistulating Crohn's disease has added a substantial amount of new data and has developed a much more logical and compelling progression of main Figures. By so doing, the present submission advances understanding of this unique trait (i.e. fistulating Crohn's disease); while major questions regarding pathogenesis and treatment of fistulating Crohn's disease remain, the present manuscript presents a compelling model of pathogenesis, beyond merely serving as a valuable single cell resource for the IBD field generally.

In particular, an enormous amount of work bringing together other datasets' fibroblasts/stromal cells is provided (Supplementary Table 2), which will be extremely useful to the community. However, I question a bit the final column, which places the world's literature through the somewhat qualitative final column lens (S1-S4, Kinchen 2018), instead of using more quantitative measures such as majority voting in CellTypist; the most valuable data, irregardless, is the primary single cell data, so not necessary to re-analyze/edit Supplementary Table 2, unless the final column can be easily adjusted/quantitated.

While we agree that it would be exceptionally helpful to expand the meta analysis with more precise comparisons with original publications, we cannot directly map all labels between fibroblast clusters as reported in their original publications due to lack of publicly available cell annotations for all datasets. While several of the studies in the meta analysis do provide these in an accessible format, for others only raw or unannotated data has been deposited. Therefore, any majority voting approach would be restricted to the studies where cell annotations are accessible, which does not represent an unbiased slice through the literature. This is further complicated by the fact that we could not obtain the raw or processed data from several key studies, as previously discussed, and could only make qualitative comparisons as presented in the table based on the reported marker gene tables reported in the original publications.

With regards to nomenclature, in the revised manuscript we have ensured that fibroblast subtypes are first described in the main text using functional/location-based terms (i.e. mucosal fibroblasts, telocytes, submucosal fibroblast) to avoid any confusion. We chiefly use corresponding S1-S4 labels as a compact short-hand for figures and tables. Finally, we believe that introducing yet another set of nomenclature in this manuscript while the Gut Cell

Atlas working group is close to finalising community-agreed gut cell ontology would only cause further confusion.

Finally, we provide the meta analysis fibroblast integrated data object with annotations for future re-use, which has been uploaded to Mendeley Data.

Much more effective 'zoom-out' and 'zoom-in' (regions of interest, ROI) images are provided to fully leverage the much finer cell resolution attainable by Xenium. Substantial mechanistic advances linking OSR2 (newly linking) and TWIST1 over-expression studies in primary fibroblasts are now provided. Key multicolor fluorescence protein-based validation are now provided for highlighted/emphasized pathways. Regarding their segmentation limitations, I think their responses are transparent and perfectly acceptable, as there are simply profound limitations with the state of cell segmentation now; because of the power of transcriptome scaling, we are finding key transcript-based domain interactions not necessarily solely through the cell segmentation → to transcriptome filter requirement most effective; in other words, biologically meaningful transcript-transcript domains/niches can be reported which are not necessarily restricted to complete cell-cell resolution via segmentation. Below please find some questions and suggested edits/improvements.

Fig 1: succinct, summary of cohorts; includes both ileal and perianal fistulae. Text of results concisely summarizes the major sources of variation for the complex sampling strategy. Effective tri-column comparison of scRNA-Seq, ST-Visium; and ST-Xenium. (moderate) add cell totals across the 3 modalities, possibly via stacked bar graph/supplementary table across samples; the field would find comparative cell estimates between dissociated cell vs. intact tissue with early fixation (Xenium; Visium) helpful, IMO, recognizing the marked differences in estimated cell totals (100K vs. 8-9 million). Within early fixation approaches comparison between Visium and Xenium should be broadly similar and an overt, high level comparison across individual samples (with the source of variance wrt sampling variability) would be helpful to provide a sense of sample-based rigor. Broad cell count modalities (i.e. stromal, epithelial, immune) would provide key rigor validation for their inferences, assuring that results are consistently observed across samples and perhaps, across platforms.

We have now added overall cell counts to the overview figure, which has been moved to **Extended Data Figure 1** in the revised manuscript. We further provide a table with more detailed cell count statistics per group in a table in **Supplementary Information**. Larger tables with counts per sample have been uploaded to Mendeley Data.

Fig 2.

(moderate) I find the Kinchen-defined stromal numbers confusing, compounded further by confusing designations stromal 3 (4), stromal 3 (5), etc. It is clear that the fistula-associated stromal (FAS) cells by UMAP are quite separate, and the major goal that this is moving toward is genes with enhanced expression in FAS (Fig 3C). Suggest either conflating these non-FAS designations by sampling source or provide representative gene names in Fig2A.

We agree that these designations are potentially confusing. We have renamed the stromal 3 cell clusters from the scRNAseq dataset to use key marker genes instead throughout all the figures in the revised manuscript. While all stromal 3 sub-clusters are broadly submucosal fibroblasts, these subpopulations represent additional functional and often spatial diversity – for example, our spatial data suggests that S3 KCNN3+ fibroblasts (as per newly revised designation) localise in and around muscularis propria specifically and express gene

expression programs related to neuronal and muscle cross-talk. While these variations are interesting, they represent normal submucosal fibroblast diversity and therefore due to space constraints, we opted not to discuss this at length in the manuscript and focus on disease-linked states instead.

(minor) line 168; diverticular; 'broadly similar' Either delete or provide greater statistical precision for this statement.

We have edited this section in the revised manuscript.

Fig 3. Whole transcriptome (Visium) and transcription factor analyses. Fistula development using TFs, fibroblast functions. Nice co-localization of COL7, Ki67 and POSTN by immunofluorescence. Rigorous inference wrt increased OSR2 (n = 84) expression, justifying subsequent over-expression studies. Over-expression by lentiviral transduction implicates a primary result of OSR2 as driving FAS formation. Nice linkage by RT-PCR of OSR2 over-expression driving an increase in COL7A.

Question: in my reading of the Picosirius red literature, it measures primarily COL1 and COL3 family members. How would the authors link/interpret results from Figure 6 (Picosirius Red) to the OSR2-COL7A linkage? Are COL7 members typically affiliated with basement membrane/epithelia in their data?

Indeed, picosirius red measures fibrillar Type I and III collagens. In line with this, we observe clear spatial mutually exclusive pattern between COL7 expression at fistula lesion edges and picosirius red staining outside these tract areas, where we believe the active remodelling and more established fibrotic zones change over. The exact function of COL7 itself is unclear in fistulae - much of the existing literature indeed focuses on functions related to securing the basement membrane at epithelial-stromal junctions, in line with COL7 mutations/deficiencies causing dystrophic epidermolysis bullosa that results in fragile skin and chronic, non-healing wounds. However, this is hard to square off against our data, as we typically see strong expression at non-epithelialised fistula edges where there is no established basement membrane. One interpretation of this would be that COL7 expression happens in preparation for tract re-epithelialisation; however, the highly disordered deposition patterns suggest that this process could be dysregulated in fistulae and could be a factor in whether ultimately a fistula tract will re-epithelialize or not. However, while it's an interesting hypothesis, it remains highly speculative and further functional work would be required to better understand these dynamics that are beyond the scope of the current manuscript. We have included a brief statement to this effect in the revised manuscript.

Fig 4. The authors are attempting to find similarities and differences between epithelial ulcers and FAS. The underlying premise, which the authors acknowledge is not established, is the extent to which fistulas arise from ulcers. This is potentially problematic, as the pinpoint mucosal fistulous openings are often qualitatively/visually differ from deep ulcers seen in ileal Crohn's; however, given the study's expansion of the ulcer cohort using the larger (n = 5000 Xenium) new dataset, this new comparison seems appropriate and valuable.

Question/moderate to major suggestions: I could not track down in Figure 4A the corresponding H&E. Furthermore the blue colors between muscularis (mucosae??) vs. macrophages are hard to distinguish. (line 1740, legend). It is quite common for the muscularis mucosae to hypertrophy in chronically inflamed regions. Are you sure that the

'submucosa' designation in main Figure 4A is correct? Possible to add the corresponding H&E?

The corresponding H&E is included in **Supplementary Data**, on page 26, where we have now added more detailed histopathologist annotations. In this particular section with severe inflammation, there is indeed focal hypertrophy of muscularis mucosa, with variable thickness throughout the profiled ROI – it is thinner in the middle of the section around ulceration, and thicker outside, which has made our initial annotations somewhat misleading. We have now added the missing muscularis mucosa annotation to the image to help clarify this.

(Minor) Line 45, abstract, I would favor deleting the term 'putative precursor' as this is a bit too speculative, in my opinion

We have removed the term putative precursor from the abstract.

(minor-moderate) in the cNMF analyses, I could not find a parameter/number of gene expression programs to error rates in order to justify the selected number of parameters/expression programs. Can this be provided in supplementary?

We have now included additional factor QC plots in **Supplementary Information**.

Fig 5. Epithelialization within fistulous tracts

Question: Because they observe squamous only with the cutaneous/externally directed fistulae, are they able with the Visium/whole transcriptome transcription factor expression identify differentiating factors between classic squamous vs. columnar epithelial cell differentiation? Do they have any dentate line (e.g. from their non-Crohn's anorectal fistulae) based data?

The differences between squamous and columnar epithelial cells are indeed very striking in our data. We observe distinct clusters separating these regions both in Visium and at single-cell resolution in Xenium, with clear gradients within squamous epithelium. In fact, two key clusters we annotated in the Xenium dataset correspond to basal and superficial squamous epithelial cells. Our analysis also highlights transcription factor differences, including members of the grainyhead family (e.g. *GRHL3*, enriched in squamous epithelium), while canonical intestinal epithelial TFs, such as *CDX2*, appear to columnar epithelium restricted.

Reviewer-Only Figure 1. *Top – External CD enterocutaneous fistula with columnar and squamous epithelium. Bottom - expression of columnar specific CDX2 and squamous specific GRHL3 transcription factors.*

We note, however, that we do not have healthy control skin samples from matched regions. Our dataset also does not include any control (healthy or non-CD) dentate line samples, so we cannot directly address transitional epithelium at this boundary. Because the squamous epithelium we capture is invariably from fistula tracts, sometimes directly merging with intestinal columnar epithelium in entero/colocutaneous fistulae, it is difficult to determine how representative these are of normal squamous epithelium. While this is a limitation, it is also a fascinating feature of the fistula anatomy, and while squamous epithelial cell differentiation examination is beyond the scope of the current manuscript, we anticipate future meta-analyses across datasets will be highly informative on this topic.

Fig 6. Tract evolution and stabilization. Col 1 and Col 3 vs. Col7

Mild-moderate suggestion. Provide supplementary table of features and clusters of Picrosirium red staining

We have deposited the full quantifications for all features for each image patch/region to **Mendeley Data**, linked under **Data Availability** statement, as this dataset is quite large. The heatmaps presented in the manuscript provide a cluster-level summary of the features.

(minor) can you link by color legend the clusters (Fig 6C) with designations in Fig 6D so that we can visually determine what the significant differences are?

We agree that this would greatly aid clarity. We have added a matching colour key in the revised manuscript.

Clarifying suggestions:

It took me awhile to understand the distinction between Extended data (attached to main Figure) vs. Supplementary figures; not necessarily ordered within the text (?). This challenge was compounded by the fact that (at least when I expanded the compressed folder) the Supplementary Figures were neither in order nor titled. Regarding the main result text, in some cases, it might make sense for the Supplementary Figures to go out of order, but with this massive amount of data, I had trouble reviewing the Supplementary Figures; this challenge was doubly difficult because for many of the main text results, large stretches of text-based results exclusively refer to Supplementary Figures.

We have heavily revised and re-organised the extended and supplementary data in the revised manuscript to align with the now much shortened version of the text to meet editorial guidelines. We hope the layout is now much clear and more logical.

Broad impact suggestions

I would have generally favored casting a broader net wrt adaptive immunity results presentation but I certainly understand the authors' myeloid-epithelial-stromal focus. Given what will inevitably be a very high interest by adaptive cell-focused immunologists (e.g. FAS-LOC zones), it is essential that the data be easily shared/downloaded and that the matching code generating the figures be provided.

Possible to provide H&E and other visual tools through an image repository? Linking to the ST data where illustrative and possible?

All datasets are provided in Mendeley Data repository together with full annotations in easily accessible Seurat data object RDS files. These are in a ready to be queried format with linked meta data and annotations, which we hope will facilitate re-use by the community. We have additionally exported ST overlay images for all samples in our dataset, uploaded together with the processed data as well as imaging data. Analysis code is available on github.

Finally, we have developed a manuscript companion data portal app freely accessible at <https://simmonslab.shinyapps.io/cd-fistula-data-portal/> which enables users to browse all spatial transcriptomics datasets in this manuscript via an interactive web interface. We will further aim to deposit these datasets to cellxgene, once the support for in situ spatial datasets is available.

Referee #3 (Remarks to the Author):

The authors have addressed all of my comments with new analyses and text. I recommend publication of this interesting and important work.

Response to reviewers for 2024-12-27425A

We thank the reviewers for their thoughtful comments throughout the review process, which have helped to greatly improve our manuscript. Please see further responses to individual comments below.

Referees' comments:

Referee #1 (Remarks to the Author):

The authors have largely addressed the concerns raised by this reviewer. My only final comment is that the manuscript title should be changed to avoid the suggestion of causality. The abstract, which reflects the major observations described in this outstanding and novel resource, does not suggest causality.

We have now re-worded the title of the manuscript to avoid invoking causality.

Referee #2 (Remarks to the Author):

This resubmission of a spatial transcriptomics effort to advance understanding of fistulating Crohn's disease has added a substantial amount of new data and has developed a much more logical and compelling progression of main Figures. By so doing, the present submission advances understanding of this unique trait (i.e. fistulating Crohn's disease); while major questions regarding pathogenesis and treatment of fistulating Crohn's disease remain, the present manuscript presents a compelling model of pathogenesis, beyond merely serving as a valuable single cell resource for the IBD field generally.

In particular, an enormous amount of work bringing together other datasets' fibroblasts/stromal cells is provided (Supplementary Table 2), which will be extremely useful to the community. However, I question a bit the final column, which places the world's literature through the somewhat qualitative final column lens (S1-S4, Kinchen 2018), instead of using more quantitative measures such as majority voting in CellTypist; the most valuable data, irregardless, is the primary single cell data, so not necessary to re-analyze/edit Supplementary Table 2, unless the final column can be easily adjusted/quantitated.

While we agree that it would be exceptionally helpful to expand the meta analysis with more precise comparisons with original publications, we cannot directly map all labels between fibroblast clusters as reported in their original publications due to lack of publicly available cell annotations for all datasets. While several of the studies in the meta analysis do provide these in an accessible format, for others only raw or unannotated data has been deposited. Therefore, any majority voting approach would be restricted to the studies where cell annotations are accessible, which does not represent an unbiased slice through the literature. This is further complicated by the fact that we could not obtain the raw or processed data from several key studies, as previously discussed, and could only make qualitative comparisons as presented in the table based on the reported marker gene tables reported in the original publications.

With regards to nomenclature, in the revised manuscript we have ensured that fibroblast subtypes are first described in the main text using functional/location-based terms (i.e. mucosal fibroblasts, telocytes, submucosal fibroblast) to avoid any confusion. We chiefly use corresponding S1-S4 labels as a compact short-hand for figures and tables. Finally, we believe that introducing yet another set of nomenclature in this manuscript while the Gut Cell

Atlas working group is close to finalising community-agreed gut cell ontology would only cause further confusion.

Finally, we provide the meta analysis fibroblast integrated data object with annotations for future re-use, which has been uploaded to Mendeley Data.

Much more effective 'zoom-out' and 'zoom-in' (regions of interest, ROI) images are provided to fully leverage the much finer cell resolution attainable by Xenium. Substantial mechanistic advances linking OSR2 (newly linking) and TWIST1 over-expression studies in primary fibroblasts are now provided. Key multicolor fluorescence protein-based validation are now provided for highlighted/emphasized pathways. Regarding their segmentation limitations, I think their responses are transparent and perfectly acceptable, as there are simply profound limitations with the state of cell segmentation now; because of the power of transcriptome scaling, we are finding key transcript-based domain interactions not necessarily solely through the cell segmentation → to transcriptome filter requirement most effective; in other words, biologically meaningful transcript-transcript domains/niches can be reported which are not necessarily restricted to complete cell-cell resolution via segmentation. Below please find some questions and suggested edits/improvements.

Fig 1: succinct, summary of cohorts; includes both ileal and perianal fistulae. Text of results concisely summarizes the major sources of variation for the complex sampling strategy. Effective tri-column comparison of scRNA-Seq, ST-Visium; and ST-Xenium. (moderate) add cell totals across the 3 modalities, possibly via stacked bar graph/supplementary table across samples; the field would find comparative cell estimates between dissociated cell vs. intact tissue with early fixation (Xenium; Visium) helpful, IMO, recognizing the marked differences in estimated cell totals (100K vs. 8-9 million). Within early fixation approaches comparison between Visium and Xenium should be broadly similar and an overt, high level comparison across individual samples (with the source of variance wrt sampling variability) would be helpful to provide a sense of sample-based rigor. Broad cell count modalities (i.e. stromal, epithelial, immune) would provide key rigor validation for their inferences, assuring that results are consistently observed across samples and perhaps, across platforms.

We have now added overall cell counts to the overview figure, which has been moved to **Extended Data Figure 1** in the revised manuscript. We further provide a table with more detailed cell count statistics per group in a table in **Supplementary Information**. Larger tables with counts per sample have been uploaded to Mendeley Data.

Fig 2.

(moderate) I find the Kinchen-defined stromal numbers confusing, compounded further by confusing designations stromal 3 (4), stromal 3 (5), etc. It is clear that the fistula-associated stromal (FAS) cells by UMAP are quite separate, and the major goal that this is moving toward is genes with enhanced expression in FAS (Fig 3C). Suggest either conflating these non-FAS designations by sampling source or provide representative gene names in Fig2A.

We agree that these designations are potentially confusing. We have renamed the stromal 3 cell clusters from the scRNAseq dataset to use key marker genes instead throughout all the figures in the revised manuscript. While all stromal 3 sub-clusters are broadly submucosal fibroblasts, these subpopulations represent additional functional and often spatial diversity – for example, our spatial data suggests that S3 KCNN3+ fibroblasts (as per newly revised designation) localise in and around muscularis propria specifically and express gene

expression programs related to neuronal and muscle cross-talk. While these variations are interesting, they represent normal submucosal fibroblast diversity and therefore due to space constraints, we opted not to discuss this at length in the manuscript and focus on disease-linked states instead.

(minor) line 168; diverticular; 'broadly similar' Either delete or provide greater statistical precision for this statement.

We have edited this section in the revised manuscript.

Fig 3. Whole transcriptome (Visium) and transcription factor analyses. Fistula development using TFs, fibroblast functions. Nice co-localization of COL7, Ki67 and POSTN by immunofluorescence. Rigorous inference wrt increased OSR2 (n = 84) expression, justifying subsequent over-expression studies. Over-expression by lentiviral transduction implicates a primary result of OSR2 as driving FAS formation. Nice linkage by RT-PCR of OSR2 over-expression driving an increase in COL7A.

Question: in my reading of the Picosirius red literature, it measures primarily COL1 and COL3 family members. How would the authors link/interpret results from Figure 6 (Picosirius Red) to the OSR2-COL7A linkage? Are COL7 members typically affiliated with basement membrane/epithelia in their data?

Indeed, picosirius red measures fibrillar Type I and III collagens. In line with this, we observe clear spatial mutually exclusive pattern between COL7 expression at fistula lesion edges and picosirius red staining outside these tract areas, where we believe the active remodelling and more established fibrotic zones change over. The exact function of COL7 itself is unclear in fistulae - much of the existing literature indeed focuses on functions related to securing the basement membrane at epithelial-stromal junctions, in line with COL7 mutations/deficiencies causing dystrophic epidermolysis bullosa that results in fragile skin and chronic, non-healing wounds. However, this is hard to square off against our data, as we typically see strong expression at non-epithelialised fistula edges where there is no established basement membrane. One interpretation of this would be that COL7 expression happens in preparation for tract re-epithelialisation; however, the highly disordered deposition patterns suggest that this process could be dysregulated in fistulae and could be a factor in whether ultimately a fistula tract will re-epithelialize or not. However, while it's an interesting hypothesis, it remains highly speculative and further functional work would be required to better understand these dynamics that are beyond the scope of the current manuscript. We have included a brief statement to this effect in the revised manuscript.

Fig 4. The authors are attempting to find similarities and differences between epithelial ulcers and FAS. The underlying premise, which the authors acknowledge is not established, is the extent to which fistulas arise from ulcers. This is potentially problematic, as the pinpoint mucosal fistulous openings are often qualitatively/visually differ from deep ulcers seen in ileal Crohn's; however, given the study's expansion of the ulcer cohort using the larger (n = 5000 Xenium) new dataset, this new comparison seems appropriate and valuable.

Question/moderate to major suggestions: I could not track down in Figure 4A the corresponding H&E. Furthermore the blue colors between muscularis (mucosae??) vs. macrophages are hard to distinguish. (line 1740, legend). It is quite common for the muscularis mucosae to hypertrophy in chronically inflamed regions. Are you sure that the

'submucosa' designation in main Figure 4A is correct? Possible to add the corresponding H&E?

The corresponding H&E is included in **Supplementary Data**, on page 26, where we have now added more detailed histopathologist annotations. In this particular section with severe inflammation, there is indeed focal hypertrophy of muscularis mucosa, with variable thickness throughout the profiled ROI – it is thinner in the middle of the section around ulceration, and thicker outside, which has made our initial annotations somewhat misleading. We have now added the missing muscularis mucosa annotation to the image to help clarify this.

(Minor) Line 45, abstract, I would favor deleting the term 'putative precursor' as this is a bit too speculative, in my opinion

We have removed the term putative precursor from the abstract.

(minor-moderate) in the cNMF analyses, I could not find a parameter/number of gene expression programs to error rates in order to justify the selected number of parameters/expression programs. Can this be provided in supplementary?

We have now included additional factor QC plots in **Supplementary Information**.

Fig 5. Epithelialization within fistulous tracts

Question: Because they observe squamous only with the cutaneous/externally directed fistulae, are they able with the Visium/whole transcriptome transcription factor expression identify differentiating factors between classic squamous vs. columnar epithelial cell differentiation? Do they have any dentate line (e.g. from their non-Crohn's anorectal fistulae) based data?

The differences between squamous and columnar epithelial cells are indeed very striking in our data. We observe distinct clusters separating these regions both in Visium and at single-cell resolution in Xenium, with clear gradients within squamous epithelium. In fact, two key clusters we annotated in the Xenium dataset correspond to basal and superficial squamous epithelial cells. Our analysis also highlights transcription factor differences, including members of the grainyhead family (e.g. *GRHL3*, enriched in squamous epithelium), while canonical intestinal epithelial TFs, such as *CDX2*, appear to columnar epithelium restricted.

Reviewer-Only Figure 1. *Top – External CD enterocutaneous fistula with columnar and squamous epithelium. Bottom - expression of columnar specific CDX2 and squamous specific GRHL3 transcription factors.*

We note, however, that we do not have healthy control skin samples from matched regions. Our dataset also does not include any control (healthy or non-CD) dentate line samples, so we cannot directly address transitional epithelium at this boundary. Because the squamous epithelium we capture is invariably from fistula tracts, sometimes directly merging with intestinal columnar epithelium in entero/colocutaneous fistulae, it is difficult to determine how representative these are of normal squamous epithelium. While this is a limitation, it is also a fascinating feature of the fistula anatomy, and while squamous epithelial cell differentiation examination is beyond the scope of the current manuscript, we anticipate future meta-analyses across datasets will be highly informative on this topic.

Fig 6. Tract evolution and stabilization. Col 1 and Col 3 vs. Col7

Mild-moderate suggestion. Provide supplementary table of features and clusters of Picrosirium red staining

We have deposited the full quantifications for all features for each image patch/region to **Mendeley Data**, linked under **Data Availability** statement, as this dataset is quite large. The heatmaps presented in the manuscript provide a cluster-level summary of the features.

(minor) can you link by color legend the clusters (Fig 6C) with designations in Fig 6D so that we can visually determine what the significant differences are?

We agree that this would greatly aid clarity. We have added a matching colour key in the revised manuscript.

Clarifying suggestions:

It took me awhile to understand the distinction between Extended data (attached to main Figure) vs. Supplementary figures; not necessarily ordered within the text (?). This challenge was compounded by the fact that (at least when I expanded the compressed folder) the Supplementary Figures were neither in order nor titled. Regarding the main result text, in some cases, it might make sense for the Supplementary Figures to go out of order, but with this massive amount of data, I had trouble reviewing the Supplementary Figures; this challenge was doubly difficult because for many of the main text results, large stretches of text-based results exclusively refer to Supplementary Figures.

We have heavily revised and re-organised the extended and supplementary data in the revised manuscript to align with the now much shortened version of the text to meet editorial guidelines. We hope the layout is now much clear and more logical.

Broad impact suggestions

I would have generally favored casting a broader net wrt adaptive immunity results presentation but I certainly understand the authors' myeloid-epithelial-stromal focus. Given what will inevitably be a very high interest by adaptive cell-focused immunologists (e.g. FAS-LOC zones), it is essential that the data be easily shared/downloaded and that the matching code generating the figures be provided.

Possible to provide H&E and other visual tools through an image repository? Linking to the ST data where illustrative and possible?

All datasets are provided in Mendeley Data repository together with full annotations in easily accessible Seurat data object RDS files. These are in a ready to be queried format with linked meta data and annotations, which we hope will facilitate re-use by the community. We have additionally exported ST overlay images for all samples in our dataset, uploaded together with the processed data as well as imaging data. Analysis code is available on github.

Finally, we have developed a manuscript companion data portal app freely accessible at <https://simmonslab.shinyapps.io/cd-fistula-data-portal/> which enables users to browse all spatial transcriptomics datasets in this manuscript via an interactive web interface. We will further aim to deposit these datasets to cellxgene, once the support for in situ spatial datasets is available.

Referee #3 (Remarks to the Author):

The authors have addressed all of my comments with new analyses and text. I recommend publication of this interesting and important work.